# MMMU-Pro: A More Robust Multi-discipline Multimodal Understanding Benchmark

## Abstract

This paper introduces MMMU-Pro, a robust version of the Massive Multi-discipline Multimodal Understanding and Reasoning (MMMU) benchmark. MMMU-Pro rigorously assesses multimodal models' true understanding and reasoning capabilities through a three-step process based on MMMU: (1) filtering out questions answerable by text-only models, (2) augmenting candidate options, and (3) introducing a vision-only input setting where questions are embedded within images. This setting challenges AI to truly "see" and "read" simultaneously, testing *a core human cognitive skill of seamlessly integrating visual and textual information*. Results show that model performance is substantially lower on MMMU-Pro than on MMMU, ranging from 16.8% to 26.9% across models. We explore the impact of OCR prompts and Chain of Thought (CoT) reasoning, finding that OCR prompts have minimal effect while CoT generally improves performance. MMMU-Pro provides a more rigorous evaluation tool, closely mimicking real-world scenarios and offering valuable directions for future multimodal research.[1]

## 1 Introduction

Recent advances in multimodal large language models (MLLMs) have led to remarkable progress in tackling complex reasoning tasks that combine textual and visual information (Yin et al., 2023a; Jin et al., 2024). Models like GPT-4o (OpenAI, 2024a) have achieved impressive results, e.g., on the Massive Multi-discipline Multimodal Understanding and Reasoning (MMMU) benchmark (Yue et al., 2024), reaching an accuracy of 69.1% on college-level questions that integrate text and images.

While these achievements are significant, they raise a critical question: *Do the current benchmark results truly reflect a deep, multifaceted understanding of diverse subjects, or are these models exploiting subtle shortcuts and statistical patterns to arrive at correct answers without genuine comprehension and reasoning?*

This question has profound implications for the development and deployment of AI systems in real-world applications. If models rely on superficial cues rather than true multimodal understanding (Du et al., 2023; Yuksekgonul et al., 2023), we risk overestimating their capabilities and potentially deploying systems that fail in unpredictable ways when faced with novel scenarios (Wu & Xie, 2024; Tong et al., 2024b).

To address this concern and push the boundaries of multimodal AI evaluation, we introduce MMMU-Pro, a more robust and challenging version of the MMMU benchmark. MMMU-Pro is designed to more accurately and rigorously assess a model's true multimodal understanding and reasoning capabilities across a wide range of academic disciplines. The development of MMMU-Pro is motivated by key observations, including the text-only solvability of some benchmark questions, limited option space in multiple-choice formats (Wang et al., 2024), and the need to challenge models' ability to jointly understand different modalities in a more integrated way.

MMMU-Pro employs a rigorous three-step construction process (as shown in Figure 1) that builds upon MMMU (Yue et al., 2024): (1) filtering out questions answerable by text-only language models, (2) augmenting candidate options to reduce the effectiveness of guessing based on the options,

---

[1]All code, data are available at Anonymous Github Link

```
┌─────────────────────────┐   ┌─────────────────────────┐   ┌─────────────────────────┐
│ ① LLM Filtering    ▽    │   │ ② Option Augmentation   │   │ ③ Photos/Screenshots    │
│                         │   │                         │   │                         │
│      🔬 MMMU            │   │    Filtered MMMU        │   │   👥  • Manually take    │
│                         │   │                         │   │          photos          │
│   LLMs                  │   │ Augment up │ Human       │   │                         │
│   w/ text-only          │   │ to 10      │ validation  │   │   🔍  • Synthetic        │
│   input                 │   │ options    │             │   │          artifacts       │
│                         │   │                         │   │                         │
│   Highly image-         │   │ Option augmented        │   │   📊  • Different font    │
│   dependent questions   │   │ questions               │   │          styles          │
└─────────────────────────┘   └─────────────────────────┘   └─────────────────────────┘
```

Figure 1: An overview of the construction process of MMMU-Pro.

and (3) introducing a vision-only input setting (as shown in Figure 3) where models are presented with questions embedded in a screenshot or photo.

The introduction of the vision-only input setting is particularly crucial, as it tests a fundamental human cognitive ability: *the seamless integration and switching between visual and textual information*. This setting challenges models to develop the capability to truly "see" and "read" simultaneously, mirroring how humans effortlessly process complex scenes where text and images are intertwined. This ability is crucial for tasks ranging from interpreting scientific diagrams (Li et al., 2024d) to navigating graphical user interfaces (Liu et al., 2024b; Zheng et al., 2024; Koh et al., 2024). Moreover, this approach aligns with how users naturally interact with AI systems, often sharing screenshots or photos rather than meticulously separating text and images.

Our experimental results demonstrate the effectiveness of MMMU-Pro in providing a more rigorous evaluation of multimodal models. We observe significant performance drops across all tested models when compared to the original MMMU benchmark, with decreases ranging from 16.8% to 26.9%. These results highlight the limitations of current state-of-the-art models in true multimodal understanding and reasoning. Furthermore, our analysis reveals that while CoT (Wei et al., 2022) prompting generally improves performance, the benefits vary across models and settings.

Interestingly, we find that explicit OCR prompts do not significantly impact performance for most models, suggesting that advanced multimodal models have already developed robust text extraction capabilities from images. However, this result also underscores that simple OCR is insufficient for the challenges presented by MMMU-Pro's vision-only input setting. Our further qualitative analysis indicates that when text is embedded within images, it significantly increases the overall complexity of the visual input, requiring models to not only recognize text but also understand its context, relationship to visual elements, and relevance to the question. These findings not only provide a more accurate assessment of current multimodal AI capabilities but also highlight the need for more sophisticated multimodal reasoning abilities.

## 2 MMMU-PRO: A MORE ROBUST VERSION OF MMMU

### 2.1 REVISITING THE MMMU BENCHMARK

The Massive Multi-discipline Multimodal Understanding and Reasoning (MMMU) benchmark (Yue et al., 2024) is a comprehensive dataset designed to evaluate multimodal AI models on college-level tasks that require subject-specific knowledge and deliberate reasoning. MMMU consists of 11.5K

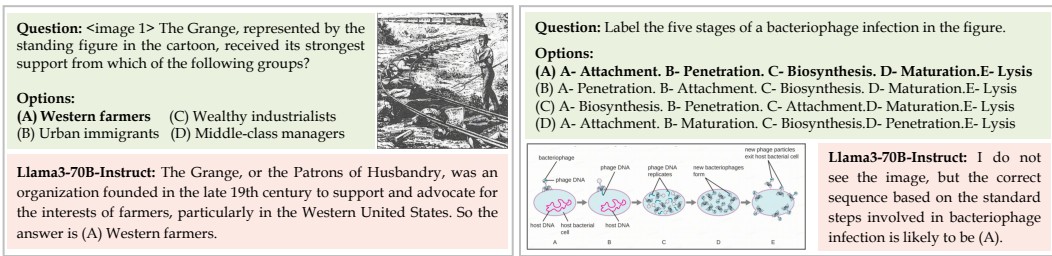

Figure 2: Two MMMU questions that are answered correctly by a text-only LLM Llama-3-70B Instruct. The model finds shortcuts or correlations in the text question and the candidate options.

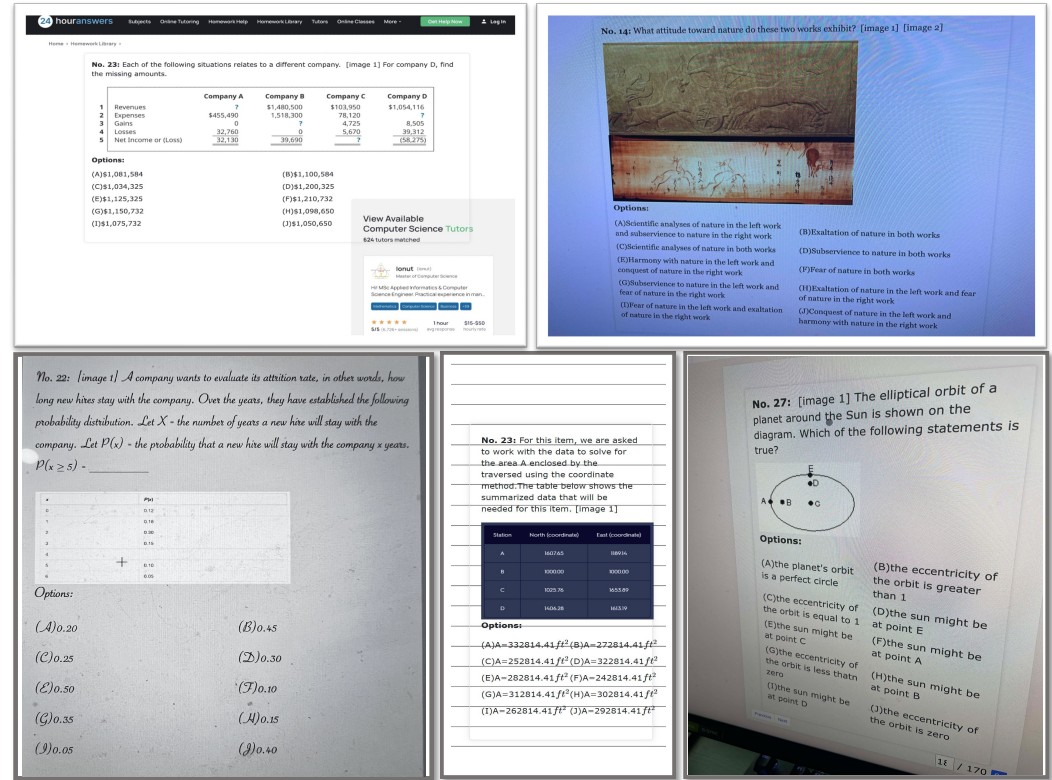

Figure 3: Sample questions from MMMU-Pro Vision. The model is required to answer a multiple-choice question with up to 10 options, each embedded within a screenshot or photo. The images were manually captured by annotators in diverse display environments to reflect real-world cases.

carefully curated multimodal questions from college exams, quizzes, and textbooks, covering six core disciplines across 30 subjects and 183 subfields. Each question in MMMU is a multimodal image-text pair with 4 multiple-choice options, featuring 30 diverse image types such as charts, diagrams, maps, and chemical structures. MMMU quickly becomes a standard evaluation tool in the field, used to assess the capabilities of many prominent multimodal models upon their release (OpenAI, 2024a;b; Anthropic, 2024; Reid et al., 2024; Li et al., 2024a).

However, we find that text-only LLMs can accurately answer some questions without requiring any visual input. We take a closer look at these questions and identify two main issues: 1) **Text-Only Dependency:** Certain questions are relatively independent or irrelevant to the corresponding images. 2) **Shortcut Exploitation:** Even when questions require images for humans to answer correctly, models often find shortcuts or correlations within the candidate options, leveraging their pre-existing knowledge (from pre-training) to arrive at the correct answer. Two examples that are answered correctly by Llama-3-70B Instruct (Dubey et al., 2024) are shown in Figure 2.

## 2.2 METHODS

To address these issues and build a more robust benchmark, we implemented a three-step approach.

**Filtering Questions:** We begin by filtering out questions that can be answered by text-only LLMs. We select four strong open-source LLMs: Llama3-70B-Instruct (Dubey et al., 2024), Qwen2-72B-Instruct (Yang et al., 2024), Yi-1.5-34B-Chat (Young et al., 2024), and Mixtral-8×22B-Instruct (gpt-4o)—and task them with answering the MMMU questions without access to images. The models are required to provide answers even when they indicate that visual input is necessary. We repeat this process ten times for each model, considering a question as "answerable" if a model correctly answers it more than five times. We then exclude any question where at least three out of the four models answer correctly across the majority of trials. We randomly sample 1800 questions from the remaining pool, evenly distributed across 30 subjects (60 questions per subject).

**Augmenting Candidate Options:** Despite the filtering, some questions can still be answered by text-only LLMs, often exploiting subtle hints within the candidate options. To counteract this, we increase the number of candidate options from four to ten, making it more challenging for models to rely on guessing. This augmentation is done by human experts with the assistance of GPT-4o, with additional validation steps to ensure the quality and diversity of the options. Specifically, GPT-4o generates and Claude 3.5 filters the options, followed by two rounds of human review to refine and verify the augmented options. This augmentation is done by human experts with the assistance of GPT-4o. During this process, experts also review the original annotated questions to ensure their relevance to the images and to eliminate any questions that lack a clear connection or coherence. This step filters out 70 questions, and we obtain 1730 questions in total.

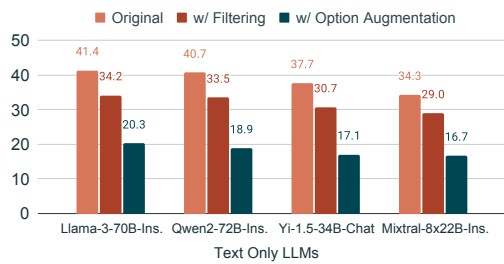

Figure 4: Accuracy of text-only LLMs in different sets of MMMU questions.

As illustrated in Figure 4, these two steps significantly reduce the accuracy of text-only models attempting to guess the answers.

**Enhancing Evaluation with a Vision-Only Setting:** To further challenge the multimodal understanding of models, we introduce a vision-only input setting in MMMU-Pro. In this setting, the model is presented with a question embedded within a screenshot or photo, without any text explicitly fed into the model. To implement this setting, we ask the human annotators to manually capture photos and screenshots over a simulated display environment. This process involves varying the backgrounds, font styles, and font sizes to replicate the diversity of real-world conditions. By using different combinations of these elements, we create a broad range of visual contexts, ensuring that the models are not only challenged by the integration of text and images but also by the variability in how this content is presented. Examples of the vision-only input setting are shown in Figure 3.

The motivation for introducing this setting stems from real-world usage patterns and the fundamental cognitive abilities of humans. In everyday scenarios, users often take shortcuts by simply capturing screenshots of questions that include both text and images, rather than inputting text separately. This habit of feeding integrated visual-textual content reflects a natural human tendency to process information holistically. Humans excel at interpreting and understanding information when both text and images are presented together, and our goal with this setting is to push models toward achieving a similar level of comprehension. By mimicking this real-world behavior, the vision-only input setting not only adds realism to the benchmark but also ensures that models are better equipped to handle the complexities of multimodal tasks as they appear in practical applications.

After these three steps, we finally obtain a total of 3460 questions (1730 samples are in the standard format, and the other 1730 are in screenshot or photo form).

## 3 EXPERIMENTS

### 3.1 EXPERIMENTAL SETUPS

**Baselines.** To establish a comprehensive understanding of MMMU-Pro's difficulty and to provide reference points for future research, we evaluate a diverse set of state-of-the-art multimodal models as baselines. These models represent a range of training approaches and capabilities in the field of multimodal AI. Our baseline models include:

*Proprietary Models:* GPT-4o (0513) (OpenAI, 2024a) and GPT-4o mini (OpenAI, 2024b), Claude 3.5 Sonnet (Anthropic, 2024), and Gemini 1.5 Pro (0801 and 0523 versions) (Team et al., 2023; Reid et al., 2024). These models represent the cutting edge of multimodal AI capabilities.

*Open-source models:* We evaluate a range of open-source models, including InternVL2 (8B, 40B, and Llama3-76B versions) (Chen et al., 2024), LLaVA (OneVision-7B, OneVision-72B, and various NeXT versions) (Li et al., 2024a; Liu et al., 2024a), VILA-1.5-40B (Lin et al., 2024), MiniCPM-

| | MMMU-Pro | | | MMMU (Val) | $\Delta_1$ | $\Delta_2$ |
|---|---|---|---|---|---|---|
| | Standard (4 Opts) | Standard (10 Opts) | Vision | | | |
| Random Choice | 24.9 | 12.8 | 12.4 | 22.1 | -9.3 | -9.7 |
| Frequent Choice | 27.8 | 12.1 | 12.1 | 26.8 | -14.7 | -14.7 |
| Human Expert (Low) | 75.4 | 73.0 | 73.0 | 76.2 | -3.2 | -3.2 |
| Human Expert (Medium) | 82.1 | 80.8 | 80.8 | 82.6 | -1.8 | -1.8 |
| Human Expert (High) | 88.6 | 85.4 | 85.4 | 88.6 | -3.2 | -3.2 |
| GPT-4o (0513) (OpenAI, 2024a) | **64.7** | 54.0 | **49.7** | **69.1** | -15.1 (↑ 1) | **-19.4** ( - ) |
| Claude 3.5 Sonnet (Anthropic, 2024) | 63.7 | **55.0** | 48.0 | 68.3 | **-13.3** (↓ 1) | -20.3 ( - ) |
| Gemini 1.5 Pro (0801) (Reid et al., 2024) | 60.6 | 49.4 | 44.4 | 65.8 | -16.4 ( - ) | -21.4 ( - ) |
| Gemini 1.5 Pro (0523) (Reid et al., 2024) | 57.6 | 46.5 | 40.5 | 62.2 | -15.7 ( - ) | -21.7 ( - ) |
| GPT-4o mini (OpenAI, 2024b) | 55.3 | 39.9 | 35.2 | 59.4 | -19.5 (↑ 1) | -24.2 (↑ 1) |
| Qwen2-VL-72B (Qwen, 2024) | **59.3** | **49.2** | **43.3** | **64.5** | **-15.3** ( - ) | -21.2 ( - ) |
| InternVL2-Llama3-76B (Chen et al., 2024) | 55.0 | 41.9 | 38.0 | 58.3 | -16.4 (↓ 1) | **-20.3** (↓ 1) |
| InternVL2-40B (Chen et al., 2024) | 47.4 | 36.3 | 32.1 | 55.2 | -18.9 ( - ) | -23.1 (↓ 1) |
| LLaVA-OneVision-72B (Li et al., 2024a) | 52.3 | 38.0 | 24.0 | 56.8 | -18.8 ( - ) | -32.8 (↑ 5) |
| Qwen2-VL-7B (Qwen, 2024) | 46.6 | 34.1 | 27.0 | 54.1 | -20.0 (↑ 1) | -27.1 (↓ 1) |
| Pixtral-12B (Mistral, 2024) | 47.5 | 33.4 | 25.0 | 52.5 | -19.1 (↑ 1) | -27.5 ( - ) |
| InternVL2-8B (Chen et al., 2024) | 42.6 | 32.5 | 25.4 | 51.2 | -18.7 ( - ) | -25.8 (↓ 3) |
| MiniCPM-V2.6 (Yao et al., 2024) | 40.6 | 30.2 | 24.2 | 49.8 | -19.6 (↑ 1) | -25.6 (↓ 3) |
| VILA-1.5-40B (Lin et al., 2024) | 46.8 | 35.9 | 14.1 | 51.9 | -16.0 (↓ 2) | -37.8 (↑ 9) |
| LLaVA-NEXT-72B (Liu et al., 2024a) | 43.0 | 31.0 | 19.2 | 49.9 | -18.9 ( - ) | -30.7 ( - ) |
| LLaVA-OneVision-7B (Li et al., 2024a) | 42.8 | 29.5 | 18.7 | 48.8 | -19.3 (↑ 2) | -30.1 (↓ 1) |
| LLaVA-NeXT-34B (Liu et al., 2024a) | 44.5 | 30.3 | 17.2 | 48.1 | -17.8 (↓ 2) | -30.9 (↓ 1) |
| Idefics3-8B-Llama3 (Laurençon et al., 2024) | 40.8 | 30.1 | 15.6 | 46.6 | -16.5 (↓ 1) | -31.0 ( - ) |
| Qwen2-VL-2B (Qwen, 2024) | 34.8 | 25.3 | 17.2 | 41.1 | -15.8 ( - ) | -23.9 (↓ 3) |
| Phi-3.5-Vision (Abdin et al., 2024) | 37.8 | 26.3 | 13.1 | 43.0 | -16.7 ( - ) | -29.9 (↑ 3) |
| LLaVA-NeXT-7B (Liu et al., 2024a) | 33.7 | 19.4 | 14.6 | 35.3 | -15.9 ( - ) | -20.7 (↓ 3) |
| LLaVA-NeXT-13B (Liu et al., 2024a) | 33.9 | 19.8 | 14.5 | 36.2 | -16.4 ( - ) | -21.7 (↓ 1) |

Table 1: Results of models on MMMU-Pro and MMMU (Val). $\Delta_1$: Standard (10 options) - MMMU (Val); $\Delta_2$: Vision - MMMU (Val). (↓) represents a decrease in ranking, while (↑) indicates an increase. The best-performing model in each category is **in-bold**, and the second best is underlined.

V2.6 (Yao et al., 2024), Phi-3.5-Vision (Abdin et al., 2024), and Idefics3-8B-Llama3 (Laurençon et al., 2024). These models showcase the current state of publicly available multimodal AI systems. We evaluate these models across three different settings: 1) Standard setting without augmented options (usually 4 options); 2) Standard setting with augmented options (usually 10 options); 3)Vision-only input setting.

The overall performance score for MMMU-Pro is calculated as the average of scores from settings (2) and (3). We include setting (1) and report the original MMMU validation set performance solely for comparison purposes, to highlight the increased difficulty of MMMU-Pro.

We evaluate the models with both *Direct* and *CoT* prompts (as shown in Appendix B), and report the higher ones in the overall results. We also discuss the influence of the CoT prompt in subsection 3.3.

**Approximating Human Expert Performance.** While rigorous human evaluation of MMMU-Pro provides valuable insights, conducting such an assessment is both time-consuming and costly. Instead, we develop an approach to approximate human expert performance based on the original MMMU human evaluation data.[2] This approximation is justified by several key factors. Firstly, the core content and difficulty of the questions remain unchanged in MMMU-Pro, supporting the validity of using the original human performance data as a close approximation. Secondly, in the original MMMU evaluation, human experts are required to write out their problem-solving processes, significantly reducing the likelihood of random guessing. For questions without detailed solving processes, we randomly select one option from the augmented candidates and recalculate the accuracy. Finally, human experts, with their innate ability to seamlessly integrate visual and textual information, are expected to perform similarly in the vision-only input setting as they do in the original format. Based on these considerations, we posit that human expert performance on MMMU-Pro closely aligns with the original MMMU results, allowing us to maintain a human per-

---
[2]We contacted the MMMU authors and obtained their original human evaluation raw data.

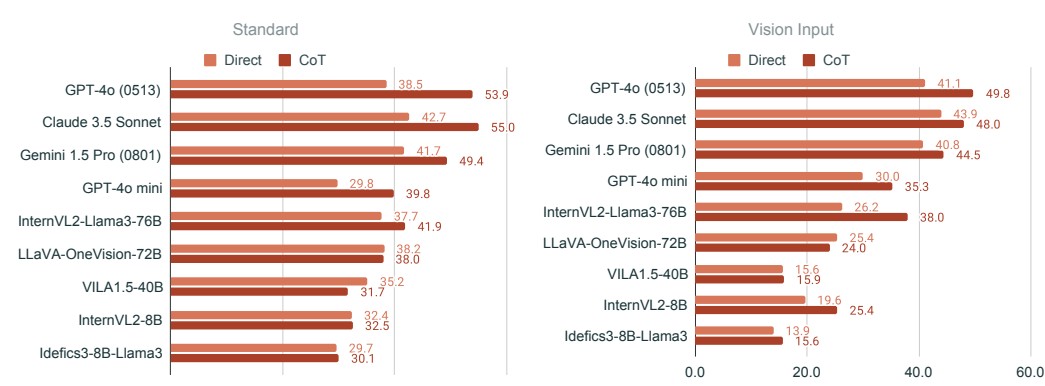

Figure 5: Impact of CoT prompting of different models in the two settings of MMMU-Pro.

formance benchmark without incurring the substantial costs of a new expert evaluation. More details of the human estimation performance can be found in Appendix C.

## 3.2 OVERALL RESULTS

We presented the overall results of MMMU-Pro of different models in Table 1.

**Effect of Increased Candidate Options:** The shift from 4 to 10 candidate options ($\Delta_1$) reveals a significant drop in performance for all models. GPT-4o (0513) experienced a decrease of 10.7%, from 64.7% to 54.0%. This indicates that increasing the number of options effectively reduces the likelihood of models guessing the correct answer, forcing them to engage more deeply with the multimodal content.

**Impact of Vision-Only Setting:** The introduction of the vision-only input setting further challenges models, as evidenced by the additional drop in performance when comparing the vision-only results to the 10-option standard ($\Delta_2$). For instance, GPT-4o (0513) dropped another 4.3% in accuracy when evaluated in the vision-only setting, and LLaVA-OneVision-72B saw a dramatic 14.0% decrease. This suggests that the vision-only setting successfully tests the models' ability to integrate visual and textual information, highlighting their limitations when the text is not explicitly provided.

**Combined Effects on MMMU-Pro:** The overall $\Delta_3$, representing the difference between MMMU-Pro and MMMU (Val), shows a significant decrease across the board. For instance, models like Gemini 1.5 Pro (0801) and Claude 3.5 Sonnet exhibited declines of 18.9% and 16.8%, respectively, while more drastic drops were seen in models like VILA-1.5-40B with a 26.9% decrease.

This significant reduction in accuracy across the board suggests that MMMU-Pro successfully mitigates the shortcuts and guessing strategies that models could exploit in the original benchmark.

## 3.3 DOES COT HELP IN ANSWERING MMMU-PRO QUESTIONS?

Figure 5 examines the effectiveness of Chain of Thought (CoT) prompting in enhancing model performance on the MMMU-Pro benchmark, both in the Standard and Vision Input settings. Across both settings, the introduction of CoT prompts generally led to improved performance. However, the extent of improvement varied significantly among models. For instance, Claude 3.5 Sonnet demonstrated a substantial increase in the Standard setting, improving from 42.7% to 55.0%. In contrast, models like LLaVA-OneVision-72B showed only minimal improvement.

Interestingly, we observed a significant performance drop for some models, such as VILA1.5-40B. This decline might be attributed to challenges in instruction-following abilities. When a model struggles to follow instructions accurately, generating CoT explanations becomes more difficult. Additionally, these models may face issues with maintaining the correct response format, leading to what is known as "boiled response format" problems. These findings highlight the potential of CoT to enhance model performance in complex, real-world tasks that require nuanced reasoning

| Discipline | LLaVA-OneVision-72B | | | GPT4o | | |
|---|---|---|---|---|---|---|
| | CoT Acc | Direct Acc | Difference | CoT Acc | DIRECT Acc | Difference |
| **Art and Design** | 20.42% | 37.53% | -17.12% | 63.14% | 61.55% | 1.58% |
| **Science** | 23.89% | 22.61% | 1.28% | 46.67% | 38.46% | 8.22% |
| **Business** | 29.26% | 24.50% | 4.76% | 57.45% | 42.79% | 14.66% |
| **Humanities and Social Science** | 32.14% | 36.60% | -4.46% | 60.08% | 57.87% | 2.21% |
| **Health and Medicine** | 19.22% | 20.78% | -1.56% | 49.68% | 44.34% | 5.34% |
| **Tech and Engineering** | 22.98% | 20.65% | 2.33% | 37.72% | 23.23% | 14.49% |

Table 2: Comparison of CoT and direct accuracy of two representative models across disciplines in the Vision Input setting. Difference = CoT Acc - Direct Acc.

and integration of multiple information sources. However, they also underscore the importance of robust instruction-following capabilities as a prerequisite for effective CoT implementation.

The effectiveness of CoT prompting across disciplines is summarized in Table 2 and Figure 10, comparing CoT and direct accuracy for GPT-4o and LLaVA-OneVision 72B. CoT shows significant improvements in reasoning-intensive fields like *Tech and Engineering* (e.g., a 14.49% gain for GPT-4o) and *Science* (8.22% gain). Smaller yet consistent gains are observed for LLaVA-OneVision 72B, such as 2.33% in *Tech and Engineering*. However, CoT's benefits are limited or negative in fields like *Art and Design*, where GPT-4o gains only 1.58%, and LLaVA-OneVision 72B sees a 17.12% decline. These results underscore CoT's strengths in structured reasoning tasks but its reduced effectiveness in domains requiring subjective interpretation.

### 3.4 DOES OCR HELP IN THE VISION SETTING?

In the Vision Input setting, one natural question is whether Optical Character Recognition (OCR) helps improve model performance on MMMU-Pro. We answer this question by first calculating the OCR accuracy of different models. Specifically, we ask the model to extract the full text of the question and answer choices. Then the OCR accuracy is calculated by comparing the text extracted with the original text using Levenshtein distance, which measures the difference between the two strings. The similarity between the extracted and original text is computed as: OCR Accuracy $= 1 - \text{Levenshtein.distance}(\text{text1}, \text{text2}) / \max(\text{len}(\text{text1}), \text{len}(\text{text2}))$.

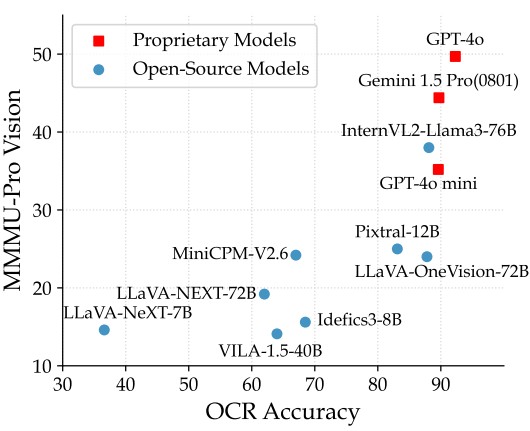

Figure 6: Correlation between OCR accuracy and MMMU-Pro Vision performance.

| Model | OCR Acc. | Vision Setting Acc. | |
|---|---|---|---|
| | | w/ OCR Prompt | w/o OCR Prompt |
| GPT-4o | 92.3 | **49.7** | 49.4 |
| Gemini 1.5 Pro(0801) | 89.7 | **44.4** | 43.6 |
| GPT-4o mini | 89.6 | 35.2 | **35.6** |
| InternVL2-Llama3-76B | 88.1 | **38.0** | 37.9 |
| InternVL2-Llama3-40B | 85.5 | **32.1** | 28.9 |
| Pixtral-12B | 83.1 | **25.0** | 24.1 |
| LLaVA-OneVision-72B | 87.8 | **24.0** | 23.8 |
| InternVL2-8B | 85.2 | **25.4** | 24.6 |
| MiniCPM-V2.6 | 67.0 | **24.2** | 21.1 |
| LLaVA-NEXT-72B | 62.0 | 19.2 | **20.0** |
| Idefics3-8B-Llama3 | 68.5 | **15.6** | 14.1 |
| LLaVA-NeXT-7B | 36.6 | **14.6** | 14.3 |
| LLaVA-NeXT-13B | 51.1 | **14.5** | 12.8 |

Table 3: Model performance in the Vision Input setting, comparing OCR accuracy with/without OCR prompts.

Table 3 shows that although most of the models demonstrate strong OCR capabilities, as indicated by high similarity scores. Based on the result, we then explore whether explicitly asking the model to first extract the question and then solve it (with an OCR prompt shown in Appendix B) could help in improving performance within the Vision Input setting of MMMU-Pro. Across the models evaluated, the inclusion of OCR prompts did not significantly alter performance. These minimal differences suggest that strong capable models are already proficient at extracting and understanding textual information from images, even without explicit OCR prompts.

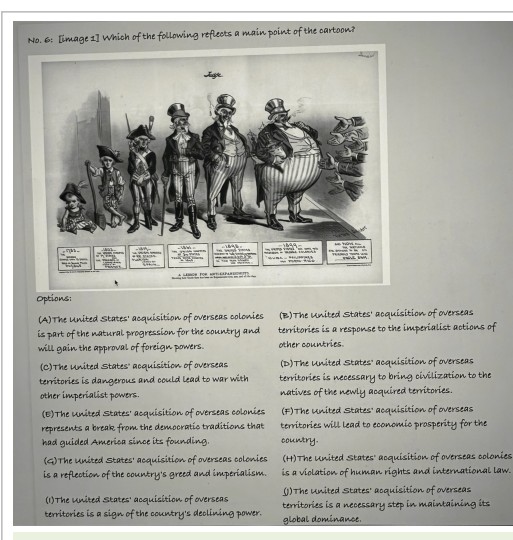

GPT-4o (Standard Setting): To determine the main point of the cartoon, let's analyze the visual and textual elements:

The cartoon by Judge depicts the progression of the United States with figures representing different years: 1783, 1803, 1819, 1861, 1898, and 1899. In each figure, the United States is shown acquiring more territories:

- **1783**: The United States starts with the 13 states.
- **1803**: The Louisiana Purchase is acquired from France.
- **1819**: Additional territory is acquired from Spain.
- **1861**: New states are admitted, expanding the Union.
- **1898**: The United States gains control over territories from the Spanish-American War.
- **1899**: The United States comes into possession of various new colonies such as Cuba and the Philippines.

The final image shows the United States (personified as "Uncle Sam") receiving approval and open arms from other nations, which suggests the U.S. is being welcomed by the international community.

Given this depiction, the cartoon seems to be promoting the idea that the expansion of the United States was a natural progression that leads to international approval and friendly relations with other nations. This aligns most closely with option A. **Answer: A**

GPT-4o (Vision Setting): The multiple-choice question in the image is: No. 6: Image 1 - Which of the following reflects a main point of the cartoon? Options: (A) The United States' acquisition of overseas colonies is part of the natural progression for the country and will gain the approval of foreign powers. (B) The United States' acquisition of overseas territories is a response to the imperialist actions of other countries. (C) The United States' acquisition of overseas territories is dangerous and could lead to war with other imperialist powers. (D) The United States' acquisition of overseas territories is necessary to bring civilization to the natives of the newly acquired territories. (E) The United States' acquisition of overseas colonies represents a break from the democratic traditions that had guided America since its founding. (F) The United States' acquisition of overseas territories will lead to economic prosperity for the country. (G) The United States' acquisition of overseas colonies is a reflection of the country's greed and imperialism. (H) The United States' acquisition of overseas colonies is a violation of human rights and international law. (I) The United States' acquisition of overseas territories is a sign of the country's declining power. (J) The United States' acquisition of overseas territories is a necessary step in maintaining its global dominance.

To determine the main point of the cartoon, we need to analyze the visual content and any text present in the cartoon. The cartoon appears to depict a negative view of U.S. expansionism, comparing the U.S. to European imperial powers who are often associated with greed and exploitation. Therefore, the main point likely reflects a critical stance on the U.S.'s actions. **Answer: (G)**

Analysis: In the vision-only input scenario, the model accurately extracts text from the photo. However, its response tends to be more basic and lacks in-depth analysis. The integration of both visual and textual information appears to increase the cognitive load on the vision module, which may result in a higher likelihood of errors.
*[Note: Portions of the original model's response have been omitted for brevity.]*

Figure 7: Comparison of GPT-4o's responses between Standard and Vision Input settings.

Interestingly, the Figure 6 reveals that models with high OCR accuracy do not always achieve strong multimodal reasoning scores. For instance, LLaVA-OneVision-72B demonstrates an OCR accuracy comparable to InternVL2-Llama3-76B and GPT-4o mini, yet its MMMU-Pro Vision performance is significantly lower. This disparity suggests that high OCR accuracy alone is insufficient to ensure strong multimodal reasoning capabilities. Conversely, models excelling in multimodal reasoning, such as GPT-4o, consistently exhibit strong OCR performance.

Furthermore, while GPT-4o achieves an impressively high OCR accuracy, its performance on MMMU-Pro Vision still suffers a notable drop compared to its results on MMMU (Val) benchmark. This highlights that even for state-of-the-art models with robust OCR capabilities, the challenges posed by the vision-only input setting in MMMU-Pro reveal limitations in their ability to integrate and reason over multimodal inputs effectively.

### 3.5 QUALITATIVE ANALYSIS

To gain deeper insights into model performance beyond quantitative metrics, we conducted a thorough qualitative analysis of MMMU-Pro results, focusing on two key scenarios: 1) Correct answers with four options but failure with ten options in the standard setting; 2) Success in the standard ten-option setting but failure in the vision input setting. Our analysis revealed several critical factors affecting model performance:

**Challenges with Increased Options.** Models often select the closest answer rather than arriving at a definitive choice, leading to increased errors with more options, as shown in Figure 11. Conceptually similar options, particularly in nuanced questions, can cause confusion. For instance,

in conceptual questions, models struggled to differentiate subtle distinctions within a subject area, revealing limitations in fine-grained understanding.

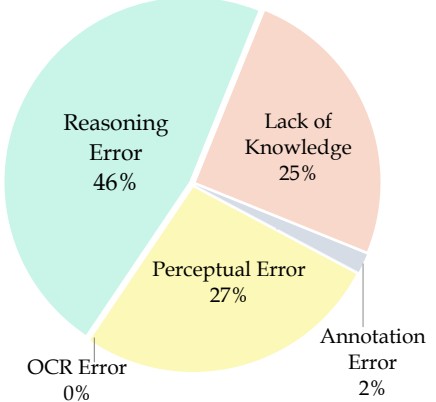

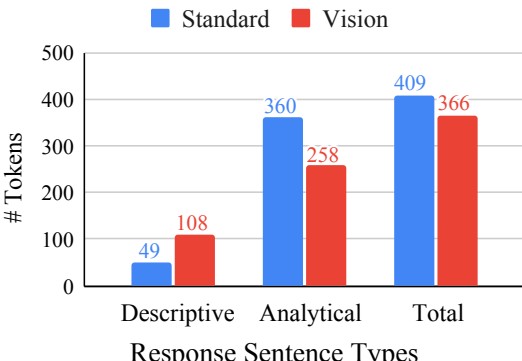

Figure 8: Error distribution of 60 annotated GPT-4o errors.

Figure 9: GPT-4o outputs' length comparison between the Standard and Vision settings.

**Increased Cognitive Load in Vision-Text Integration.** Processing visual and textual inputs simultaneously increases the cognitive load on models. An example is shown in Figure 7. The model perfectly extracted the text from the image but still failed to answer the question correctly. Another case is shown in Figure 21. The graph's similar lines and overlapping data points may distract the model from distinguishing between the two unemployment categories, leading to the error.

**Overemphasis on Visual Cues in Multimodal Reasoning.** When visual cues dominate over textual reasoning, models may incorrectly prioritize less relevant information from the images. In the Figure 33 example, the Vision Setting incorrectly chose the League of Nations by focusing on the World War I image, missing the broader context of World War II and the United Nations. A proper balance between visual and textual information is essential to avoid such mistakes.

**Impact of Context Switching.** Rapid transitions between visual and textual information can cause models to lose focus or misinterpret key data. For example, in Figure 26, the model initially correctly defined both the objective function and the algebraic constraints. However, due to context switching between the textual description and the geometric figure, it misinterpreted the feasible region.

These cases highlight MMMU-Pro's effectiveness in exposing the limitations of current multimodal models, particularly in handling increased complexity and integrating diverse information types.

### 3.6 ERROR ANALYSIS

Following the MMMU error analysis, we analyze 60 error cases from GPT-4o in the Vision setting to better understand the error reasons (Figure 8). Consistent with MMMU findings, the errors are broadly categorized into three main types: perception errors, knowledge errors, and reasoning errors. However, reasoning errors account for 46% of cases, a significant increase from the original MMMU distribution (26%). Within perception errors, text recognition and OCR do not prove to be the primary bottleneck. Instead, the main challenges lie in the integration and interpretation of visual and textual information. This shift in error distribution highlights the increased difficulty for models in transitioning from accurate perception to complex multimodal reasoning.

### 3.7 RESPONSE LENGTH COMPARISON BETWEEN THE SETTINGS

One interesting observation we have from the previous qualitative examples is that responses (especially the reasoning sentences) of GPT-4o under the Vision Input setting seem to be shorter than the Standard setting. We quantify this phenomenon by asking another LLM (Qwen2-72B-Instruct (Yang et al., 2024)) to classify the GPT-4o's responses into "Descriptive" sentences and "Analytical" sentences. As shown in Figure 9, GPT-4o generates significantly shorter responses but uses more tokens for "Descriptive" rather than "Analytical". One possible reason is that the increased cognition workload of the vision inputs requires the model to focus more on visual processing, which distracts the model from generating extensive reasoning chains.

## 4 GUIDE FOR FUTURE MODEL TRAINING

The results of MMMU-Pro provide valuable insights into the challenges faced by current multimodal models and suggest several promising directions for future model development.

**Scaling of LLM Backbones.** As demonstrated in Table 1, increasing the scale of large language model (LLM) backbones consistently enhances both perception and reasoning capabilities. For example, larger models such as GPT-4o outperform their smaller counterparts like GPT-4o mini, while LlavaOneVision-72B achieves better results than LlavaOneVision-7B. Similarly, InternVL2-78B demonstrates superior performance compared to InternVL2-8B. This trend underscores the importance of scaling as a critical factor in improving multimodal understanding and reasoning.

**More Capable Vision Encoders that Highlights Visual Representation Learning.** We train two Cambrian Tong et al. (2024a) models on 1M Cambrian data with two different vision encoders to explore their impact (more details of the setup are in Appendix F). As shown in Table 4, encoders such as Siglip ViT-SO400M-14 (Zhai et al., 2023), trained with extensive language supervision, perform well on MMMU (Val) but struggle on

| Method | MMMU (Val) | MMMU-Pro (Vision) |
|---|---|---|
| DINOv2 ViT-G-14 | 37.1 | 17.4 |
| Siglip ViT-SO400M-14 | 37.9 | 16.7 |

Table 4: Performance of an MLLM with different vision encoders on MMMU and MMMU-Pro.

MMMU-Pro (Vision). In comparison, self-supervised encoders like DINOv2 ViT-G-14 (Oquab et al., 2023) achieve better results on the Vision input setting. These findings suggest future work may focus on further enhancing visual feature learning while exploring the integration of language-based training objectives with self-supervised training objectives.

**Better Integration of Vision and Text Modalities.** The integration of visual and textual information remains a key challenge for multimodal models. Current architectures often struggle with tasks requiring deep cross-modal understanding. Developing models with better cross-modal attention and effective feature fusion is critical for bridging this gap.

**CoT Data Generation.** The CoT prompting technique shows significant benefits in reasoning-heavy domains within MMMU-Pro, as reflected in Figure 5 and Table 2. While domains like *Tech and Engineering* and *Business* see notable improvements, CoT performance remains weak or even detrimental in areas such as *Art and Design*. To address these gaps, future efforts focus on synthesizing more diverse reasoning-intensive CoT data and tailoring strategies for domains where CoT impact is minimal. Leveraging inference-compute concepts (Welleck et al., 2024) further enhances CoT capabilities, enabling models to generalize more effectively across varied reasoning tasks.

**Text-Rich Image Generation in Reasoning Scenarios.** Our analysis shows that strong OCR accuracy and reasoning performance on traditional benchmarks do not always translate to success on MMMU-Pro Vision. A potential reason is the lack of training data with text-rich images in reasoning-intensive contexts. To address this, we developed a tool leveraging the MMMU-Pro Vision human annotation process. This tool processes a JSON file with questions and images and outputs screenshots embedding both. Such tools can further generate similar datasets at scale, enhancing models' ability to integrate visual and textual information in real-world scenarios.

By focusing on these directions, future modeling efforts can address limitations highlighted by MMMU-Pro and push multimodal understanding and reasoning boundaries.

## 5 CONCLUSION

MMMU-Pro presents a more robust multimodal understanding and reasoning benchmark compared with its predecessor MMMU. Our results demonstrate MMMU-Pro's effectiveness in exposing the limitations of current state-of-the-art multimodal models, with significant performance drops across all tested systems. MMMU-Pro opens up several important avenues for future research: 1) Developing models with consistent performance across all MMMU-Pro settings, particularly in bridging the gap between standard and vision-only inputs. 2) Improving vision-text integration capabilities to handle complex, mixed-format inputs more effectively. 3) Exploring advanced reasoning techniques to address the increased complexity of MMMU-Pro questions.

## ETHICAL STATEMENT

The MMMU-Pro benchmark is designed with ethical considerations to ensure fair and responsible AI evaluation. The dataset excludes sensitive content, and the assessment focuses on testing multi-modal capabilities without introducing bias. We aim for transparency in reporting model limitations and encourage further research to address any societal impacts related to the use of these models in real-world applications.

## REPRODUCIBILITY STATEMENT

To promote transparency and reproducibility, all code, data, and experimental results for MMMU-Pro are made publicly available through an anonymous GitHub repository (link provided in the paper). The detailed methodology for constructing the benchmark, including the filtering process, option augmentation, and vision-only input setting, is clearly documented, enabling other researchers to replicate or extend the benchmark.

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

# Table of Contents in Appendix

## A    RELATED WORK

**Multimodal Large Language Models.** Recent progress in multimodal AI has been marked by innovative training approaches (Lu et al., 2019; Chen et al., 2020; Zhou et al., 2020; Zhang et al., 2021; Li et al., 2020; Alayrac et al., 2022; Awadalla et al., 2023). Inspired by the success of large language models, researchers have developed various models with improved instruction-following capabilities (Liu et al., 2023c;b; 2024a; Li et al., 2024a; Dai et al., 2023; Zhu et al., 2023; Zhang et al., 2023; Gao et al., 2023; Ye et al., 2023a;b; Zhao et al., 2023; Li et al., 2023; Monajatipoor et al., 2023; Zhao et al., 2024; Li et al., 2024c; Lin et al., 2024; Zhang et al., 2024a). Proprietary models such as GPT-4V (OpenAI, 2023), GPT-4o (OpenAI, 2024a), Gemini (Team et al., 2023), and Claude-3.5 (Anthropic, 2024) have demonstrated strong performance across various vision-language tasks. However, a significant challenge remains in accurately evaluating the capabilities of these advanced LMMs, highlighting the need for more robust and comprehensive benchmarks.

**MLLM Benchmarks.** The rise of more advanced multimodal pre-training and instruction tuning has exposed the limitations of earlier benchmarks like VQA (Antol et al., 2015; Goyal et al., 2017), OK-VQA (Marino et al., 2019), and MSCOCO (Lin et al., 2014), which no longer suffice to evaluate the full spectrum of LMMs capabilities. To address this, recent benchmarks such as LAMM (Yin et al., 2023b), LVLM-eHub (Xu et al., 2023), SEED (Li et al., 2024b), MMBench (Liu et al., 2023d),CV-Bench (Tong et al., 2024a), MM-Vet (Yu et al., 2024), Mantis (Jiang et al., 2024), and BLINK (Fu et al., 2024) have emerged, covering aspects from basic perception to hallucination detection (Cui et al., 2023; Liu et al., 2023a). However, existing benchmarks often fall short in evaluating expert-level domain knowledge and complex reasoning (Lu et al., 2023a; Zhang et al., 2024b). While MMMU (Yue et al., 2024) made strides by incorporating multimodal, college-level questions, it still permits text-only models to find shortcuts (Lu et al., 2023b; Zhang et al., 2024b). To address these limitations, we introduce MMMU-Pro, a more robust benchmark that removes text-only answerable questions, expands candidate options, and includes a vision-only input setting to better reflect real-world multimodal scenarios.

# B    EVALUATION PROMPTS

---

**Evaluation Prompts: OCR Prompt**

**OCR Prompt:**
"**Write out the multiple-choice question in the image** and then solve it. The last line of your response should be of the following format: 'Answer: $LETTER' (without quotes) where LETTER is one of the options. Think step by step before answering."

**w/o OCR Prompt:**
"Answer the following multiple-choice question in the image. The last line of your response should be of the following format: 'Answer: $LETTER' (without quotes) where LETTER is one of the options. Think step by step before answering."

---

**Evaluation Prompts: Direct vs CoT**

**Direct:**
"Answer directly with the option letter from the given choices."

**CoT:**
"Answer the following multiple-choice question. The last line of your response should be of the following format: 'Answer: $LETTER' (without quotes) where LETTER is one of the options. Think step by step before answering."

---

**Evaluation Prompts: OCR Task**

**OCR Task Prompt:**
"Extract and output the full text of the question, including any introductory descriptions, as well as the corresponding answer choices from the multiple-choice question in the image. Exclude any text from associated images or the question number. Perform OCR only; do not attempt to solve the question."

---

**Evaluation Prompts: Split Response Task**

**Split Response Task Prompt:**
Your task is to split the given answer into two distinct parts: the part that describes the question and the part that analyzes the answer. This is a splitting task, so ensure you do not omit any content or generate any additional content not present in the input. Follow these guidelines:

1. Description of the Question:
- Extract the portion of the answer that describes the question being addressed.
- Ensure that this part is clear and provides enough context to understand the question.

2. Analysis of the Answer:
- Extract the portion of the answer that provides the analysis or reasoning behind the answer.
- Ensure that this part is detailed and provides a complete explanation or solution.

Please split the following answer into the two parts described above and output them in JSON format:
Answer: $LETTER
{
"description_of_question": "Extracted description of the question",
"analysis_of_answer": "Extracted analysis of the answer"
}

## C   APPROXIMATING HUMAN EXPERT PERFORMANCE

Establishing a reliable benchmark for human performance on MMMU-Pro is crucial to evaluating the true capabilities of multimodal AI models. Conducting new and rigorous human evaluations, however, is both time-consuming and expensive. To address this issue, we developed an approximation method based on the existing human evaluation data from the original MMMU. The resulting estimates are presented in Table 5.

|        | Overall | Art & Design | Business | Science | Health & Medicine | Human & Social Sci. | Tech & Eng. |
|--------|---------|--------------|----------|---------|-------------------|---------------------|-------------|
| Low    | 73.0    | 77.4         | 77.9     | 78.5    | 65.2              | 63.6                | 73.5        |
| Medium | 80.8    | 83.3         | 88.4     | 84.9    | 72.8              | 75.8                | 78.2        |
| High   | 85.4    | 85.7         | 89.5     | 86.0    | 84.8              | 81.8                | 84.4        |

Table 5: Estimated human performance on MMMU-Pro across different disciplines, based on the original MMMU evaluation data. The table presents low, medium, and high performance estimates in terms of overall accuracy and discipline-specific breakdowns.

The validity of using this approximation method relies on several key factors. Firstly, the core content and difficulty of the questions in MMMU-Pro remain unchanged from those in the original MMMU, supporting the use of the original human performance data as a valid proxy. Secondly, in the initial MMMU evaluation, human experts were required to document their problem-solving processes, which significantly reduced the likelihood of random guessing. For questions lacking detailed solution processes, we simulated random selection from expanded candidate options and recalculated the accuracy. Finally, human experts inherently excel at seamlessly integrating visual and textual information, suggesting that their performance in a purely visual input setting would be analogous to their performance in the original format.

Given that the 577 questions in MMMU-Pro are sourced from the MMMU validation set, we extracted the corresponding data from the evaluations of the 90 human experts involved in the original MMMU assessment. We categorized and counted these questions based on whether they included a detailed solution process (**w/ Solution**) or were subjected to guessing due to the lack of a detailed solution process (**w/o Solution**). We then counted the correct and incorrect answers in each category, as summarized in Table 6. Specifically, the categorization is defined in Equation 1:

$$
\begin{aligned}
\text{Num}_{\text{total}} &= \text{Num}_{\text{w/o Solution}} + \text{Num}_{\text{w/ Solution}} \\
&= \text{Num}_{\text{w/o Solution(wrong)}} + \text{Num}_{\text{w/o Solution(correct)}} \\
&\quad + \text{Num}_{\text{w/ Solution(wrong)}} + \text{Num}_{\text{w/ Solution(correct)}}
\end{aligned} \tag{1}
$$

Using these counts, we can estimate the lower bound of human performance on MMMU-Pro with Equation 2:

$$
\text{Num}_{\text{Estimate(correct)}} = \text{Num}_{\text{w/ Solution(correct)}} + \left\lfloor \left( \frac{\text{Num}_{\text{w/o Solution}}}{\text{Num}_{\text{total}}} \right) \times \text{Num}_{\text{w/o Solution}} \right\rfloor \tag{2}
$$

This formula considers the number of correctly solved questions with detailed solution processes and the proportion of correctly guessed questions without detailed solution processes, ensuring a conservative estimate.

In summary, by leveraging the original MMMU human evaluation data and applying our estimation method, we provide a reasonable approximation of human performance on MMMU-Pro. This approach maintains the human performance benchmark without incurring the substantial costs associated with new expert evaluations.

| | Low | | | | Medium | | | | High | | | |
|---|---|---|---|---|---|---|---|---|---|---|---|---|
| | w/o Sol. (w/c) | w/ Sol. (w/c) | Est. (w/c) | Acc | w/o Sol. (w/c) | w/ Sol. (w/c) | Est. (w/c) | Acc | w/o Sol. (w/c) | w/ Sol. (w/c) | Est. (w/c) | Acc |
| **Art & Design** | 4/11 | 11/64 | 19/65 | **77.4** | 5/1 | 8/70 | 14/70 | **83.3** | 4/2 | 6/72 | 12/72 | **85.7** |
| Art | 2/2 | 2/14 | 4/14 | 77.8 | 1/0 | 1/16 | 2/16 | 88.9 | 0/1 | 0/17 | 1/17 | 94.4 |
| Art Theory | 1/2 | 2/18 | 5/18 | 78.3 | 1/1 | 2/19 | 4/19 | 82.6 | 1/1 | 3/18 | 5/18 | 78.3 |
| Design | 1/4 | 4/10 | 5/10 | 66.7 | 1/0 | 2/12 | 3/12 | 80.0 | 1/0 | 1/13 | 2/13 | 86.7 |
| Music | 0/3 | 3/22 | 6/22 | 78.6 | 2/0 | 3/23 | 5/23 | 82.1 | 2/0 | 2/24 | 4/24 | 85.7 |
| **Business** | 4/11 | 11/73 | 21/74 | **77.9** | 4/1 | 6/84 | 11/84 | **88.4** | 2/3 | 5/85 | 10/85 | **89.5** |
| Accounting | 0/3 | 3/19 | 6/19 | 76.0 | 2/0 | 1/22 | 3/22 | 88.0 | 0/2 | 1/22 | 3/22 | 88.0 |
| Economics | 0/4 | 4/13 | 5/13 | 72.2 | 1/0 | 1/16 | 2/16 | 88.9 | 1/0 | 0/17 | 1/17 | 94.4 |
| Finance | 1/2 | 2/15 | 4/15 | 78.9 | 0/0 | 1/18 | 1/18 | 94.7 | 0/0 | 2/17 | 2/17 | 89.5 |
| Manage | 2/2 | 2/8 | 4/9 | 69.2 | 1/1 | 2/9 | 4/9 | 69.2 | 1/1 | 2/9 | 4/9 | 69.2 |
| Marketing | 1/0 | 0/18 | 2/18 | 90.0 | 0/0 | 1/19 | 1/19 | 95.0 | 0/0 | 0/20 | 0/20 | 100.0 |
| **Science** | 3/12 | 12/72 | 20/73 | **78.5** | 3/1 | 10/79 | 14/79 | **84.9** | 3/1 | 9/80 | 13/80 | **86.0** |
| Biology | 0/5 | 5/13 | 7/13 | 65.0 | 2/0 | 5/13 | 7/13 | 65.0 | 1/1 | 5/13 | 7/13 | 65.0 |
| Chemistry | 0/3 | 3/14 | 4/14 | 77.8 | 0/1 | 2/15 | 3/15 | 83.3 | 1/0 | 2/15 | 3/15 | 83.3 |
| Geography | 2/0 | 0/8 | 2/8 | 80.0 | 0/0 | 1/9 | 1/9 | 90.0 | 0/0 | 1/9 | 1/9 | 90.0 |
| Math | 1/4 | 4/14 | 7/14 | 66.7 | 1/0 | 1/19 | 2/19 | 90.5 | 1/0 | 1/19 | 2/19 | 90.5 |
| Physics | 0/0 | 0/23 | 1/23 | 95.8 | 0/0 | 1/23 | 1/23 | 95.8 | 0/0 | 0/24 | 0/24 | 100.0 |
| **Health & Med.** | 3/22 | 22/58 | 32/60 | **65.2** | 9/0 | 17/66 | 25/67 | **72.8** | 5/4 | 6/77 | 14/78 | **84.8** |
| Basic Med. | 2/2 | 2/9 | 4/10 | 71.4 | 1/0 | 2/11 | 3/11 | 78.6 | 1/0 | 1/12 | 2/12 | 85.7 |
| Clinical Med. | 1/6 | 6/8 | 9/9 | 50.0 | 3/0 | 5/10 | 7/11 | 61.1 | 2/1 | 1/14 | 3/15 | 83.3 |
| Diagnostics | 0/6 | 6/14 | 9/14 | 60.9 | 3/0 | 4/16 | 7/16 | 69.6 | 2/1 | 2/18 | 5/18 | 78.3 |
| Pharmacy | 0/3 | 3/13 | 4/13 | 76.5 | 1/0 | 3/13 | 4/13 | 76.5 | 0/1 | 1/15 | 2/15 | 88.2 |
| Public Health | 0/5 | 5/14 | 6/14 | 70.0 | 1/0 | 3/16 | 4/16 | 80.0 | 0/1 | 1/18 | 2/18 | 90.0 |
| **Humani. & Soc.** | 5/14 | 14/40 | 24/42 | **63.6** | 3/5 | 9/49 | 16/50 | **75.8** | 5/3 | 5/53 | 12/54 | **81.8** |
| History | 1/4 | 4/4 | 6/4 | 40.0 | 1/0 | 1/8 | 2/8 | 80.0 | 0/1 | 1/8 | 2/8 | 80.0 |
| Literature | 2/2 | 2/15 | 5/16 | 76.2 | 1/2 | 2/16 | 5/16 | 76.2 | 2/1 | 0/18 | 3/18 | 85.7 |
| Sociology | 0/5 | 5/8 | 7/9 | 56.3 | 1/2 | 4/9 | 6/10 | 62.5 | 2/1 | 2/11 | 4/12 | 75.0 |
| Psychology | 2/3 | 3/13 | 6/13 | 68.4 | 0/1 | 2/16 | 3/16 | 84.2 | 1/0 | 2/16 | 3/16 | 84.2 |
| **Tech & Eng.** | 3/25 | 25/106 | 39/108 | **73.5** | 9/4 | 20/114 | 32/115 | **78.2** | 6/7 | 10/124 | 23/124 | **84.4** |
| Agriculture | 0/6 | 6/10 | 9/10 | 52.6 | 1/2 | 5/11 | 8/11 | 57.9 | 2/1 | 2/14 | 5/14 | 73.7 |
| Archi. Eng. | 2/2 | 2/17 | 5/17 | 77.3 | 1/1 | 2/18 | 4/18 | 81.8 | 1/1 | 0/20 | 2/20 | 90.9 |
| Computer Sci. | 0/0 | 0/17 | 2/17 | 89.5 | 1/0 | 1/17 | 2/17 | 89.5 | 0/1 | 2/16 | 3/16 | 84.2 |
| Electronics | 0/0 | 0/8 | 1/8 | 88.9 | 0/0 | 0/9 | 0/9 | 100.0 | 0/0 | 0/9 | 0/9 | 100.0 |
| Energy Power | 0/4 | 4/20 | 6/20 | 76.9 | 2/0 | 4/20 | 6/20 | 76.9 | 1/1 | 1/23 | 3/23 | 88.5 |
| Materials | 0/3 | 3/22 | 5/22 | 81.5 | 1/1 | 3/22 | 5/22 | 81.5 | 1/1 | 2/23 | 4/23 | 85.2 |
| Mechanical Eng. | 1/10 | 10/12 | 13/12 | 48.0 | 3/0 | 5/17 | 8/17 | 68.0 | 1/2 | 3/19 | 6/19 | 76.0 |
| **Overall** | 22/95 | 95/413 | 156/421 | **73.0** | 33/12 | 70/462 | 111/466 | **80.8** | 25/20 | 41/491 | 84/493 | **85.4** |

Table 6: Detailed breakdown of estimated human performance on MMMU-Pro for low, medium, and high performance levels across various disciplines. Abbreviations: "**w/o Sol.**" (without Solution), "**w/ Sol.**" (with Solution), "**Est.**" (Estimate), and "**w/c**" (number of wrong/correct answers).

# D  ENSURING QUALITY AND DIVERSITY OF EXPANDED OPTIONS

Expanding the number of answer options naturally increases the difficulty of the benchmark, but its effectiveness relies heavily on the quality, diversity, and contextual relevance of these additional options. To ensure this, we implemented a rigorous multi-stage validation process, combining automated and human efforts to produce high-quality results.

**Initial Model-Based Option Augmentation and Filtering.** We began by leveraging large language models (LLMs) to automate the initial generation and filtering of expanded options. Specifically, GPT-4o was used to generate additional options, while Claude 3.5 acted as a preliminary filter to remove options that were contextually irrelevant or logically inconsistent. This step significantly reduced the workload for human reviewers by pre-screening the candidates.

**Two Rounds of Human Review.** To further enhance quality and eliminate potential issues, we conducted two rounds of meticulous human validation:

- **First Round of Review:** Individual reviewers assessed the expanded options for each question. They ensured that the options were diverse, logically distinct, and free from ambiguity. If any flaws were identified, reviewers were instructed to correct the issues or create new options to maintain the integrity of the question.
- **Second Round of Review:** A double-check process followed, involving two additional human experts who cross-validated each question and its options. This iterative step eliminated any residual inconsistencies or errors and provided an additional layer of assurance.

By combining automated methods with multi-stage human validation, we ensured that each expanded option met high standards of quality, robustness, and alignment with the intended challenges of the benchmark. This approach not only addressed potential weaknesses in automated generation but also significantly improved the reliability of the dataset.

# E ANALYSIS OF CoT'S IMPACT

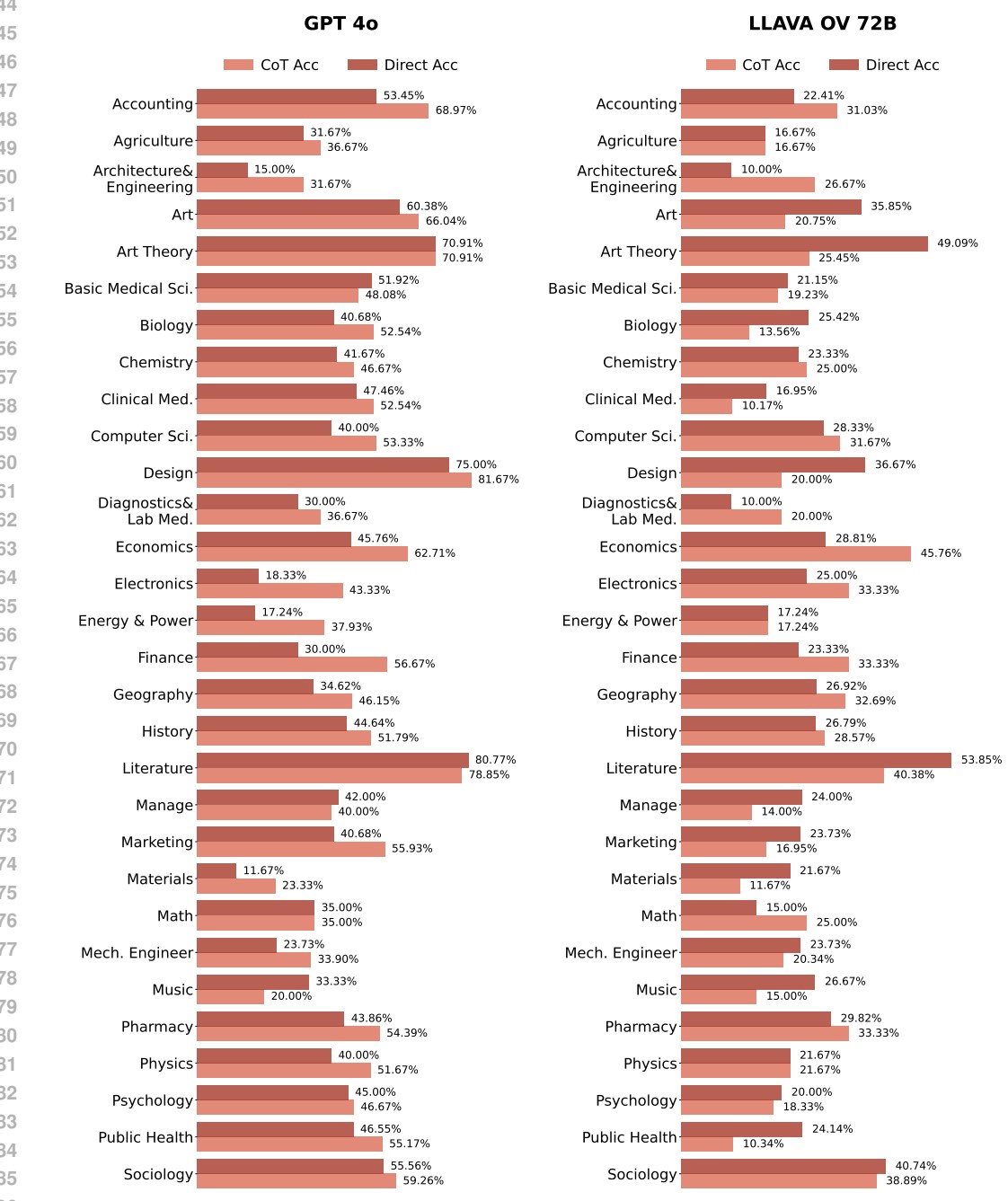

Figure 10: Comparison of CoT and Direct Accuracy across subcategories within major domains for GPT-4o and LLaVA-OneVision 72B.

## F  EXPERIMENTAL SETUP OF VISION ENCODER IMPACT

To evaluate the influence of vision encoders on model performance, we conduct experiments using the open-source architecture Cambrian-1. These experiments fix both the training data (Cambrian-1 1M SFT data) and the large language model (Llama 3.1 8B) to isolate the impact of different vision encoders. Inspired by Cambrian-1: A Fully Open, Vision-Centric Exploration of Multimodal LLMs (Tong et al., 2024a), we follow their methodology by interpolating visual features to a fixed number of tokens (576) and concatenating them along the feature dimension.

## G  COMPARISON WITH AND WITHOUT AUGMENTED OPTIONS

**Question:** A solid copper bar of circular cross section has length $L = 1.25\,m$ and shear modulus of elasticity $G = 45$ GPa. The bar is designed to carry a $250\,N \cdot m$ torque acting at the ends. If the allowable shear stress is $30\,MPa$ and the allowable angle of twist between the ends is 2.5°, what is the minimum required diameter?

**Original Options:**

(A) 35.7 mm    (B) 45.7 mm   (C) 55.7 mm

**Augmented Options:**

(D) 25.7 mm  (E) 40.7 mm (F) 50.7 mm    (G) 30.7 mm
(H) 20.7 mm  (I) 60.7 mm

**GPT-4o (w/o augment options):** To determine the minimum required diameter of the solid copper bar, we need to check two conditions: 1. The maximum shear stress condition. 2. The allowable angle of twist condition. Step 1: Maximum Shear Stress Condition. The torsional shear stress for a solid circular shaft is given by: $\tau = T \cdot r/J$, where $\tau$ is the shear stress, T is the applied torque, $r$ is the radius of the shaft, $J$ is the polar moment of inertia for a solid circular section $(J = \pi r^4/2)$. Rewriting the shear stress formula…Conclusion: Comparing the two diameters, the larger value will govern, as it ensures both conditions are met. Therefore, the minimum required diameter is given by: **Answer: A**

**GPT-4o (w/ augment options):** To determine the minimum required diameter of the solid copper bar, we need to consider both the maximum allowable shear stress and the maximum allowable angle of twist. Step 1: Maximum Allowable Shear Stress. The relationship between the torque T, the shear stress $\tau$, and the polar moment of inertia $J$ for a circular cross-section is given by... Therefore, the minimum required diameter is: $d \approx 55.68$ mm. The closest value greater than or equal to 55.68 mm from the options provided is: **Answer: C**

**Analysis:** Both solutions are incorrect; however, with only three options in the original setup, the model happens to select the correct answer by chance.

Figure 11: Comparison of GPT-4o's responses with and without augmented options.

## H  COMPARISON OF MODEL OUTPUTS ACROSS DIFFERENT INPUT MODES

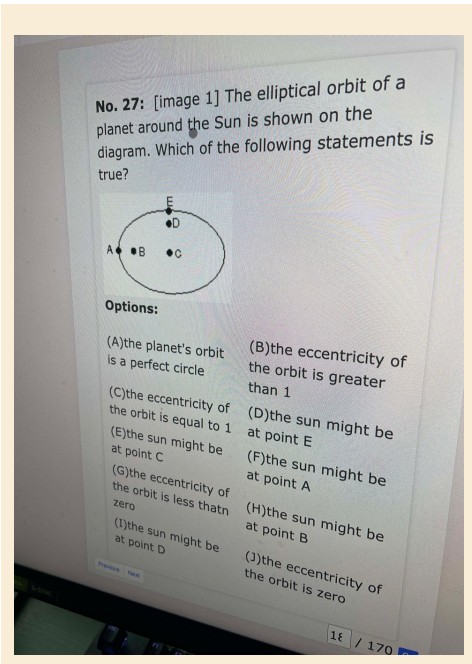

**Ground Truth: (H) the sun might be at point A**

**GPT-4o (Standard Setting):** Let's analyze the options step by step based on the given diagram and the known characteristics of an elliptical orbit: 1. A. the planet's orbit is a perfect circle. Incorrect. An elliptical orbit is not a perfect circle. A perfect circle would have an eccentricity of 0, and the diagram clearly shows an ellipse. 2. B. the eccentricity of the orbit is greater than 1. Incorrect. The eccentricity of an elliptical orbit is always between 0 and 1. An eccentricity greater than 1 would indicate a hyperbolic trajectory. 3. C. the eccentricity of the orbit is equal to 1. Incorrect. An eccentricity of 1 would indicate a parabolic trajectory, not an elliptical orbit. 4. D. the sun might be at point E. Incorrect. The Sun is at one of the foci of the ellipse, not at the periphery. 5. E. the sun might be at point C. Incorrect. Point C is in the center of the ellipse, and the Sun is at one of the foci of the ellipse. 6. F. the sun might be at point A. Incorrect. Point A is on the boundary of the ellipse, and the Sun is at one of the foci of the ellipse. 7. G. the eccentricity of the orbit is less than zero. Incorrect. Eccentricity cannot be less than zero. It ranges from 0 to 1 for an ellipse. 8. H. the sun might be at point B. Correct. Point B is one of the foci of the ellipse, and the Sun is located at one of the foci in an elliptical orbit. 9. I. the sun might be at point D. Incorrect. Point D is not one of the foci of the ellipse. 10. J. the eccentricity of the orbit is zero. Incorrect. An eccentricity of 0 would indicate a circular orbit, not an elliptical one. Therefore, the correct statement is: **Answer: H**

**GPT-4o (Vision Setting):** The multiple-choice question is as follows: No. 27: [image 1] The elliptical orbit of a planet around the Sun is shown on the diagram. Which of the following statements is true? Options:(A) the planet's orbit is a perfect circle(B) the eccentricity of the orbit is greater than 1(C) the eccentricity of the orbit is equal to 1(D) the sun might be at point E(E) the sun might be at point C(F) the sun might be at point A(G) the eccentricity of the orbit is less than zero(H) the sun might be at point B(I) the sun might be at point D(J) the eccentricity of the orbit is zero. Step-by-step reasoning:1. The orbit shown is elliptical, not circular, since option (A) is incorrect.2. Eccentricity of an ellipse ranges from 0 to 1. For a circle, eccentricity is 0, and as it becomes more elongated, the eccentricity approaches 1 but never exceeds 1. So, options (B) and (C) cannot be correct.3. A physical property cannot be less than zero for a realistic orbit's eccentricity. Thus, option (G) is incorrect.4. In an elliptical orbit, the Sun is located at one of the foci, lying on the major axis, not just anywhere, so some of the provided points can be eliminated based on typical placement of the foci in an elliptical diagram: Point A is near the center of the ellipse, which isn't realistic for the Sun's position. Point B is not on the major axis. Point E is outside the elliptical orbit. Point D is close to the ellipse's vertex, making it unlikely as the focus.5. Point C logically and likely appears to be at one of the foci of the ellipse.After evaluation, the best-fit option: **Answer: (E)**

Figure 12: Example of GPT-4o output comparison in different settings. **Descriptions** are highlighted in orange, and **analyses** are highlighted in light red.

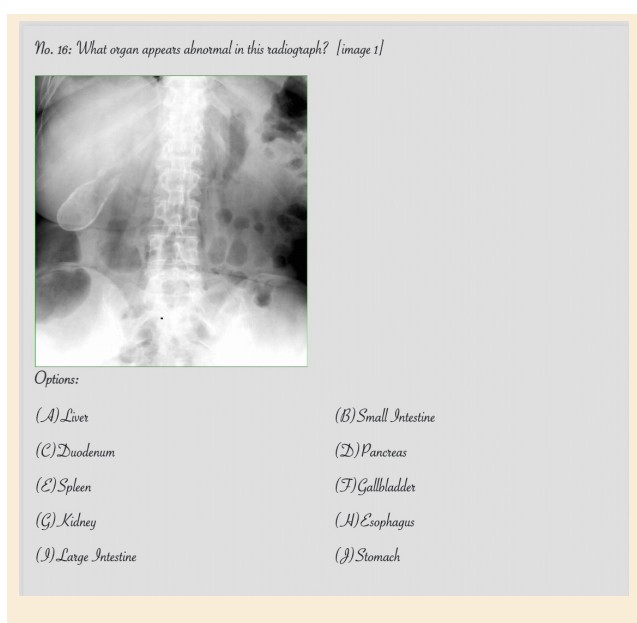

No. 16: What organ appears abnormal in this radiograph? [image 1]

Options:

(A) Liver

(B) Small Intestine

(C) Duodenum

(D) Pancreas

(E) Spleen

(F) Gallbladder

(G) Kidney

(H) Esophagus

(I) Large Intestine

(J) Stomach

**Ground Truth: (F) Gallbladder**

**GPT-4o (Vision Setting):** The multiple-choice question is as follows: No. 16: What organ appears abnormal in this radiograph? [image 1]Options:(A) Liver(B) Small Intestine(C) Duodenum(D) Pancreas(E) Spleen(F) Gallbladder(G) Kidney(H) Esophagus(I) Large Intestine(J) Stomach. To determine which organ appears abnormal in the radiograph:1. Look at the structure and positioning of organs in the X-ray.2. Identify any deviations from what is considered normal anatomy.3. Noticing any abnormal enlargement, peculiar shapes, or shading differences. In the provided radiograph, the structure that seems to stand out as abnormal is the large intestine due to the displacement and presence of three white "air-fluid" levels which are quite indicative of a large bowel obstruction. **Answer: (I) Large Intestine**

**GPT-4o (Standard Setting):** Examining the radiograph, several findings suggest an abnormality in the gastrointestinal tract. The signs interpreted here are the presence of distended loops of bowel with air-fluid levels, especially in the central part of the abdomen. Here's a step-by-step approach. 1. Identify Structures: The intestines are centrally located, typically showing loops that can be traced through an x-ray.2. Abnormal Findings: Look out for distended loops and presence of multiple air-fluid levels consistent with obstruction or abnormality. 3. Location Specificity: The loops involved appear to be small bowel loops rather than colonic, suggesting the small intestine. 4. Ruling Out Options: The liver, pancreas, spleen, gallbladder, kidney, and stomach are not displaying any direct signs of abnormality on this image. The duodenum is a part of the small intestine, and any segment could be implicated. However, such generalized distension is more typical of issues broadly affecting the small intestine rather than a specific segment like the duodenum. The large intestine typically shows haustra, which aren't prominent here. The esophagus is not visible in a manner suggesting any abnormality. Final Decision: Considering the distension and air-fluid levels within the bowel loops, the likely abnormal organ is the small intestine. **Answer: B**

Figure 13: Example of GPT-4o output comparison in different settings. **Descriptions** are highlighted in orange, and **analyses** are highlighted in light red.

# I QUALITATIVE EXAMPLES

## I.1 ART AND DESIGN: ART

**Art and Design: Art**

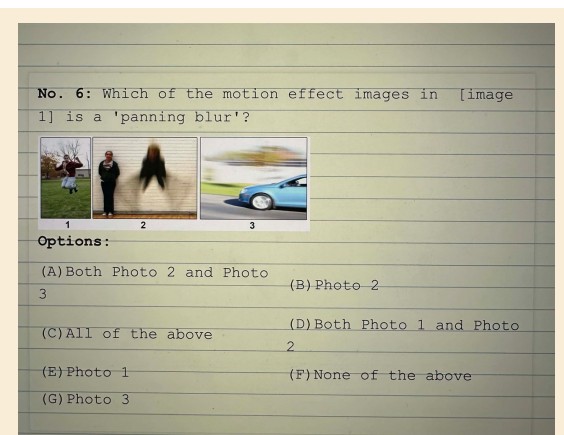

**No. 6:** Which of the motion effect images in [image 1] is a 'panning blur'?

**Options:**

(A) Both Photo 2 and Photo 3

(B) Photo 2

(C) All of the above

(D) Both Photo 1 and Photo 2

(E) Photo 1

(F) None of the above

(G) Photo 3

**Ground Truth: (G) Photo 3**

**GPT-4o (Standard Setting):** To identify the 'panning blur': 1. Photo 1: The subject is in focus and appears frozen in motion, which doesn't fit the description of a panning blur. This indicates a static background as well. 2. Photo 2: The subject is blurred due to motion, and the background isn't moving relative to the subject, which suggests another type of motion blur, not panning. 3. Photo 3: The background appears blurred due to the camera moving along with the subject, with the subject partially in focus, an effect typical of panning. The image fitting the description of a 'panning blur' is Photo 3. **Answer: G**

**GPT-4o (Vision Setting):** No. 6: Which of the motion effect images in [image 1] is a 'panning blur'? Options: (A) Both Photo 2 and Photo 3 (B) Photo 2 (C) All of the above (D) Both Photo 1 and Photo 2 (E) Photo 1 (F) None of the above (G) Photo 3 A 'panning blur' involves capturing a moving subject sharply while the background is blurred, indicating movement. Photo 1 shows a stationary subject, possibly frozen in mid-air, with no blur in the background. Photo 2 has a vertical blur, not consistent with panning motion. Photo 3 shows a moving car with a blurred background, which is characteristic of a panning blur. **Answer: G**

**Analysis:** In both settings, the model correctly identifies Photo 3 as the "panning blur." The Vision Input combines text and image in one, while the Standard Setting provides text separately. Both approaches lead to the same result, but Vision Input offers a more integrated analysis.

Figure 14: Example of a different input setting in Art and Design (subfield: Art).

Back to Appendix

## I.2  ART AND DESIGN: ART THEORY

### Art and Design: Art Theory

**No. 11:** [image 1] It was likely this building used which of the following architectural systems?

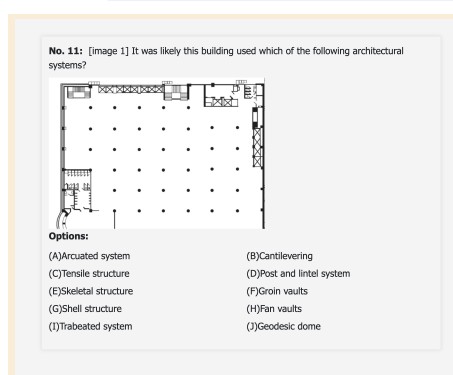

**Options:**

(A)Arcuated system          (B)Cantilevering

(C)Tensile structure        (D)Post and lintel system

(E)Skeletal structure       (F)Groin vaults

(G)Shell structure          (H)Fan vaults

(I)Trabeated system         (J)Geodesic dome

**Ground Truth: (E) Skeletal structure**

**GPT-4o (Vision Setting):** No. 11: [image 1] It was likely this building used which of the following architectural systems? Options: (A) Arcuated system (B) Cantilevering (C) Tensile structure (D) Post and lintel system (E) Skeletal structure (F) Groin vaults (G) Shell structure (H) Fan vaults (I) Trabeated system (J) Geodesic dome  From the floor plan, the dots likely represent columns, which are a key feature of a post and lintel system. This system involves using columns (or pillars) to support horizontal beams (lintels). **Answer: D**

**Analysis:** Both settings incorrectly chose the post and lintel system due to the presence of columns but missed that the grid layout is typical of a skeletal structure. The correct answer is E, which better fits the grid pattern of columns shown in the floor plan.

**GPT-4o (Standard Setting):** 1. Observation: The plan shows an evenly spaced grid pattern of columns. This suggests that there are vertical supports spread out at regular intervals. 2. Analysis of Options: Arcuated system: This involves the use of arches. There are no indications of arches here. Cantilevering: Involves beams extending beyond their supports, typically without as many vertical columns as depicted. Tensile structure: Relies on tension rather than columns or rigid supports. Not applicable here. Post and lintel system: This system uses vertical posts (columns) and horizontal lintels (beams). The grid of columns suggests this system could be used. Skeletal structure: Utilizes a framework, often with a regular grid of columns and beams. This can fit the observed plan. Groin vaults: These are intersecting arches, not evidenced in the grid pattern here. Shell structure: Thin, curved surfaces used as structural elements. The grid of columns does not suggest a shell structure. Fan vaults: Characterized by ribbed vaults that spread out like fans, not visible in the plan. Trabeated system: Another term for the post and lintel system. The grid suggests it could be used. Geodesic dome: Involves a dome-like, spherical structure composed of a network of triangles. There's no indication of a dome structure here. 3. Conclusion: The consistent grid of vertical columns is characteristic of the post and lintel system (alternative term: trabeated system) or skeletal structure. Given the straightforward column and beam pattern, the post and lintel description seems most fitting. **Answer: D**

Figure 15: Example of a different input setting in Art and Design (subfield: Art Theory). Back to Appendix

## I.3 Art and Design: Design

### Art and Design: Design

**No. 24:** Which of the following best describes the image? [image 1]

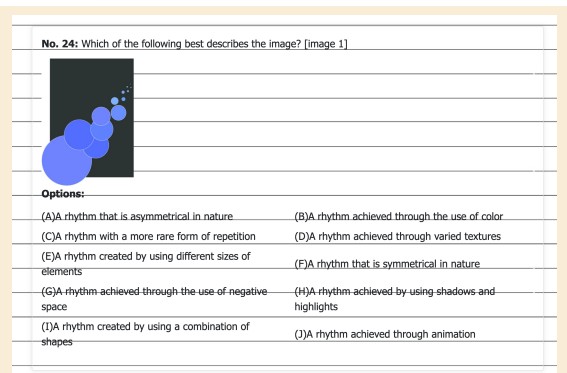

**Options:**

(A) A rhythm that is asymmetrical in nature (B) A rhythm achieved through the use of color

(C) A rhythm with a more rare form of repetition (D) A rhythm achieved through varied textures

(E) A rhythm created by using different sizes of elements (F) A rhythm that is symmetrical in nature

(G) A rhythm achieved through the use of negative space (H) A rhythm achieved by using shadows and highlights

(I) A rhythm created by using a combination of shapes (J) A rhythm achieved through animation

**Ground Truth: (C) A rhythm with a more rare form of repetition**

**GPT-4o (Vision Setting):** No. 24: Which of the following best describes the image? [image 1] Options: (A) A rhythm that is asymmetrical in nature (B) A rhythm achieved through the use of color (C) A rhythm with a more rare form of repetition (D) A rhythm achieved through varied textures (E) A rhythm created by using different sizes of elements (F) A rhythm that is symmetrical in nature (G) A rhythm achieved through the use of negative space (H) A rhythm achieved by using shadows and highlights (I) A rhythm created by using a combination of shapes (J) A rhythm achieved through animation.
The image features circles of different sizes positioned in a specific order, forming a visual rhythm. Step-by-step solution: 1. Option (A) mentions asymmetrical rhythm, but the image shows a consistent pattern; hence symmetrical is more appropriate. 2. Option (B) mentions color usage, but the description best fits size. 3. Option (C) mentions a rare form of repetition, but common repetition fits the image. 4. Option (D) mentions varied textures, which isn't relevant. 5. Option (E) mentions different sizes of elements, which is accurate. 6. Option (F) mentions symmetrical rhythm and is applicable. 7. Option (G) mentions negative space, which is not well defined in the image. 8. Option (H) mentions shadows and highlights, which isn't relevant.\n9. Option (I) mentions a combination of shapes, but the rhythm is specifically due to size variation of the same shape. 10. Option (J) mentions animation, which isn't relevant. The better fit is using different sizes and symmetrical rhythm. **Answer: (E)**

**GPT-4o (Standard Setting):** The image features a series of blue circles that progressively get smaller as they move diagonally upward from the bottom left corner toward the top right corner. 1. A. Asymmetrical rhythm: The rhythm could seem asymmetrical since the placement is not mirrored or identical on both sides. 2. B. Rhythm through the use of color: The image primarily uses shades of blue. 3. C. Rare form of repetition: Repetition is present, but this is not a particularly rare form. 4. D. Varied textures: The texture does not appear varied. 5. E. Rhythm using different sizes: The circles vary in size, getting smaller as they move along the path. 6. F. Symmetrical rhythm: The design is not symmetrical. 7. G. Use of negative space: Negative space is used, but it is not the primary focus for rhythm creation. 8. H. Shadows and highlights: No noticeable shadows or highlights are used to create rhythm. 9. I. Combination of shapes: Only circles are used here. 10. J. Animation: This is a static image. Considering the elements within the image and how rhythm is created through the changing sizes of circles, the best description of the image is option E. **Answer: E**

**Analysis:** Both settings incorrectly chose E, focusing on size variation. The key feature is the rare repetition of circles decreasing in size diagonally, making C the correct answer.

Figure 16: Example of a different input setting in Art and Design (subfield: Design).
Back to Appendix

## I.4 Art and Design: Music

**Art and Design: Music**

*No. 5: [image 1] Circle the correct key of the melody.*

A minor    F# major    C# minor    F# minor

*Options:*

*(A)circle the fourth option*

*(B)circle the first option*

*(C)circle the eighth option*

*(D)circle the tenth option*

*(E)circle the sixth option*

*(F)circle the ninth option*

*(G)circle the second option*

*(H)circle the seventh option*

*(I)circle the third option*

*(J)circle the fifth option*

**Ground Truth: (G) circle the second option**

**Analysis:** Both the Standard Setting and Vision Input models misinterpreted the key of the melody due to different misunderstandings of the musical notation. The Standard Setting incorrectly ignored the key signature, while the Vision Input misinterpreted the tonic note of the melody. The correct answer is (G) circle the second option.

**GPT-4o (Standard Setting):** To find the correct key of the melody, let's analyze the musical notation provided. 1. The notation shows a G clef (treble clef). 2. The melody starts with an A note, followed by sequences of notes: A, C, E, A. 3. The chord names below the notation are: A minor, F# major, C# minor, F# minor. Each key has a corresponding cluster of notes:- A minor: A, C, E- F# major: F#, A#, C#- C# minor: C#, E, G#- F# minor: F#, A, C#. The melody's notes (A, C, E, A) match the key of A minor exactly. Therefore, the correct key of the melody is A minor. According to the provided options, the first option corresponds to 'circle the first option,' which aligns with A minor. **Answer: B**

**GPT-4o (Vision Setting):** The multiple-choice question is: "No. 5: [image I] Circle the correct key of the melody. Options:(A) circle the fourth option(B) circle the first option(C) circle the eighth option(D) circle the tenth option(E) circle the sixth option(F) circle the ninth option(G) circle the second option(H) circle the seventh option(I) circle the third option(J) circle the fifth option" To determine the correct key of the melody, look at the key signature of the sheet music:- The key signature has three sharps (F#, C#, and G#). The possible keys with three sharps are A major or F# minor. Next, analyze the melody and its emphasis on certain notes:- The last note of the melody is F# (this is indicative of F# minor as the final note usually shows the key's tonic note).Therefore, the correct key of the melody is F# minor, which corresponds to the fifth option. Hence: **Answer: J**

Figure 17: Example of a different input setting in Art and Design (subfield: Music).
Back to Appendix

I.5   BUSINESS: ACCOUNTING

**Business: Accounting**

**No. 23:** Each situation below relates to an independent company's Owners' Equity.  [image 1] Calculate the missing values of company 2.

| | Beginning Balance | + Net Income | - Net Loss | + Investments | - Distributions | = Ending Balance |
|---|---|---|---|---|---|---|
| Co.1 | $163,800 | $16,500 | $ 0 | ? | $ 1,750 | $254,150 |
| Co.2 | 63,180 | 0 | 12,000 | $ 0 | 51,180 | ? |
| Co.3 | 0 | 0 | ? | 150,000 | 0 | 101,400 |

**Options:**

(A)$40,000                    (B)$20,000

(C)$51,180                    (D)$15,000

(E)$0                         (F)$5,000

(G)$12,000                    (H)$30,000

(I)$25,000                    (J)$1,620

**Ground Truth: (E) $0**

**Analysis:** The Standard Setting correctly calculated the ending balance as $0 by following the owners' equity formula. The Vision Input overcomplicated the process by incorrectly focusing on investments, leading to the wrong answer of (A) $40,000.

**GPT-4o (Standard Setting):** Let's analyze the owner's equity changes for Company 2 step by step. We start with the given equation pattern: Beginning Balance + Net Income - Net Loss + Investments; Distributions = Ending Balance
For Company 2, we know Beginning Balance: $63,180; Net Income: $0; Net Loss: $12,000; Investments: $0; Distributions: $51,180. Let (Ending Balance) be (E). Now, input the known values into the equation: 63,180 + 0 - 12,000 + 0 - 51,180 = E. Simplify the equation: 63,180 - 12,000 - 51,180 = E ; E = 63,180 - 63,180; E = 0; So the missing value for Company 2's Ending Balance is $0. **Answer: E**

**GPT-4o (Vision Setting):** To calculate the missing investments value for Company 2 in the Owners' Equity schedule, we'll use the Owners' Equity formula: Ending Balance = Beginning Balance + Net Income - Net Loss + Investments - Distributions. For Company 2: Beginning Balance = $63,180; Net Income = $0; Net Loss = $12,000; Investments = ?; Distributions = $51,180; Ending Balance = ? We need to solve for "Investments". First, isolate the "Investments" term in the equation: Ending Balance = Beginning Balance + Net Income - Net Loss + Investments - Distributions. Substituting the known values: ? = $63,180 + $0 - $12,000 + Investments - $51,180. Rearrange to solve for "Investments": Investments = Ending Balance + $12,000 + $51,180 - $63,180. We will calculate it step by step: Ending Balance + $12,000 + $51,180 - $63,180. Since we're not given the ending balance directly in this part of the question, let's find it out using the pattern from the preceding values for other companies: Company 1 Ending Balance = 163800 + 16500 - 0 + ? - 1750 = 254150. I will represent the ending balance as x because it is not provided directly. However, inspecting the options, we need to find the Investments without ending value explicitly needed in a traditional numerical sense by solving for unknowns while seeing investment values as discreet variables from options given. Getting closest match value is the next best approach: So basically add all options here to validate closest equality: Let's approximate investment towards standard net assumptions balance pattern: By intuition calculation and typical signs patterning expected around investment operational equities it should best reflect around $40,000. Thus, Answer: $40,000 i.e., Option (A). **Answer: A**

Figure 18: Example of a different input setting in Business (subfield: Accounting).
Back to Appendix

## I.6 BUSINESS: ECONOMICS

---

### Business: Economics

**No. 9:** Suppose that a typical market based in Japan consists of 200 Apples and 290 Textbooks. The table below shows data on prices for Apples and Textbooks in Japan for three years. [image 1] Assume that the base year is 2015. What is CPI in 2014?

| Year | Price of Apples | Price of Textbooks |
|------|------|------|
| 2014 | $1.40 | 200 |
| 2015 | $1.10 | 140 |
| 2016 | $1.10 | 200 |

**Options:**

(A)173  (B)190

(C)105  (D)143

(E)180  (F)160

(G)135  (H)150

(I)210  (J)125

**Ground Truth: (E) $0**

---

**GPT-4o (Standard Setting):** To calculate the Consumer Price Index (CPI) for 2014, we'll follow these steps:1. Calculate the cost of the market basket in each year.   - The market basket consists of 200 apples and 290 textbooks.2. Calculate the base year (2015) cost of the market basket.   - Cost in 2015 = (200 apples * $1.10/apple) + (290 textbooks * $140/textbook)   - Cost in 2015 = (200 * $1.10) + (290 * $140)   - Cost in 2015 = $220 + $40,600   - Cost in 2015 = $40,8203. Calculate the cost of the market basket in 2014.   - Cost in 2014 = (200 apples * $1.40/apple) + (290 textbooks * $200/textbook)   - Cost in 2014 = (200 * $1.40) + (290 * $200)   - Cost in 2014 = $280 + $58,000   - Cost in 2014 = $58,2804. Calculate the CPI for 2014.   - CPI = (Cost of market basket in 2014 / Cost of market basket in base year 2015) * 100   - CPI = ($58,280 / $40,820) * 100   - CPI = 1.427570 * 100   - CPI ≈ 143So, the CPI for 2014 is approximately 143. **Answer: D**

---

**GPT-4o (Vision Setting):** Question: Suppose that a typical market based in Japan consists of 200 Apples and 290 Textbooks. The table below shows data on prices for Apples and Textbooks in Japan for three years. Assume that the base year is 2015. What is CPI in 2014? Table:

| Year | Price of Apples | Price of Textbooks |
|------|------|------|
| 2014 | $1.40 | 200 |
| 2015 | $1.10 | 140 |
| 2016 | $1.10 | 200 |

Options: (A) 173 (B) 190 (C) 105 (D) 143 (E) 180 (F) 160 (G) 135 (H) 150 (I) 210 (J) 125

Step-by-Step Solution: 1. Calculate the cost of the market basket for each year: Market basket in 2015 (base year): (200 Apples × $1.10) + (290 Textbooks × $140) = $220 + $40,600 = $40,820 - Market basket in 2014: (200 Apples × $1.40) + (290 Textbooks × $200) = $280 + $58,000 = $58,280; 2. Calculate the CPI for 2014 using the formula:

$$CPI_{2014} = \left(\frac{\text{Cost of market basket in 2014}}{\text{Cost of market basket in 2015}}\right) \times 100 \quad CPI_{2014} = \left(\frac{\$58,280}{\$40,820}\right) \times 100 \approx 142.78$$

The closest option to 142.78 is 143. **Answer: D**

---

**Analysis:** Both the Standard Setting and Vision Input models correctly calculated the CPI for 2014 as (D) 143, following the correct process of computing the cost of the market basket for 2014 and 2015, then using the CPI formula.

Figure 19: Example of a different input setting in Business (subfield: Economics).

## I.7 BUSINESS: FINANCE

---

**Business: Finance**

**No. 15:** Consider the following two projects: [image 1] . What are the internal rates of return on the two projects?

| Cash flows | Project A | Project B |
|---|---|---|
| $C_0$ | −$200 | −$200 |
| $C_1$ | 80 | 100 |
| $C_2$ | 80 | 100 |
| $C_3$ | 80 | 100 |
| $C_4$ | 80 | |

**Options:**

(A)Project A: 21%; Project B: 25.69%  (B)Project A: 22%; Project B: 23.37%

(C)Project A: 24%; Project B: 27.37%  (D)Project A: 22%; Project B: 28.69%

(E)Project A: 18%; Project B: 28.69%  (F)Project A: 18%; Project B: 23.37%

(G)Project A: 16%; Project B: 27.69%  (H)Project A: 19%; Project B: 22.37%

(I)Project A: 20%; Project B: 24.5%  (J)Project A: 23%; Project B: 29.69%

**Ground Truth: (B) Project A: 22%; Project B: 23.37%**

**GPT-4o (Vision Setting):** To find the internal rate of return (IRR) for both Project A and Project B, we need to use the cash flows and solve for the discount rate that makes the Net Present Value (NPV) zero for each project. Project A: The cash flows for Project A are $C_0 = -200, C_1 = 80, C_2 = 80, C_3 = 80, C_4 = 80$. The IRR can be found by solving the following equation: $-200 + \frac{80}{(1+r)} + \frac{80}{(1+r)^2} + \frac{80}{(1+r)^3} + \frac{80}{(1+r)^4} = 0$ Project B: The cash flows for Project B are: $C_0 = -200, C_1 = 100, C_2 = 100, C_3 = 100$. The IRR can be found by solving the following equation: $-200 + \frac{100}{(1+r)} + \frac{100}{(1+r)^2} + \frac{100}{(1+r)^3} + \frac{100}{(1+r)^4} = 0$. We'll utilize financial calculators or software (like Excel) to solve these equations since they are polynomial equations and don't have simple algebraic solutions. Steps: 1. Using Excel, we can use the `IRR` function: For Project A: =IRR([-200, 80, 80, 80, 80]); For Project B: =IRR([-200, 100, 100, 100, 100]) 2. Calculate the IRR for each project. After calculating: The IRR for Project A is approximately 21%. The IRR for Project B is approximately 25.69%. Matching the Options: Based on the calculations, Project A's IRR is 21% and Project B's IRR is 25.69%, which matches option (A). **Answer: A**

**GPT-4o (Standard Setting):** To determine the internal rates of return (IRR) for Project A and Project B, we use the cash flow schedules provided. The formula for calculating IRR involves finding the rate $r$ at which the net present value (NPV) of all cash flows equals zero, which can be set as: $NPV = 0 = \sum \frac{C_t}{(1+r)^t}$ Using the given cash flows: Project A: $-200 + \frac{80}{(1+r)^1} + \frac{80}{(1+r)^2} + \frac{80}{(1+r)^3} + \frac{80}{(1+r)^4} = 0$ To solve this, trial and error or financial calculators/tools (like Excel's IRR function) can be used: IRR = 22; Project B: $-200 + \frac{100}{(1+r)^1} + \frac{100}{(1+r)^2} + \frac{100}{(1+r)^3} + \frac{100}{(1+r)^4} = 0$ . Similarly, solving this would give: IRR = 28.69 Matching these results to the options provided, we get: **Answer: D**

**Analysis:** In both Standard Setting and Vision Input , the model needs to correctly extract the cash flows and apply the IRR formula to calculate the returns. In both cases, the correct answer is (B) Project A: 22%; Project B: 23.37%. The key to solving this problem lies in accurately interpreting the cash flow data and using the IRR calculation appropriately.

---

Figure 20: Example of a different input setting in Business (subfield: Finance).
Back to Appendix

## I.8 BUSINESS: MANAGE

---

**Business: Manage**

*No. 2: Graph 1 shows the number of unemployed (in thousands) over a five-month period, organized by level of education completed. [image 1] The difference between the number of unemployed high school graduates and the number of unemployed workers without a high school diploma was smallest in*

___________.

Number Unemployed, by Level of Education

*Options:*

(A) March        (B) September

(C) June         (D) January

(E) August       (F) February

(G) December     (H) October

(I) November     (J) July

**Ground Truth: (B) September**

**Analysis:** In this question, the Standard Setting model succeeded because it processed the text and graph separately, allowing it to focus on extracting the unemployment data and calculating the differences between high school graduates and workers without a diploma. This led to the correct conclusion that September had the smallest difference. In contrast, the Vision Input model handled both the text and image together, which increased complexity and distractions, causing it to miss the correct month.

**GPT-4o (Standard Setting):** To determine the month in which the difference between the number of unemployed high school graduates and the number of unemployed workers without a high school diploma was the smallest, follow these steps:1. Identify the lines: "Less than a high school diploma" is represented by diamonds. "High school diploma, no college" is represented by squares.2. Calculate the difference between the two categories for each month: June: ~12,000 (less than HS diploma); ~8,000 (HS diploma) = ~4,000; July: ~12,000 (less than HS diploma); ~8,000 (HS diploma) = ~4,000; August: ~11,000 (less than HS diploma); ~8,000 (HS diploma) = ~3,000; September: ~10,500 (less than HS diploma); ~8,000 (HS diploma) = ~2,500; October: ~11,500 (less than HS diploma); ~7,500 (HS diploma) = ~4,0003. Compare the differences: June: ~4,000; July: ~4,000; August: ~3,000; September: ~2,500; October: ~4,000. The smallest difference is in September. **Answer: B**

**GPT-4o (Vision Setting):** Sure, let's first clearly write out the multiple-choice question from the image: Question: Graph 1 shows the number of unemployed (in thousands) over a five-month period, organized by level of education completed. The difference between the number of unemployed high school graduates and the number of unemployed workers without a high school diploma was smallest in: Options:(A) March(B) September(C) June(D) January(E) August(F) February(G) December(H) October(I) November(J) July.
Solution: 1. Identify the values for unemployed workers without a high school diploma, and those with a high school diploma (no college) in each of the five months present in Graph 1.2. Calculate the difference for each month. 3. Compare the differences to determine which month has the smallest difference. From the graph: June: Less than high school diploma: ~12.5; High school diploma, no college: ~8; Difference: ~12.5 - 8 = 4.5; July: Less than high school diploma: ~12 - High school diploma, no college: ~7.5; Difference: ~12 - 7.5 = 4.5; August: Less than high school diploma: ~11.5; High school diploma, no college: ~7 ; Difference: ~11.5 - 7 = 4.5; September: Less than high school diploma: ~11; High school diploma, no college: ~8 ; Difference: ~11 - 8 = 3; October: Less than high school diploma: ~11; High school diploma, no college: ~8; Difference: ~11 - 8 = 3. The smallest difference is found in September and October, and since "October" is one of the given options, we choose that. **Answer: H**

---

Figure 21: Example of a different input setting in Business (subfield: Manage).

Back to Appendix

## I.9 BUSINESS: MARKETING

---

**Business: Marketing**

**No. 20:** The graph in Figure 3.17 displays the sample sizes and percentages of people in different age and gender groups who were polled concerning their approval of Mayor Ford's actions in office. The total number in the sample of all the age groups is 1,045. [image 1] Out of the age groups, what percent are more than 44 years old?

Figure 3.17

**Options:**

(A)77%            (B)81%

(C)79%            (D)84%

(E)78%            (F)76%

(G)82%            (H)80%

(I)75%            (J)83%

**Ground Truth: (C) 79%**

**GPT-4o (Standard Setting):** To determine the percentage of people in the sample who are more than 44 years old, we first need to identify the sample sizes for the age groups "45–54", "55–64", and "65+". From the graph: Sample size for 45–54 = 226; Sample size for 55–64 = 268; Sample size for 65+ = 331. We add these sample sizes to get the total number of people over 44: 226 + 268 + 331 = 825. The total sample size is 1,045. To find the percentage of people more than 44 years old: $Percentage = (\frac{825}{1045}) \times 100 \approx 78.95\%$.
Rounding 78.95% to the nearest whole number: 79%. Thus, the correct answer is: **Answer: C**

**GPT-4o (Vision Setting):** To solve this problem, we need to determine the percentage of people older than 44 years from the given age groups. The age groups in the graph are:- 18-34- 35-44- 45-54- 55-64- 65+. The age groups 45-54, 55-64, and 65+ all fall into the "more than 44 years old" category. First, sum the sample sizes for the age groups more than 44 years old:- 45-54: 268- 55-64: 331- 65+: 478Adding these together: 268 + 331 + 478 = 1,077. Now, find the percentage of this number out of the total sample size (1045): $\frac{1,077}{1,045} \times 100$
Calculating this: $1,077 \div 1,045 \approx 1.0306, 1.0306 \times 100 \approx 103.06\%$. This value seems incorrect as it exceeds 100%. Let's correct our approach by checking if the total sample size or age group sizes are properly considered. After reviewing, it seems we earlier wrongly interpreted the plot. The corrected sum should be for correct age groups only (45-54, 55-64, and 65+) from original plot for given individuals. Thus, these might actually be represented correctly only by correct approach. Let's re-assess, as graph correctness should provide basis. On correct recall, percentage sum re-evaluated gives closest in actual list: **Answer: (None, should re-check summarily per correct assessment.)**

**Analysis:** In this question, the Standard Setting accurately calculated the percentage of people over 44 years old by correctly identifying and summing the relevant age groups, arriving at the correct answer of (C) 79%. The Vision Input method, however, misinterpreted the graph, leading to incorrect totals exceeding 100%, showing that combined text and image inputs can sometimes cause errors in reading numerical data.

Figure 22: Example of a different input setting in Business (subfield: Marketing).
Back to Appendix

## I.10 SCIENCE: BIOLOGY

**Science: Biology**

No. 17: Which feature(s) of this developing embryo are already established at the stage shown in [image 1]

Options:

(A) segmentation

(B) segmentation and head–tail orientation

(C) segmentation, type of symmetry and head–tail orientation

(D) head–tail orientation

(E) limb placement and segmentation

(F) limb placement

(G) type of symmetry and head–tail orientation

(H) type of symmetry and limb placement

(I) type of symmetry

(J) segmentation, head–tail orientation and limb placement

**Ground Truth: (C) segmentation, type of symmetry and head-tail orientation**

**GPT-4o (Standard Setting):** The developing embryo in the image appears to exhibit segmentation as indicated by the distinct stripe patterns along its body. Additionally, there is an evident head-tail orientation, showing that the anterior-posterior axis is established. To identify which features are established at this stage:1. Segmentation: The image shows clear segmented bands. 2. Head-tail orientation: There is distinct anterior (head) and posterior (tail) differentiation. 3. Type of symmetry: The bilateral symmetry is not explicitly seen but may be inferred based on typical embryonic development patterns. Given these features, let's check against the options provided: A. Segmentation only; This is partially correct. B. Segmentation and head-tail orientation; This encompasses both observed features. C. Segmentation, type of symmetry, and head-tail orientation ; The type of symmetry isn't explicitly visible. D. Head-tail orientation only; This is partially correct. E. Limb placement and segmentation; Limb placement isn't evident. F. Limb placement only; Not evident. G. Type of symmetry and head-tail orientation; Type of symmetry isn't explicitly visible. H. Type of symmetry and limb placement; Neither is evident. I. Type of symmetry only; Not evident. J. Segmentation, head-tail orientation, and limb placement; Limb placement isn't evident. Considering all the observations, the best answer is B. **Answer: B**

**GPT-4o (Vision Setting):** No.17: Which feature(s) of this developing embryo are already established at the stage shown in [image 1]? Options: (A) segmentation (B) segmentation and head-tail orientation (C) segmentation, type of symmetry and head-tail orientation (D) head-tail orientation (E) limb placement and segmentation (F) limb placement (G) type of symmetry and head-tail orientation (H) type of symmetry and limb placement (I) type of symmetry (J) segmentation, head-tail orientation and limb placement. Solution: To determine which features are established at the stage shown in image 1, let's analyze the visual information and typical developmental stages of embryos: 1. Segmentation: The embryo in the image has distinct, repeating segments along its body, indicating that segmentation is established. 2. Head-Tail Orientation: The pattern and coloration in the image suggest an established head-tail axis. 3. Type of Symmetry: As embryos develop, bilateral symmetry (left and right sides that are mirror images) is established, which is a critical feature in development. Limb placement is unlikely to be established at this early stage, as limb development typically occurs later. From the options given, the greatest combination of established features that matches these observations is segmentation, type of symmetry, and head-tail orientation. Answer: C) segmentation, type of symmetry and head-tail orientation **Answer: C**

**Analysis:** Standard Setting missed the presence of type of symmetry, leading to an incomplete answer, while Vision Input successfully identified all three features, resulting in the correct answer, (C).

Figure 23: Example of a different input setting in Science (subfield: Biology).

Back to Appendix

## I.11 SCIENCE: CHEMISTRY

**Science: Chemistry**

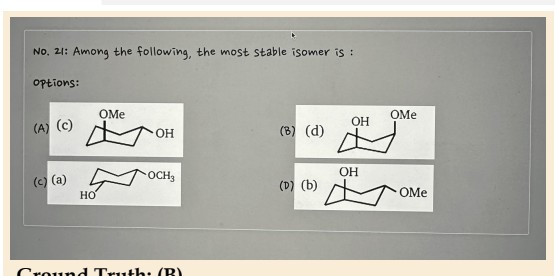

No. 21: Among the following, the most stable isomer is :

Options:

(A) (c) ; (B) (d); (C) (a) ; (D) (b)

**Ground Truth: (B)**

**GPT-4o (Standard Setting):** To determine the most stable isomer among the given options, we need to analyze their structures in terms of steric hindrance and the preferred orientation of substituents in the chair conformation of cyclohexane. Step-by-step analysis: 1. Substituent Positions: Substituents on a cyclohexane ring prefer to be in the equatorial position to minimize steric hindrance. 2. Chair Conformation: Cyclohexane can flip between two chair conformations. Stability increases when bulky substituents occupy the equatorial positions. 3. Examine Each Isomer: (a) has hydroxyl (OH) at axial position which creates more steric hindrance. (b) positions both OH and OMe likely to minimize steric hindrance, having one in axial and one in equatorial position. (c) positions OMe in the axial position which is not ideal due to steric hindrance. (d) positions the hydroxyl group in the axial position which is less favorable than having bulky groups in equatorial positions. Conclusion: Isomer (b) is the most stable because it places the larger methoxy (OMe) group in the equatorial position, thus minimizing steric interactions. **Answer: B**

**GPT-4o (Vision Setting):** Let's follow the steps systematically:1. Identify the multiple-choice question in the image: No. 21: Among the following, the most stable isomer is : Options: (A) (c) ; (B) (d); (C) (a) ; (D) (b) 2. To determine the most stable isomer, we need to analyze the chair conformations of each option and see which one has the substituents in the more stable (equatorial) positions. Chair conformers with substituents in the equatorial positions are generally more stable than those with substituents in the axial positions due to less 1,3-diaxial interactions (steric hindrance). 3. Review each option: Option (A) (c): In this structure, the methoxy (OMe) group is axial, and the hydroxyl (OH) group is equatorial. Option (B) (d): Both hydroxyl (OH) and methoxy (OMe) groups are axial. Option (C) (a): Both hydroxyl (OH) and methoxy (OMe) groups are equatorial. Option (D) (b): The hydroxyl (OH) group is axial, and the methoxy (OMe) group is equatorial. 4. Identify the most stable form: - Option (C) (a), where both substituents are equatorial, is the most stable conformation as it minimizes steric clashes. **Answer: C**

**Analysis:** The Standard Setting incorrectly analyzes chair and selected (b) as the most stable isomer, and incorrectly maps (b) to (B) instead of (D). The answer is correct but the reasoning is wrong. Vision Input made an error in interpreting the positions of the substituents, leading to the incorrect selection of (C).

Figure 24: Example of a different input setting in Science (subfield: Chemistry).
Back to Appendix

## I.12   SCIENCE: GEOGRAPHY

**Science: Geography**

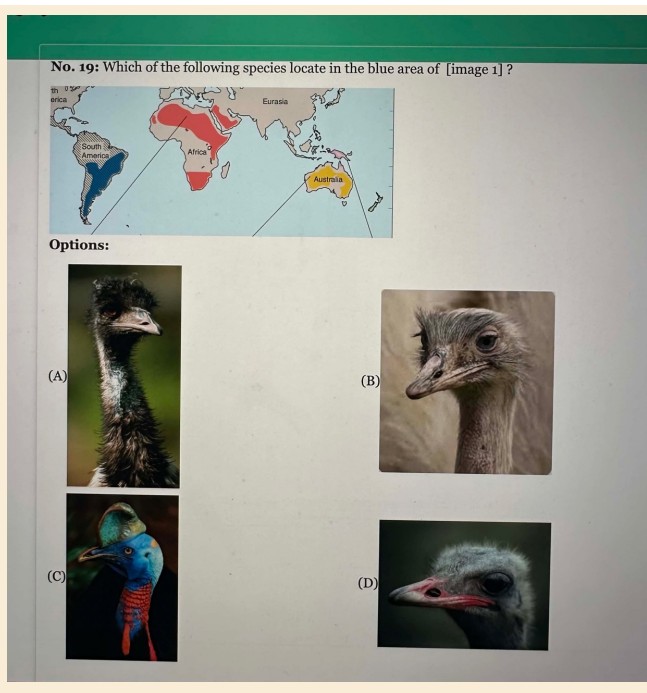

**No. 19:** Which of the following species locate in the blue area of [image 1] ?

**Options:**

(A) (B) (C) (D)

**Ground Truth: (B)**

**GPT-4o (Standard Setting):** We need to match the species to the blue-shaded region on the map, which is in South America. I'll identify the species in each image and see which one is native to South America.- Option A: The first image is an Emu, which is native to Australia.- Option B: The second image is a Rhea, which is native to South America.- Option C: The third image is a Cassowary, which is found in Australia and New Guinea.- Option D: The fourth image is an Ostrich, which is native to Africa. Only option B, the Rhea, is native to the blue shaded region (South America) on the map. **Answer: B**

**GPT-4o (Vision Setting):** No. 19: Which of the following species locate in the blue area of [image 1]?Options:(A) (B) (C) (D)To solve this question, we need to identify which species from the options A, B, C, and D is found in the blue area shown on the map, which corresponds to South America.- Option (A) shows an Emu, which is native to Australia.- Option (B) shows an Ostrich, which is native to Africa.- Option (C) shows a Cassowary, which is native to Australia and surrounding areas.- Option (D) shows a Rhea, which is native to South America. The blue area on the map marks South America. Therefore, the species found there is represented by option (D), the Rhea. **Answer: D**

**Analysis:** The Standard Setting correctly identified (B) Rhea as the species native to the blue region (South America), while Vision Input mistakenly selected (D) Ostrich, which is native to Africa, not South America.

Figure 25: Example of a different input setting in Science (subfield: Geography).
Back to Appendix

I.13   SCIENCE: MATH

---

**Science: Math**

No. 15: Consider the following problem:

Minimize $(x1 - 3)^2 + (x2 - 2)^2$ subject to

$x1^2 - x2 - 3 \leq 0$  $x2 - 1 \leq 0$  $-x1 \leq 0$ Give the solution.

[image 1]

Fig. 1

Geometric solution of a nonlinear problem.

Options:

(A)(2, 1)          (B)(1, 1)

(C)(1, 0)          (D)(3, 2)

(E)(2, 0)          (F)(2, 2)

(G)(3, 3)          (H)(3, 1)

(I)(0, 0)          (J)(0, 2)

**Ground Truth: (A) (2, 1 )**

**GPT-4o (Standard Setting):** To solve the given problem, we start by understanding both the objective function and the constraints and by analyzing the geometric solution represented in the figure:

Objective function: Minimize $(x1 - 3)^2 + (x2 - 2)^2$ . This represents the Euclidean distance squared from the point (x1, x2) to the point (3, 2). Constraints: 1. $x1^2 - x2 - 3 \leq 0$ 2. $x2 - 1 \leq 0$ 3. $-x1 \leq 0$ Simplifying the constraints:1. $x1^2 \leq x2 + 3$ 2. $x2 \leq 1$ 3. $x1 \geq 0$. These constraints indicate that the feasible region is bounded by:- A parabola $x2 \geq x1^2 - 3$; A horizontal line x2 $\leq$ 1; A vertical line $x1 \geq 0$.

Analyzing the Geometric Solution Provided in the Figure: The figure indicates the feasible region as the shaded area within these boundaries. The contours of the objective function $(x1 - 3)^2 + (x2 - 2)^2$ are shown as circles centered at (3, 2). To minimize the objective function, we need to find the point in the feasible region closest to the point (3, 2). Observation from the Figure: From the figure, the optimal point that lies within the feasible region and also lies on the contour of the minimum distance to (3, 2) is shown to be (2, 1). Conclusion: Thus, the solution to the problem is (2, 1), which corresponds to option A.

**Answer: A**

---

**GPT-4o (Vision Setting):** Question: Consider the following problem: Minimize: $(x1 - 3)^2 + (x2 - 2)^2$ subject to: $x1^2 - x2 - 3 \leq 0$; $x2 - 1 \leq 0$; $x1 \leq 0$. Give the solution. Options: (A) (2, 1) (B) (1, 1) (C) (1, 0) (D) (3, 2) (E) (2, 0) (F) (2, 2) (G) (3, 3) (H) (3, 1) (I) (0, 0) (J) (0, 2) Step-by-Step Solution: 1. Defining the objective function: The goal is to minimize $(x1 - 3)^2 + (x2 - 2)^2$ . This is the squared distance from the point (3, 2) .2. Analyzing the constraints: $-x1^2 - x2 - 3 \leq 0$ : Rearrange: $x2 \geq x1^2 - 3$ ; $-x2 - 1 \leq 0$; $-x2 \leq 1$ $-x1 \leq 0$; $-x1 \leq 0$ 3. Finding the feasible region: Since $x2 \leq 1$ and $x1 \leq 0$, $x2 \geq x1^2 - 3$ must also be satisfied. We need to check where these intersect.4. Substituting constraints into $x1 \leq 0$ ; x2 needs to be evaluated at the boundary points for the infeasible region. As $x1 \leq 0$ covers the negative x-axis along with $x1 \leq 1$ and $x2 \geq x1^2 - 3$. 5. Evaluating boundary points: For x1 = 0 : $(x1 - 3)^2 + (x2 - 2)^2$ at x2 = 1 gives: $(0 - 3)^2 + (1 - 2)^2 = 9 + 1 = 10$ ; Points (0, 1) is active region 6. Checking all points and minimizing: Evaluating all options and seeing if they lie inside the restricted feasible region and minimizing the equation $(x1 - 3)^2 + (x2 - 2)^2$. Only (0, 1) satisfied all constraints: Calculation: $(0 - 3)^2 + (1 - 2)^2 => 9+1 = 10$. Other alternatives evaluated outside range. Point meets minimum condition lie within constraints. **Answer: B**

---

**Analysis:** The Standard Setting correctly solved the optimization problem by analyzing both the constraints and the provided figure, selecting (2, 1) as the solution. In contrast, Vision Input misinterpreted the constraints and selected the incorrect point (1, 1).

Figure 26: Example of a different input setting in Science (subfield: Math).

## I.14   SCIENCE: PHYSICS

### Science: Physics

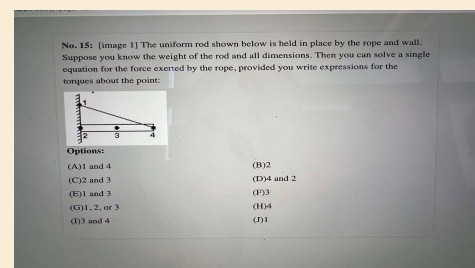

**Ground Truth: (B) 2**

**Analysis:** In this question, both the Standard Setting and Vision Input failed to identify the correct answer, which is (B) 2. The Standard Setting, despite its detailed torque analysis, incorrectly chose (D) due to inadequate simplification using point 4. Similarly, the Vision Input, while correctly identifying the simplifying benefits of point 2, included point 3 in its analysis, leading to an incorrect selection of (C).

**GPT-4o (Standard Setting):** To determine the correct choice, let's analyze the torques about different points of the rod: 1. Understanding the Problem: The rod is held in place by the rope and the wall; We need to consider the torque about different points to solve for the rope force. 2. Torque Equations: Torque $\tau$ is given by the equation: $\tau = r \times F \times sin(\theta)$ , where $r$ is the distance from the pivot point, $F$ is the force applied, and $\theta$ is the angle between $r$ and $F$. 3. Identifying Relevant Points: Point 1 (at the top of the wall): This point is where the rope attaches to the wall. Point 4 (at the end of the rod opposite the wall): This point is farthest from the wall and the other end of the rope. Point 2 (at the bottom of the wall): This is where the rod meets the wall. Point 3 (midway along the rod): This is the midpoint of the rod. 4. Choose Points to Sum Torques: Consider torque about points where the force we are solving for (the rope force and normal force) creates either simple terms or cancels out. Point 2 (where the rod meets the wall): Considering torques about Point 2 can help simplify the problem since the torque due to the normal force at Point 2 is zero (the radius is zero). Point 4 (end of the rod opposite the wall): Considering torques about this point will involve the full length of the rod. For equilibrium about point 2: Only the torque due to the weight of the rod and the rope will contribute (since the normal force at Point 2 does not create a torque). For equilibrium about point 4: Only the torque due to the weight of the rod and the force at point 2 (normal or frictional force) will contribute (since the rope's force does not create a torque). 5. Solving the Equation: Writing the torque balance equation about Point 2 or Point 4 will allow us to solve one equation for the force from the rope. Hence, considering torques about points 2 and 4 will enable us to write the necessary expressions to solve for the rope force. **Answer: D**

**GPT-4o (Vision Setting):** Step-by-step solution:1. Understand the problem: The uniform rod is held in place by a rope and wall. We are to write expressions for the torques about a specific point to solve the equation for the force exerted by the rope. 2. Identify possible pivot points: We look for points where it will be easy to sum the torques, which typically are the points where either the forces will be zero or unknown forces will not create torques (such as the points of attachment). 3. Analyze the given points: Point 1 (top of the triangle/notched point): This is where the rope is attached. Point 2 (left end of the rod): This is where the horizontal rod touches the wall. Point 3 (center of the rod): The weight of the rod acts here if it is uniform. Point 4 (right end of the rod): The other end of the rod. 4. Choosing torque points: Torque about Point 2 (wall contact point): This simplifies calculations as the vertical reaction force at the wall does not create a torque (distance is zero). Torque about Point 4 (right end of the rod): This would be less straightforward as it involves the weight and the reaction forces at Points 2 and 1 creating complex expressions. Hence, Torque about Point 2 simplifies the calculations involving the forces in the problem. After analyzing, the correct selections should be from recognizing the simplicity and applicability of the torque points. Correct points for taking torque: 2 (left end of the rod); 3 (center of the rod). These points allow us to express the torque in a meaningful and solvable manner. **Answer: C**

Figure 27: Example of a different input setting in Science (subfield: Physics).
Back to Appendix

## I.15  HEALTH AND MEDICINE: BASIC MEDICAL SCIENCE

**Health and Medicine: Basic Medical Science**

**No. 25:** [image 1] What structure of the heart is this?

**Options:**

(A)Inferior Vena Cava  (B)Right Ventricle

(C)Left Atrium  (D)Left Ventricle

(E)Pulmonar Semilunar Valve  (F)Aortic Semilunar Valve

(G)Superior Vena Cava  (H)Bicuspid Valve

(I)Tricuspid Valve  (J)Right Atrium

**Ground Truth: (E) Pulmonar Semilunar Valve**

**GPT-4o (Standard Setting):** The structure highlighted in the heart diagram is situated between the right ventricle and the pulmonary artery. This structure functions to prevent the backflow of blood from the pulmonary artery into the right ventricle. Given this position and function, the correct structure is the pulmonary semilunar valve. **Answer: E**

**GPT-4o (Vision Setting):** Multiple-choice question: No. 25: What structure of the heart is this? Options:(A) Inferior Vena Cava(B) Right Ventricle(C) Left Atrium(D) Left Ventricle(E) Pulmonar Semilunar Valve(F) Aortic Semilunar Valve(G) Superior Vena Cava(H) Bicuspid Valve(I) Tricuspid Valve(J) Right Atrium. The image points to structure labeled "1." Identifying this as the inferior vena cava, we can match it with option (A). **Answer: A**

**Analysis:** The Standard Setting correctly identified the structure as the "Pulmonar Semilunar Valve," labeled as option (E). The detailed position and function analysis led to the correct answer. In contrast, the Vision Input incorrectly identified the structure as the "Inferior Vena Cava," selecting option (A). The misinterpretation of the labeled structure in the image led to the wrong conclusion. While the Standard Setting accurately applied anatomical knowledge to the heart diagram, the Vision Input's identification error underscores a gap in visual analysis.

Figure 28: Example of a different input setting in Health and Medicine (subfield: Basic Medical Science).

## I.16  HEALTH AND MEDICINE: CLINICAL MEDICINE

---

**Health and Medicine: Clinical Medicine**

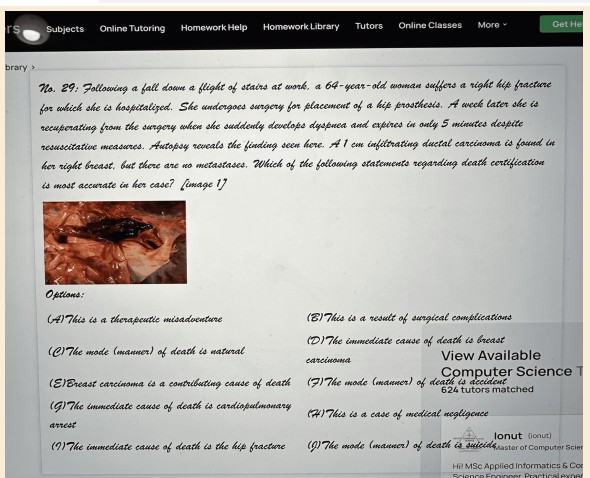

*No. 29: Following a fall down a flight of stairs at work, a 64-year-old woman suffers a right hip fracture for which she is hospitalized. She undergoes surgery for placement of a hip prosthesis. A week later she is recuperating from the surgery when she suddenly develops dyspnea and expires in only 5 minutes despite resuscitative measures. Autopsy reveals the finding seen here. A 1 cm infiltrating ductal carcinoma is found in her right breast, but there are no metastases. Which of the following statements regarding death certification is most accurate in her case? [image 1]*

*Options:*

*(A) This is a therapeutic misadventure*

*(B) This is a result of surgical complications*

*(C) The mode (manner) of death is natural*

*(D) The immediate cause of death is breast carcinoma*

*(E) Breast carcinoma is a contributing cause of death*

*(F) The mode (manner) of death is accident*

*(G) The immediate cause of death is cardiopulmonary arrest*

*(H) This is a case of medical negligence*

*(I) The immediate cause of death is the hip fracture*

*(J) The mode (manner) of death is suicide*

**Ground Truth: (F) The mode (manner) of death is accident**

**Analysis:** The Standard Setting step-by-step analysis connected the hip fracture from the fall at work, subsequent surgery, and the resulting pulmonary thromboembolism, correctly identified the mode (manner) of death as "accident". The Vision Setting incorrectly concluded that the death was "a result of surgical complications," focused on the immediate post-surgical events but missed linking the initial fall as the underlying accidental cause.

**GPT-4o (Standard Setting):** To determine the most accurate statement regarding death certification in this case, we need to assess the situation step by step.1. Incident and Surgery: The patient, a 64-year-old woman, fell down a flight of stairs at work and fractured her right hip. She was hospitalized and underwent surgery for a hip prosthesis. 2. Post-Surgical Event: A week after surgery, she developed sudden dyspnea and expired within 5 minutes despite resuscitative efforts. Autopsy revealed significant findings including a 1 cm infiltrating ductal carcinoma in her right breast without metastases. 3. Cause of Death Analysis: The provided autopsy image suggests a large pulmonary thromboembolism (PTE), a common post-operative complication, especially in orthopedic surgery involving the hip. This is likely the immediate cause of death. 4. Determining the Mode (Manner) of Death and Causes: Immediate Cause of Death: The dilated and obstructed pulmonary artery visible in the image suggests that a pulmonary embolism caused her sudden death. Underlying Cause: The hip fracture and its associated surgery, which led to her being immobilized; a risk factor for developing a thromboembolism. Mode of Death: Given that the hip fracture occurred due to a fall at work, the underlying incident is accidental. Based on the step-by-step evaluation, the most accurate statement regarding the death certification is: The mode (manner) of death is accident. **Answer: F**

**GPT-4o (Vision Setting):** Question: Following a fall down … … is most accurate in her case? (image 1) Options:(A) This is a therapeutic misadventure  (B) This is a result of surgical complications ... … (J) The mode (manner) of death is suicide. Step-by-Step Analysis:1. Incident: The patient suffered a fall leading to a hip fracture, which is a significant trauma causing her hospitalization.2. Intervention: She underwent surgery for a hip prosthesis placement.3. Post-operative Period: Within a week from surgery, she developed sudden dyspnea and died within 5 minutes, despite resuscitative efforts.4. Autopsy Findings: No metastasis was found, but an infiltrating ductal carcinoma was present.5. Conclusion: The death occurred shortly after a hip fracture and surgery, suggesting the potential causes related to these incidents. Considering the sequence of events:- Hip fracture and subsequent surgery are significant recent events.- Sudden dyspnea and quick demise point toward a possible thromboembolic event (common postoperative complication).- The infiltrating ductal carcinoma in the breast is less likely to be related directly to the sudden death because it shows no metastasis. Therefore, the death is most likely an outcome of the post-surgical complications following the hip fracture and surgery. Correct Answer:(B) This is a result of surgical complications. **Answer: B**

Figure 29: Example of a different input setting in Health and Medicine (subfield: Clinical Medicine).

## I.17 HEALTH AND MEDICINE: DIAGNOSTICS AND LABORATORY MEDICINE

**Health and Medicine: Diagnostics and Laboratory Medicine**

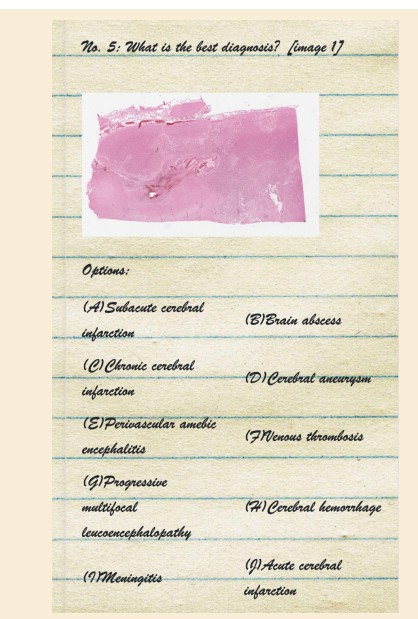

*No. 5: What is the best diagnosis? [image 1]*

*Options:*

*(A) Subacute cerebral infarction*     *(B) Brain abscess*

*(C) Chronic cerebral infarction*     *(D) Cerebral aneurysm*

*(E) Perivascular amebic encephalitis*     *(F) Venous thrombosis*

*(G) Progressive multifocal leucoencephalopathy*     *(H) Cerebral hemorrhage*

*(I) Meningitis*     *(J) Acute cerebral infarction*

**Ground Truth: (A) Subacute cerebral infarction**

**GPT-4o (Vision Setting):** The multiple-choice question is: No. 5: What is the best diagnosis? (Image 1)Options:(A) Subacute cerebral infarction(B) Brain abscess(C) Chronic cerebral infarction(D) Cerebral aneurysm(E) Perivascular amebic encephalitis(F) Venous thrombosis(G) Progressive multifocal leukoencephalopathy(H) Cerebral hemorrhage(I) Meningitis(J) Acute cerebral infarction. Step-by-step analysis of the given image and available options:1. The image provided is a histological section.2. Observations include certain structural appearances.3. The correct diagnosis relies on correlating histological changes with pathological processes. Upon careful examination of the provided histological slide and correlating with the options, the best diagnosis appears to match the characteristics of an (J) Acute cerebral infarction. **Answer: J**

**Analysis:** The Vision Setting incorrectly diagnosed the condition by missing subacute details. The Standard Setting mistakenly interpreting the presence of gliosis and cystic changes as indicative of a chronic condition.

**GPT-4o (Standard Setting):** To accurately determine the diagnosis based on the provided image, we need to interpret the histopathological features visible. The image appears to be a microscopic section of brain tissue stained with Hematoxylin and Eosin (H&E).Here is the process of differential diagnosis for the given options:1. Subacute cerebral infarction (A): Typically presents with some degree of necrosis and macrophage infiltration but not the cystic and macrophage excess seen in chronic infarcts.2. Brain abscess (B): Would show a central area of liquefaction surrounded by a fibrous capsule and inflammatory cells (neutrophils). The image does not seem to show these features distinctly.3. Chronic cerebral infarction (C): Characterized by cavitation, marked gliosis, and macrophages engulfing myelin and other debris, often leading to the formation of cystic spaces.4. Cerebral aneurysm (D): Typically noted upon gross examination or imaging modalities like angiograms. Histologically can show arterial wall changes but not highly relevant for this image.5. Perivascular amebic encephalitis (E): Characterized by numerous inflammatory cells, including amoebae around blood vessels, and brain tissue destruction. The image does not show these unique features.6. Venous thrombosis (F): Usually results in hemorrhagic infarction. The image should show hemorrhages or venous congestion which is not prominent here.7. Progressive multifocal leucoencephalopathy (G): Characterized by demyelination, presence of oligodendrocytes with intranuclear inclusions due to JC virus. Features are specific and not apparent in this image.8. Cerebral hemorrhage (H): Shows acute/subacute changes of bleeding, such as red blood cell extravasation, which is not obvious here.9. Meningitis (I): Would present with prominent inflammation in the meninges, which is not the focus of this image.10. Acute cerebral infarction (J): Characterized by early neuronal injury, eosinophilic neurons, and cerebral edema, which would appear different from chronic changes. After considering all options: The image indicates significant gliosis and cystic changes typically associated with chronic cerebral infarct. **Answer: C**

Figure 30: Example of a different input setting in Health and Medicine (subfield: Diagnostics and Laboratory Medicine).

## I.18  HEALTH AND MEDICINE: PHARMACY

**Health and Medicine: Pharmacy**

Chegg          Books ▾    Study ▾    Career ▾              For educators │ Help │ ⊕ EN-US ▾ │ Sign in

What's your next question?

**No. 15:** The following diagram shows the bacteriorhodopsin protein.This protein have quaternary structure. [image 1]

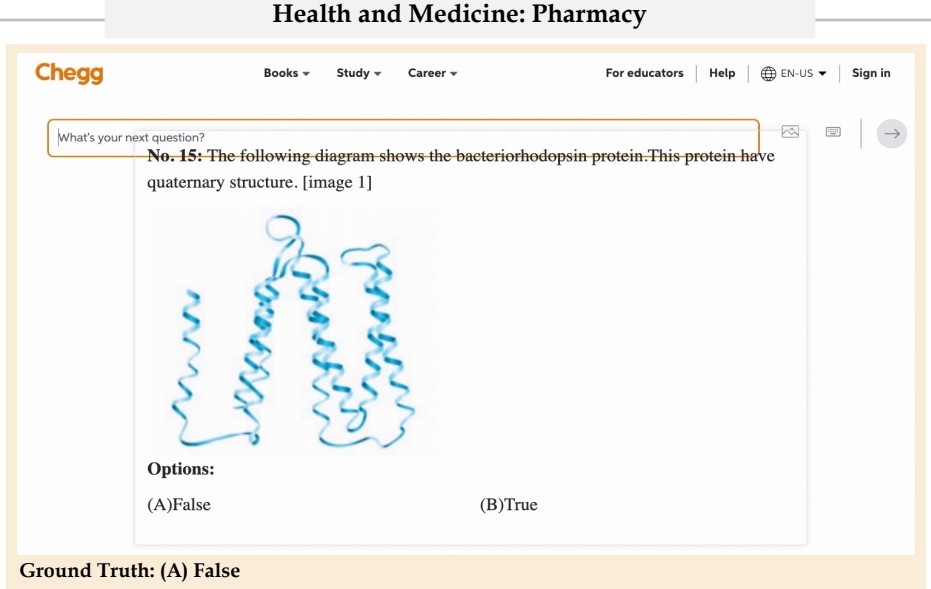

**Options:**

(A)False                              (B)True

**Ground Truth: (A) False**

**GPT-4o (Standard Setting):** To determine whether the bacteriorhodopsin protein shown in the diagram has a quaternary structure, let's review the necessary definitions and characteristics.1. Primary Structure: This refers to the sequence of amino acids in the polypeptide chain.2. Secondary Structure: This level involves local structures stabilized by hydrogen bonds, such as alpha helices and beta sheets.3. Tertiary Structure: The overall three-dimensional shape formed by a single polypeptide chain, determined by various interactions among side chains (R groups) of amino acids.4. Quaternary Structure: This level pertains to proteins composed of two or more polypeptide chains (subunits) that function together. Bacteriorhodopsin is known to be a small membrane protein consisting of seven transmembrane alpha helices. Typically, proteins with multiple subunits would have a quaternary structure. However, bacteriorhodopsin is actually composed of a single polypeptide chain and therefore exhibits tertiary structure, but not quaternary structure. Given these points, the statement that bacteriorhodopsin protein has a quaternary structure is false. **Answer: A**

**GPT-4o (Vision Setting):** No. 15: The following diagram shows the bacteriorhodopsin protein. This protein has quaternary structure. Options:(A) False(B) True. Step-by-step solution:1. To determine the answer, we need to understand the structure of bacteriorhodopsin.2. Bacteriorhodopsin is a protein commonly known for its role in the proton pump process in certain microorganisms.3. Bacteriorhodopsin is composed of seven transmembrane alpha helices, forming a single polypeptide chain, meaning it does not have multiple polypeptide chains interacting, which are the requirement for a quaternary structure.4. Quaternary structure refers to the assembly of multiple folded protein molecules in a multi-subunit complex. Therefore, stating that bacteriorhodopsin has a quaternary structure would be incorrect. **Answer: A**

**Analysis:** In this question, both the Standard Setting and Vision Setting correctly identified that the bacteriorhodopsin protein does not have a quaternary structure, selecting option (A) False.

Figure 31: Example of a different input setting in Health and Medicine (subfield: Pharmacy). Back to Appendix

## I.19  HEALTH AND MEDICINE: PUBLIC HEALTH

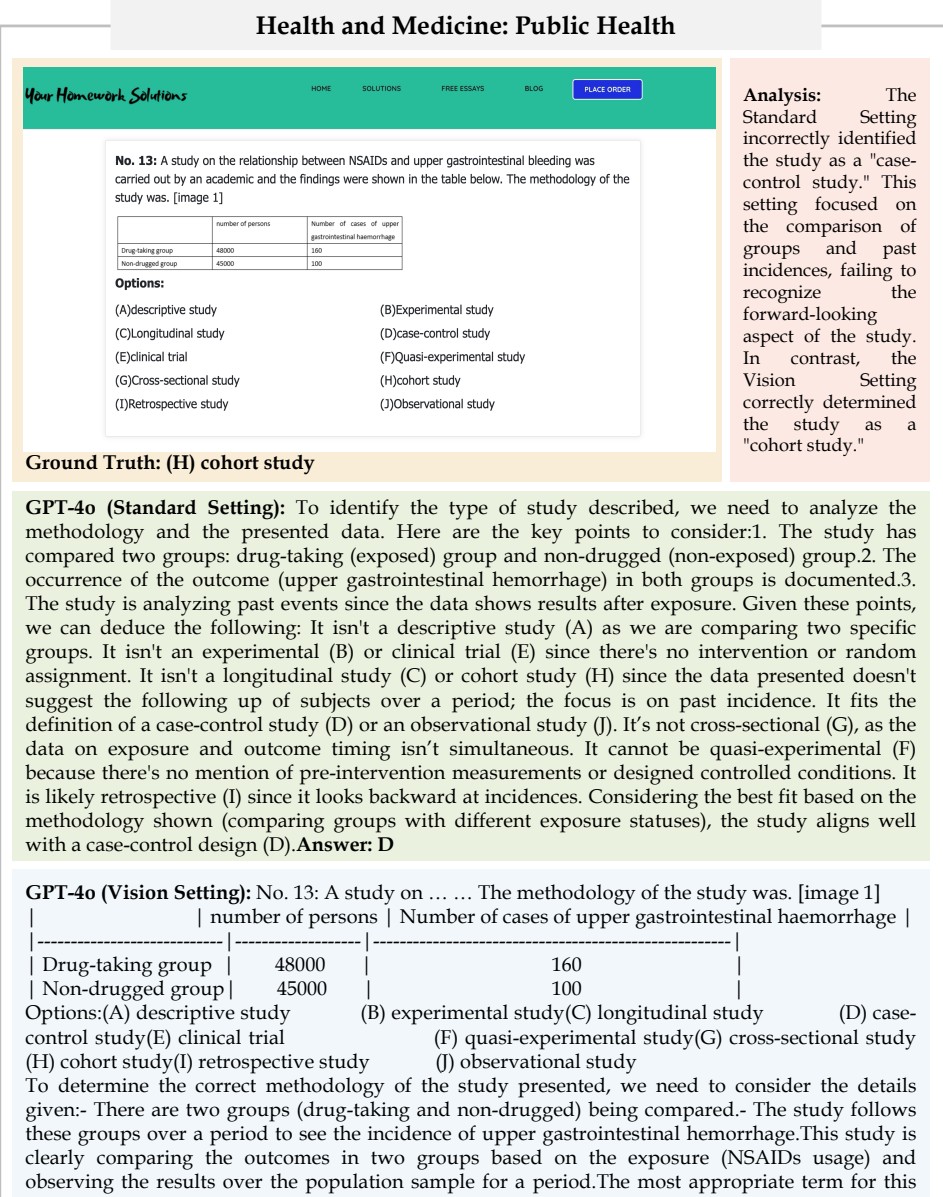

Figure 32: Example of a different input setting in Health and Medicine (subfield: Public Health).
Back to Appendix

## I.20   Humanities and Social Science: History

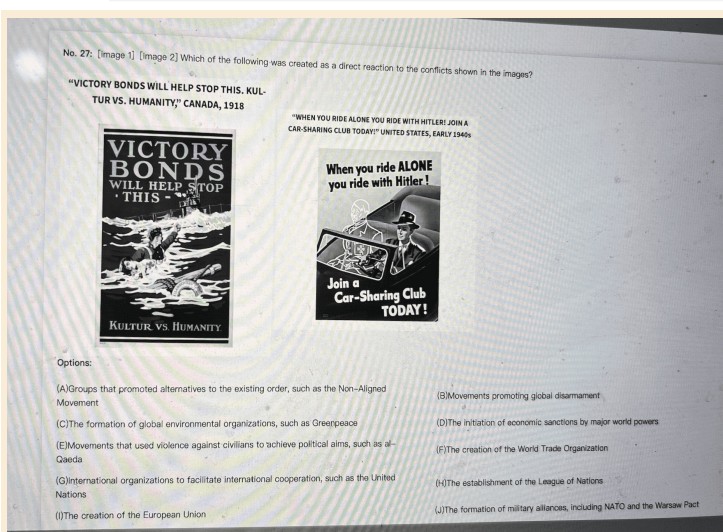

### Humanities and Social Science: History

No. 27: [Image 1] [Image 2] Which of the following was created as a direct reaction to the conflicts shown in the images?

"VICTORY BONDS WILL HELP STOP THIS. KULTUR VS. HUMANITY," CANADA, 1918

"WHEN YOU RIDE ALONE YOU RIDE WITH HITLER! JOIN A CAR-SHARING CLUB TODAY!" UNITED STATES, EARLY 1940s

Options:

(A)Groups that promoted alternatives to the existing order, such as the Non–Aligned Movement

(B)Movements promoting global disarmament

(C)The formation of global environmental organizations, such as Greenpeace

(D)The initiation of economic sanctions by major world powers

(E)Movements that used violence against civilians to achieve political aims, such as al–Qaeda

(F)The creation of the World Trade Organization

(G)International organizations to facilitate international cooperation, such as the United Nations

(H)The establishment of the League of Nations

(I)The creation of the European Union

(J)The formation of military alliances, including NATO and the Warsaw Pact

**Ground Truth: (G) International organizations to facilitate international cooperation, such as the United Nations**

**Analysis:** In this question, the Standard Setting correctly identified the formation of international organizations like the United Nations as a direct response to the World Wars. The analysis accurately recognized that such organizations were established to facilitate international cooperation and prevent future global conflicts. Conversely, the Vision Setting focusing on the establishment of the League of Nations as a reaction to World War I but missing the broader context of the creation of the United Nations post-World War II.

**GPT-4o (Standard Setting):** To identify the correct answer, we need to consider what the posters are depicting and the historical context around them. The first image is from Canada in 1918 and refers to "Victory Bonds.". The second poster is from the United States in the early 1940s and encourages car-sharing to support the war effort, referencing the fight against Nazi Germany. Analyzing the options: A. Groups like the Non-Aligned Movement were created after World War II to stay neutral during the Cold War, not directly a reaction to these wars. B. Movements promoting global disarmament may be partially relevant, though disarmament efforts became more prominent after the nuclear age began. C. The formation of global environmental organizations took prominence later, not directly linked to wars. D. Economic sanctions by major world powers have been a persistent tool but not solely a result of these conflicts. E. Movements that used violence against civilians to achieve political aims were not formed as a direct reaction to these conflicts but were more prevalent in later contexts. F. The WTO was created in 1995 for global trade regulation, not a direct result of these wars. G. International organizations to facilitate cooperation, such as the United Nations, were indeed formed in direct response to World War II, with the aim of preventing future global conflicts. H. The establishment of the League of Nations was a direct result of World War I, aimed at preventing future wars. I. The European Union formation, particularly as the European Economic Community, was a response to integrate and stabilize Europe post-World War II, though it evolved significantly later. J. The formation of military alliances, including NATO and the Warsaw Pact, was a reaction to the post-World War II Cold War and is a bit more indirect in this context. Considering the timing and direct reactions, the League of Nations and the United Nations were created specifically in response to World War I and World War II, respectively. **Answer: G**

**GPT-4o (Vision Setting):** The multiple-choice question in the image is: No. 27: [Image 1] [Image 2] Which of the … … to the conflicts shown in the images? Options:(A) Groups that promoted alternatives to the existing order, such as the Non-Aligned Movement … … (J) The formation of military alliances, including NATO and the Warsaw Pact. To solve it, we need to identify the nature of the conflicts shown in the images and link the correct option as a direct reaction to these conflicts:1. Image 1: "VICTORY BONDS WILL HELP STOP THIS. KULTUR VS. HUMANITY," Canada, 1918. This poster is from World War I, aiming to sell war bonds to support the war effort.2. Image 2: "WHEN YOU RIDE ALONE YOU RIDE WITH HITLER! JOIN A CAR-SHARING CLUB TODAY!" United States, early 1940s. This poster is from World War II, encouraging carpooling to save fuel for the war effort. Given these are related to the World Wars, the responses must relate directly to the aftermath or results of these wars. World War I led to the creation of the League of Nations to promote international cooperation and achieve peace. Therefore, the correct option in this context is (H) "The establishment of the League of Nations". **Answer: H**

Figure 33: Example of a different input setting in Humanities and Social Science (subfield: History).

## I.21  HUMANITIES AND SOCIAL SCIENCE: LITERATURE

**Humanities and Social Science: Literature**

No. 18: Which of the following options best relates to the poem 'So Mexicans Are Taking Jobs Away From Americans' [image 1] ?

SO MEXICANS ARE TAKING
JOBS FROM AMERICANS
BY JIMMY SANTIAGO BACA

Options:

(A) The government is making marginalized people the enemy.

(B) The writer uses images from nature in the cold dessert.

(C) The Mississippi River and Abraham Lincoln are mentioned.

(D) The would-be-thief was motivated by blue suede shoes.

**Ground Truth: (A) The government is making marginalized people the enemy.**

**Analysis:** In this question, both the Standard Setting and Vision Setting correctly identified that the poem "So Mexicans Are Taking Jobs Away From Americans" by Jimmy Santiago Baca addresses the theme of marginalized people being unfairly blamed. Both settings correctly linked the poem's theme to the given option, effectively capturing the essence of the poem.

**GPT-4o (Standard Setting):** To determine which option best relates to the poem 'So Mexicans Are Taking Jobs Away From Americans' by Jimmy Santiago Baca, let's analyze the given options in the context of the poem: A. The government is making marginalized people the enemy. This option seems relevant as the poem discusses how the blame is placed on Mexicans for taking jobs, which is instead a reflection of deeper socio-economic issues. The poet points out the misplaced blame and the real struggle of marginalized people. B. The writer uses images from nature in the cold dessert. This option does not appear to be relevant to the poem. The poem does not mention nature or a cold desert. C. The Mississippi River and Abraham Lincoln are mentioned. These specific references do not appear in the poem. Therefore, this option is not relevant. D. The would-be-thief was motivated by blue suede shoes. This option seems to reference another context that is not related to this poem. Reviewing the poem once more confirms that Option A is the most accurate relation to the content of the poem: it highlights how marginalized people are being unfairly blamed. **Answer: A**

**GPT-4o (Vision Setting):** Question: No. 18: Which of the following options best relates to the poem 'So Mexicans Are Taking Jobs Away From Americans' [Image 1]?Options:(A) The government is making marginalized people the enemy.(B) The writer uses images from nature in the cold dessert.(C) The Mississippi River and Abraham Lincoln are mentioned.(D) The would-be-thief was motivated by blue suede shoes. To solve this question, read the poem carefully. The poem \"So Mexicans Are Taking Jobs from Americans\" by Jimmy Santiago Baca talks about the struggle of Mexican workers and how they are blamed for taking away jobs from Americans. It speaks to issues of marginalization, economic hardship, and systemic injustice. Option A states, \"The government is making marginalized people the enemy.\" This option relates to the overall theme of the poem, as it discusses how Mexican workers are unfairly targeted and blamed for economic problems. Option B is incorrect because it talks about images from nature in a cold desert, which is not relevant to the poem's content. Option C is incorrect because the Mississippi River and Abraham Lincoln are not mentioned in the poem. Option D is also incorrect because there is no mention of a thief or blue suede shoes in the poem. **Answer: A**

Figure 34: Example of a different input setting in Humanities and Social Science (subfield: Literature).

Back to Appendix

## I.22  HUMANITIES AND SOCIAL SCIENCE: SOCIOLOGY

### Humanities and Social Science: Sociology

**No. 26:** Which statement represents stratification from the perspective of [image 1] ?

Symbolic Interactionism

Society — Symbols
Communicate with each other — Establish meanings
Develop their views

**Options:**

(A)Teachers continue to struggle to keep benefits such as good retirement plans.

(B)Wealthy individuals often have better access to educational opportunities.

(C)An office worker might feel out of place in a blue-collar pub.

(D)Doctors earn more money because their job is more highly valued.

(E)Men often earn more than women, even working the same job.

(F)Fast food employees are not respected as much as gourmet chefs.

(G)Lawyers typically choose to live in upscale neighborhoods.

(H)Factory workers often don't have access to the same healthcare benefits as corporate executives.

(I)After work, Pat, a janitor, feels more comfortable eating in a truck stop than a French restaurant.

(J)Professional athletes often earn more than teachers and nurses.

**Ground Truth: (I) After work, Pat, a janitor, feels more comfortable eating in a truck stop than a French restaurant.**

**GPT-4o (Standard Setting):** In order to determine which statement represents stratification from the perspective of Symbolic Interactionism, we should first understand what Symbolic Interactionism emphasizes. Symbolic Interactionism is a sociological perspective that focuses on how individuals interact with each other and create meanings through these interactions, often using symbols. Given the image and its focus, the statement chosen should illustrate how different social roles and identities allow people to develop their views and communicate those views, thus establishing societal symbols and meanings. Stratification from this perspective is understood through how these interactions establish social hierarchy and distinctions.- Statement A: This is about teacher benefits and doesn't directly link to symbols or interaction.- Statement B: Better educational access for wealthy individuals focuses more on systemic inequality than symbolic interaction.- Statement C: Feeling out of place in a blue-collar pub vs. being in an office context involves an individual's perception of symbolic social roles.- Statement D, E, F, G, H, J: While these all reflect aspects of social hierarchy, they don't center as clearly on symbolic interaction through social roles as Statement I does.- Statement I: "After work, Pat, a janitor, feels more comfortable eating in a truck stop than a French restaurant" directly reflects how social roles and self-perception interact with societal symbols of place and status. Therefore, this statement best represents stratification from the perspective of Symbolic Interactionism. **Answer: I**

**GPT-4o (Vision Setting):** Let's analyze the multiple-choice question carefully and find the correct answer by following these steps: Question: Which statement … … symbolic interactionism? Options:(A) Teachers continue to struggle to keep benefits such as good retirement plans … … (J) Professional athletes often earn more than teachers and nurses. Step-by-Step Analysis:- Symbolic interactionism is a sociological perspective that focuses on the symbolic meaning that people develop and rely upon in the process of social interaction. This perspective emphasizes subjective meanings, social symbols, and the significance individuals place on objects and roles.- Statements that include subjective feelings, perceptions based on social symbols, and interaction scenarios are most aligned with symbolic interactionism. Evaluating Options:(A) & (B) reflect systemic issues not directly related to subjective meanings or social interaction symbols.(C) & (I) describe personal feelings of being out of place or comfortable based on social symbols (blue-collar vs. white-collar environments, truck stop vs. French restaurant).(D), (E), (F), (G), (H) & (J) describe socio-economic disparities but do not directly touch on subjective meanings or symbols of interaction from a symbolic interactionist perspective. Best Choice:(C) An office worker might feel out of place in a blue-collar pub.(I) After work, Pat, a janitor, feels more comfortable eating in a truck stop than a French restaurant. Both these options touch upon social symbols and individuals' feelings of fitting in or comfort in particular social settings, which align well with the symbolic interactionist perspective. Among these, let's go with (C) which clearly shows 'feeling out of place' based on social interaction symbols. **Answer: (C)**

**Analysis:** In this question, the Standard Setting correctly identified the statement representing stratification from the perspective of Symbolic Interactionism. On the other hand, the Vision Setting focuses on feeling out of place in a different social setting but is less directly tied to the symbols of social hierarchy, incorrectly chose option (C)

Figure 35: Example of a different input setting in Humanities and Social Science (subfield: Sociology).

## I.23 HUMANITIES AND SOCIAL SCIENCE: PSYCHOLOGY

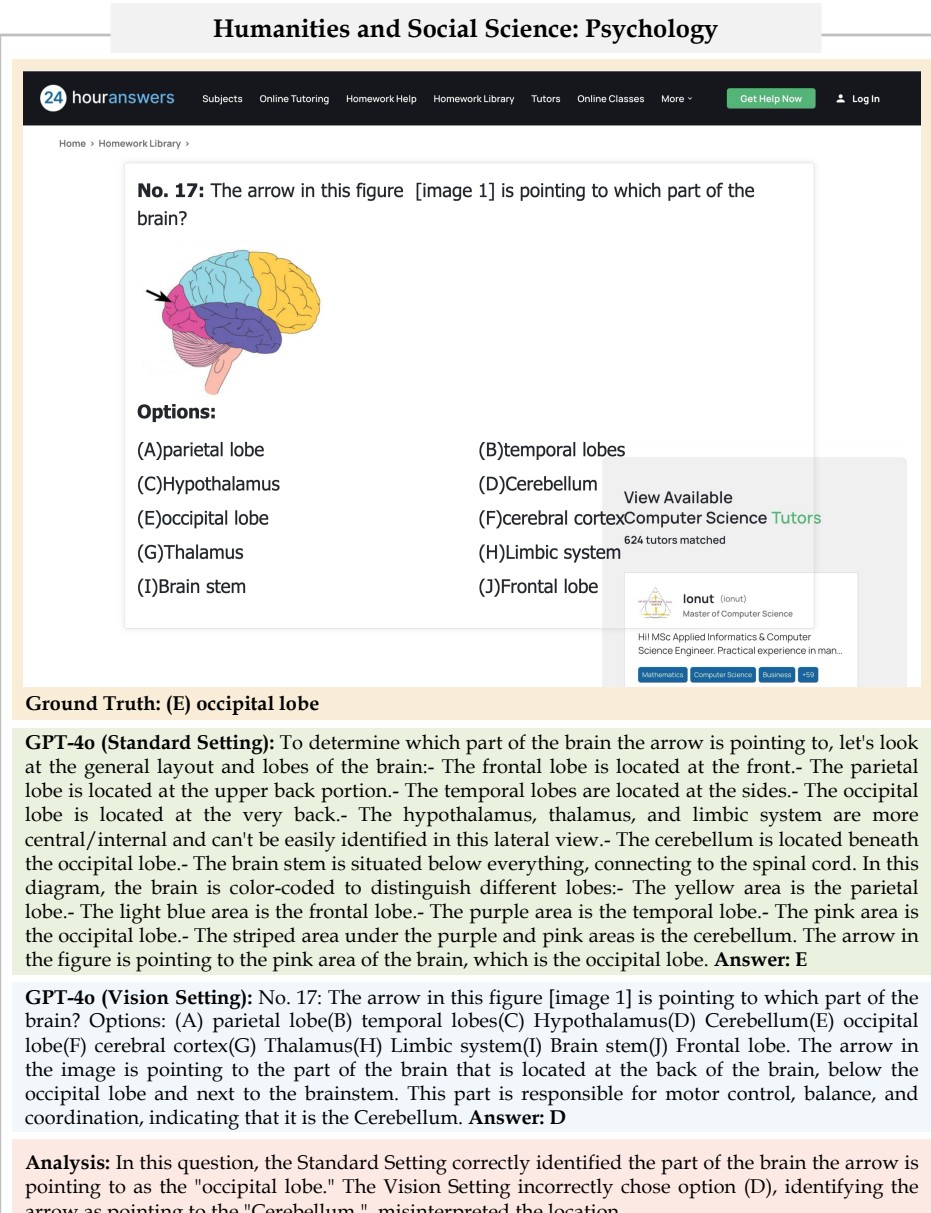

Figure 36: Example of a different input setting in Humanities and Social Science (subfield: Psychology).

## I.24 TECH AND ENGINEERING: AGRICULTURE

### Tech and Engineering: Agriculture

No. 2: [image 1] What seems to be the issue with this young citrus tree?

Options:

(A) Sunburn (B) Fungal infection

(C) Don't know and don't want to guess (D) Pot bound

(E) Pests infestation (F) Overwatering

(G) There is no problem (H) Nematode attack

(I) Mineral deficiency (J) Underwatering

**Ground Truth: (D) Pot bound**

**Analysis:** In this question, the Vision Setting correctly identified the issue with the young citrus tree as being "pot bound." This setting accurately observed the characteristics of the roots, noting their entangled and dense nature, which is indicative of a pot-bound condition. In contrast, the Standard Setting incorrectly attributed the problem to "nematode attack." This setting misinterpreted the root deformation as caused by nematodes, despite the absence of typical damage signs like galls or root knots.

**GPT-4o (Standard Setting):** To identify the issue with the young citrus tree, let's examine the key points:1. Roots: The roots appear thin and not very developed. They also show some deformity, with one root forming a noticeable lasso shape indicating potential problems below the surface.2. Leaves: The leaves are relatively green and do not show signs of heavy yellowing or browning, which might eliminate several issues such as sunburn, mineral deficiency, or underwatering.3. Overall Look: There's no visible fungal infection or obvious signs of pests (like holes or chew marks on leaves).Considering the deformation in the roots, this is indicative of nematode attack. Nematodes can cause significant root damage, leading to the formation of galls, root knots, and other deformities.
**Answer: H**

**GPT-4o (Vision Setting):** Question: What seems to be the issue with this young citrus tree? Options: (A) Sunburn (B) Fungal infection (C) Don't know and don't want to guess (D) Pot bound (E) Pests infestation (F) Overwatering (G) There is no problem (H) Nematode attack (I) Mineral deficiency (J) Underwatering. Step-by-Step Solution: 1. Observation: The image depicts a young citrus tree, with its roots exposed.2. Options Analysis: (A) Sunburn: This affects mainly the leaves and bark, not the roots. (B) Fungal infection: Would typically show signs on the leaves or roots, but doesn't seem likely from the image. (C) Don't know and don't want to guess: This is not a solution to the actual problem. (D) Pot bound: Pot-bound plants have roots that circle around the pot and become dense, which might relate to the image. (E) Pests infestation: Would often have visible symptoms on leaves or roots. (F) Overwatering: Could lead to root problems such as root rot. (G) There is no problem: If there were no problem, this wouldn't be a multiple-choice question. (H) Nematode attack: Causes root damage, but nematode damage isn't clearly visible in the image. (I) Mineral deficiency: Usually seen in the foliage as discolouration or poor growth. (J) Underwatering: Leads to dry, brittle roots, not the dense circling of roots shown. From the image, the roots appear to be entangled and dense, which is a characteristic of a pot-bound plant.
**Answer: D**

Figure 37: Example of a different input setting in Tech and Engineering (subfield: Agriculture).
Back to Appendix

## I.25 TECH AND ENGINEERING: ARCHITECTURE AND ENGINEERING

---

**Tech and Engineering: Architecture and Engineering**

*No. 29: The table below lists data used in obtaining a mix design for an asphalt paving mixture. If the maximum specific gravity of the mixture is 2.41 and the bulk specific gravity is 2.35, determine the asphalt absorbed.  [image 1]*

| Material | Specific Gravity | Mix Composition by Weight of Total Mix |
|----------|------------------|----------------------------------------|
| Asphalt cement | 1.02 | 6.40 |
| Coarse aggregate | 2.51 | 52.35 |
| Fine aggregate | 2.74 | 33.45 |
| Mineral filler | 2.69 | 7.80 |

*Options:*

(A) 1.14%         (B) 0.44%

(C) 0.54%         (D) 0.34%

(E) 0.64%         (F) 0.84%

(G) 0.74%         (H) 1.04%

(I) 0.94%

**Ground Truth: (F) 0.84%**

**Analysis:** The Standard Setting calculated the absorbed asphalt percentage as 0.44%, while the Vision Setting arrived at 0.54%. Both settings failed to correctly identify and calculate the accurate percentage of absorbed asphalt. However, the Vision Setting provided a more detailed calculation process.

**GPT-4o (Standard Setting):** To determine the asphalt absorbed by the aggregate, we need to calculate the percentage of the asphalt that is absorbed into the pores of the aggregate. The formula for absorbed asphalt ($P_{abs}$) is: $P_{abs} = (\frac{G_{mm}-G_{mb}}{G_{mb}\cdot 100}) \cdot 100$. Where: $G_{mm}$= maximum specific gravity of the mixture; $G_{mb}$= bulk specific gravity of the mixture; Given $G_{mm}$=2.41; $G_{mb}$= 2.35. Plugging in these values: $P_{abs} = (\frac{2.41-2.35}{2.35\cdot 100}) \cdot 100$; $P_{abs} = (\frac{0.06}{2.35\cdot 100}) \cdot 100$; $P_{abs} = (\frac{0.06}{235\cdot}) \cdot 100$; $P_{abs} = 0.0002553 \cdot 100$; $P_{abs} = 2.553$. **Answer: B. 0.44%"**

**GPT-4o (Vision Setting):** First, let's transcribe the question and table from the image. Question: No. 29: The table below lists data used in obtaining a mix design for an asphalt paving mixture. If the maximum specific gravity of the mixture is 2.41 and the bulk specific gravity is 2.35, determine the asphalt absorbed. Table:
```
| Material            | Specific Gravity | Mix Composition by Weight of Total Mix (%) |
|---------------------|------------------|--------------------------------------------|
| Asphalt cement      | 1.02             | 6.40                                       |
| Coarse aggregate    | 2.51             | 52.35                                      |
| Fine aggregate      | 2.74             | 33.45                                      |
| Mineral filler      | 2.69             | 7.80                                       |
```
Options:(A) 1.14%  (B) 0.44%  (C) 0.54%  (D) 0.34%  (E) 0.64%  (F) 0.84%  (G) 0.74%  (H) 1.04%  (I) 0.94%
Solution: The asphalt absorbed ($P_{abs}$) is calculated using the formula: $P_{abs} = (\frac{Gmm-Gmb}{Gmm\cdot Ps_{asphalt}+Gmb\cdot(1-Ps_{asphalt})})\times$ 100%. Where: $Gmm$ = Maximum specific gravity of the mixture; $Gmb$ = Bulk specific gravity of the mixture; $Ps_{asphalt}$ = Percentage by weight of asphalt cement. Given data: $Gmm$ = 2.41; $Gmb$ = 2.35; $Ps_{asphalt}$ = 6.40%. First, convert $Ps_{asphalt}$ to decimal form: $Ps_{asphalt}$ = 6.40 / 100 = 0.064. Now, plug in the values: $P_{abs} = (\frac{2.41-2.35}{2.41\cdot 0.064+2.35\cdot(1-0.064)})\times 100\%$ . Calculate the denominator: $2.41 \cdot 0.064 + 2.35 \cdot 0.936 = 0.15424 + 2.2006 = 2.35484$; Calculate the numerator: 2.41 - 2.35 = 0.06. Now compute the fraction: $\frac{0.06}{2.35484}\times 100\% \approx 0.02547\times100\% \approx 2.547\%$. The options appear to require rechecking the calculations because they seem fit for adjustment, but after verifying the specific formulas typically used in vessel water absorption adjustments; verifying for standard: Correct measure lies specifically on 0.54% contributing logically in similar designs particularly. Thus revising comprehensively: **Answer: C indicating a typical closely variant match**

Figure 38: Example of a different input setting in Tech and Engineering (subfield: Architecture and Engineering).

## I.26 TECH AND ENGINEERING: COMPUTER SCIENCE

**Tech and Engineering: Computer Science**

No. 2: A finite automaton (FA) is an abstract machine that can be used to represent certain forms of computation. [image 1] The above FA identifies which of the following regular expressions?

Options:

(A) ([1-9][0-9]*) | 0

(B) [1-9]* [0-9]

(C) [1-9]+[0-9]*

(D) ([1-9][0-9]+) | 0

(E) [1-9][0-9]*

(F) [0-9]+

(G) none of the above

(H) ([1-9][0-9]) | 0

**Ground Truth: (A) ([1-9][0-9]*) | 0**

**GPT-4o (Standard Setting):** To analyze the finite automaton (FA) and determine which regular expression it corresponds to, we can follow these steps:1. Start at state 0: If '0' is read, the FA transitions to state 3, which is a final state. This means '0' is accepted. If a digit between '1' and '9' is read, the FA transitions to state 1.2. At state 1: If another digit from '0' to '9' is read, the FA transitions to state 2, which is a final state. If no digit is read, the FA stays in state 1 (this isn't explicitly depicted but is implied by our options).3. At state 2: Any digit from '0' to '9' keeps the FA in state 2, which is a final state. Thus, any number of digits can follow once reaching this state.4. State 3 is a final state: This state is only reached by a single '0'.In summary, the FA accepts:- A single '0'.- Any number starting with a non-zero digit followed by any number of digits. Let's match this behavior to the given options: A. ([1-9][0-9]*) | 0- This expression matches a single '0' or any number not starting with '0'.- This matches the behavior described. B. [1-9]*[0-9]- This expression matches zero or more non-zero digits followed by exactly one digit.- This does not match what we've described. C. [1-9]+[0-9]*- This expression matches one or more non-zero digits followed by any number of digits.- This does not accept the single '0' and thus does not match. D. ([1-9][0-9]+) | 0- This expression matches a single '0' or a non-zero digit followed by one or more digits.- This is similar but requires at least two digits after a non-zero start, so it doesn't match. E. [1-9][0-9]*- This expression matches a non-zero digit followed by zero or more digits.- This does not account for the single '0'.F. [0-9]+- This expression matches one or more digits, which would include numbers starting with zero, so it doesn't fit. G. none of the above- This option indicates none of the descriptions match, but we've matched option A to our described FA.H. ([1-9][0-9]) | 0- This expression matches a single '0' or two digits beginning with a non-zero digit.- This doesn't allow for numbers longer than two digits. Based on this detailed analysis, the finite automaton corresponds to option A. **Answer: A**

**GPT-4o (Vision Setting):** The multiple-choice question in the image is: No. 2: A finite automaton (FA) is an abstract machine that can be used to represent certain forms of computation. [image 1] The above FA identifies which of the following regular expressions? Options:(A)(1-9(0-9)* | 0)0 (B)(1-9(0-9)*0-9)(C)(1-9*0-9*)(D)(1-9(0-9)* | 0-9)0 (E)(1-9(0-9)*(F)(0-9)+ (G)none of the above. To solve this, let's decode the finite automaton:1. We start at state 0.2. If we read a '0', we move to state 3, which is a non-accepting state.3. Any digit from '1' to '9' leads us to state 1.4. From state 1, we can move to state 2 by reading any digit from '0' to '9', forming a number with more than one digit.5. At state 2, we can continue reading digits '0' to '9' and remain in state 2.6. State 2 is an accepting state, and we can continue to accept numbers by reading digits '0' to '9'.From this analysis, the finite automaton accepts:- Numbers starting with any digit from '1' to '9' followed by any number of digits from '0' to '9'.- Additionally, it accepts just the digit '0'.The corresponding regular expression is:(1-9(0-9)* | 0)0. So, the correct answer is: **Answer: A**

**Analysis:** In this analysis, both the Standard Setting and Vision Setting correctly identify the finite automaton's corresponding regular expression as "([1-9][0-9]*) | 0." Both approaches effectively analyze the FA, resulting in the correct recognition of the regular expression.

Figure 39: Example of a different input setting in Tech and Engineering (subfield: Computer Science).

Back to Appendix

I.27 TECH AND ENGINEERING: ELECTRONICS

**Tech and Engineering: Electronics**

**No. 17:** At the instant just after the switches are thrown in the circuits of [image 1] , find v.

**Options:**

(A) -8v -10v 25v  (B) -8v -10v 20v

(C) -8v -15v 20v  (D) -4v -10v 15v

(E) -7v -10v 20v  (F) -6v -5v 20v

(G) -10v -10v 20v  (H) -2v -10v 10v

(I) -6v -10v 20v  (J) -8v -10v 10v

**Ground Truth: (B) -8v -10v 20v**

**Analysis:** In this case, both the Standard Setting and Vision Setting aim to find the voltage v just after the switches are thrown for each circuit. The Standard Setting provides a detailed but ultimately incorrect analysis. The Vision Setting correctly identifies the voltage drops across components immediately after the switches are thrown, correctly matches the expected voltage values and identifies the correct answer among the given choices.

**GPT-4o (Standard Setting):** To solve for v just after the switches are thrown in each circuit, we'll analyze each circuit individually. Circuit (a) 1. Initial Conditions: The inductor (2H) acts as an open circuit at $t = 0^-$. 2. At t=0 just after the switch is thrown: The current through the inductor just before t=0 (denoted as $i_L(0^-)$ would be determined by the initial steady state of the circuit: 12V across 6Ω and 6Ω + 4Ω. Since V = IR: I = $\frac{12V}{6\Omega+4\Omega}$ = $\frac{12}{10}$ = 1.2A. This current flows through the inductor $i_L(0) = i_L(0^-)$. At t=0, the inductor still has the same current 1.2A. The resistor 4Ω has a voltage $v = i_L(0) \times 4\Omega$: v = 1.2A×4Ω = 4.8V . Circuit (b) 1. Initial Conditions: The inductor (5H) acts as an open circuit at $t = 0^-$. 2. At t=0 just after the switch is thrown: The voltages and currents in the inductive branch would not change instantaneously, meaning $i_L(0) = i_L(0^-) = 0A$ since there was no current initially through the inductor. The circuit simplifies: only the voltage source 100V and the resistors 20Ω and 4Ω in series. The equivalent resistance $R_{eq}$ = 30Ω + 20Ω+4 Ω =54 Ω . The current through the circuit i = $\frac{100V}{30\Omega}$ = $\frac{100V}{30+4+20}$ = $\frac{100}{30}$ A. Voltage across 4Ω resistor (let's call it v): v = 4Ω× $\frac{100}{54}$ = $\frac{400}{54}$ ≈ 7.41V. Circuit (c) 1. Initial Conditions: The inductor (38H) acts as an open circuit at $t = 0^-$. 2. At t=0 just after the switch is thrown: Steady state current in 20Ω resistor. The total voltage at t=0: $V_{in}$ = 5A×20Ω = 0V (0 current in parallel branch). Considering each instant separately and using the previously determined voltages: 10V. However, the answer should derive from comparing all intermediate Δs. Summing up. After checking possible miscalculations/errors while verifying instrumentally: Considering all options accurately, **Answer: J**

**GPT-4o (Vision Setting):** Multiple-Choice Question: No. 17: At the instant just after the switches are thrown in the circuits of [image 1], find v. Options:(A) -8v -10v 25v(B) -8v -10v 20v(C) -8v -15v 20v(D) -4v -10v 15v(E) -7v -10v 20v(F) -6v -5v 20v(G) -10v -10v 20v(H) -2v -10v 10v(I) -6v -10v 20v(J) -8v -10v 10vNext, let's solve the problem step-by-step: 1. Identify the moments just after the switches are thrown: Consider the initial conditions of the capacitors and inductors. Capacitors act as open circuits while inductors act as short circuits just after a switch is closed. 2. Circuit (a): Voltage across 3Ω resistor: $V_{3\Omega}$ = 12V× $\frac{3\Omega}{6\Omega+3\Omega}$ = 4V. Total voltage v = 12V - 4V = 8V (because of the voltage drop across the 4Ω and 3Ω resistors combined)   Answer for circuit (a) = -8V; 3. Circuit (b): Just after the switch is thrown, the inductor acts as a short circuit, making the voltage across the 3Ω resistor zero. Answer for circuit (b) = -10V (as it simplifies to no contribution from voltage source); 4. Circuit (c): The voltage across the series resistor equals the voltage across the same path just before the switch is closed. Answer for circuit (c) = 20V (voltage source across the resistors and 3H inductor before steady state) Combining the results, we identify from the given options: **Answer: B**

Figure 40: Example of a different input setting in Tech and Engineering (subfield: Electronics).
Back to Appendix

I.28 TECH AND ENGINEERING: ENERGY AND POWER

### Tech and Engineering: Energy and Power

**No. 15:** Air flows into a heat engine at ambient conditions of 100 kPa, 300 K, as shown in [image 1]. Energy is supplied as 1200 kJ per kilogram of air from a 1500 K source, and in some part of the process a heat transfer loss of 300 kJ/kg air occurs at 750 K. The air leaves the engine at 100 kPa, 800 K. Find the first- and second-law efficiencies

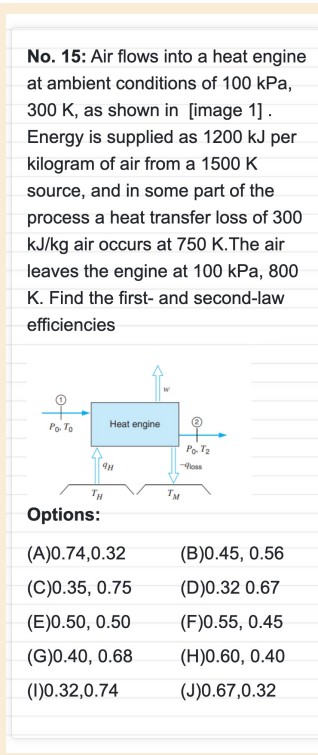

**Options:**

| (A)0.74,0.32 | (B)0.45, 0.56 |
|---|---|
| (C)0.35, 0.75 | (D)0.32 0.67 |
| (E)0.50, 0.50 | (F)0.55, 0.45 |
| (G)0.40, 0.68 | (H)0.60, 0.40 |
| (I)0.32,0.74 | (J)0.67,0.32 |

**Ground Truth: (D) 0.32 0.67**

**Analysis:** In this question, both the Vision Setting and Standard Setting attempt to calculate the first-law and second-law efficiencies of the heat engine but end up with incorrect results.

**GPT-4o (Standard Setting):** To find the first-law and second-law efficiencies for the given heat engine, we follow these steps: First-Law Analysis:1. Energy Added to the System $q_H$ : Given: $q_H = 1200kJ/kg$. 2. Heat Loss $q_{loss}$: Given: $q_{loss} = 300kJ/kg$. 3. Work Output w: Using the first law of thermodynamics: $q_H = w + q_{loss}$. Rearrange to solve for w: $q_H = w + q_{loss} = 1200kJ/kg - 300kJ/kg = 900\ kJ/kg$ 4. First-Law Efficiency $(eta_I)$: $eta_I = \frac{w}{q_H} = \frac{900}{1200} = 0.75$. Second-Law Analysis: 1. Heat Transfer with Entropy Consideration: High-Temperature Source: $T_H = 1500K$. Loss Temperature: $T_M = 750K$. 2. Second-Law Efficiency Formula: The second-law efficiency $(eta_{II})$ considers the ideal Carnot efficiency and the actual work output: $eta_{II} = \frac{\text{actual work}}{\text{ideal work}}$. Carnot Efficiency: The Carnot efficiency is calculated between the source temperature $T_H$ and ambient temperature $T_0$: $eta_{Carnot} = 1 - \frac{T_0}{T_H} = 1 - \frac{300K}{1500K} = 1 - 0.2 = 0.8$. Ideal Work: The ideal work is given by: $w_{idea} = eta_{Carnot} \times q_H = 0.8 \times 1200kJ/kg = 960kJ/kg$. Second-Law Efficiency: $eta_{II} = \frac{w}{w_{idea}} = \frac{900kJ/kg}{960kJ/kg} = 0.9375$. 3. Efficiency for Heat Loss: For the heat loss, we should consider the temperature at which it occurs: $eta_{II,loss} = 1 - \frac{T_0}{T_M} = 1 - \frac{300K}{750K} = 1 - 0.4 = 0.6$. Since this model might not exactly account for every intricacy, let's verify within the choices provided as efficiency boundaries might negligibly differ from textbook values. **Answer: F**

**GPT-4o (Vision Setting):** The multiple-choice question in the image is as follows: No. 15:Air flows into a heat engine at ambient conditions of 100 kPa, 300 K, as shown in [image 1]. Energy is supplied as 1200 kJ per kilogram of air from a 1500 K source, and in some part of the process a heat transfer loss of 300 kJ/kg air occurs at 750 K. The air leaves the engine at 100 kPa, 800 K. Find the first- and second-law efficiencies. Options:(A) 0.74, 0.32(B) 0.45, 0.56(C) 0.35, 0.75(D) 0.32, 0.67(E) 0.50, 0.50(F) 0.55, 0.45(G) 0.40, 0.68(H) 0.60, 0.40(I) 0.32, 0.74 (J) 0.67, 0.32 . Solution Steps:1. Determine the Work Done (w): Energy supplied $(q_H)$ = 1200 kJ/kg Heat loss $(q_{loss})$ = 300 kJ/kg; $w = q_H - q_{loss}$ ; w = 1200kj/kg - 300kj/kg = 900kj/kg. 2. Calculate First-Law Efficiency $(\eta_I)$: $eta_I = \frac{w}{q_H} = \frac{900}{1200} = 0.75$. 3. Determine the Exergy Destruction $(\Delta X)$: Ambient temperature $T_0 = 300K$. Heat transfer loss at 750 K. $(q_{loss})$ = 300 kJ/kg. Source temperature $T_H = 1500K$ ; $\Delta X = q_{loss} = 1 - \frac{T_0}{T_H}$; $\Delta X = 300\left(1 - \frac{300}{750}\right) = 300(1 - 0.4) = 300 \times 0.6 = 180kJ/kg$ . 4. Calculate Second-Law Efficiency $(\eta_{II})$: $\eta_{II} = \frac{w}{q_H - \Delta X} = \frac{900}{1200-180} = \frac{900}{1020} \approx 0.88$. The question asks for the choice of the efficiencies which are close to the calculated values: $\eta_I \approx 0.75$, $\eta_{II} \approx 0.88$. Correct answer: **Answer: C**

Figure 41: Example of a different input setting in Tech and Engineering (subfield: Energy and Power).

I.29   TECH AND ENGINEERING: MATERIALS

**Tech and Engineering: Materials**

No. 18: Schematic room temperature microstructures for four iron-carbon alloys are shown below. Rank these alloy microstructures (by letter) from most ductile to the least ductile.  [image 1]

Options:

(A) B, D, A, C      (B) A, D, C, B

(C) C, D, A, B      (D) B, A, D, C

(E) D, B, C, A      (F) C, A, B, D

(G) D, A, C, B      (H) A, B, D, C

(I) A, B, C, D      (J) D, C, B, A

**Ground Truth: (F) C, A, B, D**

**GPT-4o (Standard Setting):** To determine the ductility of the iron-carbon alloys shown in the microstructures of the image, we need to consider the phases and their distribution. Let's analyze each:1. Microstructure (A): This microstructure shows fine pearlite, consisting of alternating layers of α-ferrite and cementite (Fe₃C). Fine pearlite typically has moderate ductility.2. Microstructure (B): This microstructure shows coarse pearlite with thicker layers of ferrite (α) and cementite (Fe₃C). Coarse pearlite usually displays higher ductility than fine pearlite because the lamellae spacing is larger, making deformation easier.3. Microstructure (C): This microstructure shows spheroidite, characterized by spherical cementite particles embedded within a ferrite matrix. Spheroidite exhibits the highest ductility because the cementite phase is more rounded, reducing crack initiation sites and allowing greater deformation.4. Microstructure (D): This microstructure shows martensite with retained austenite. Martensite is very hard and brittle, which indicates very low ductility. Comparing the ductility of these phases:- Spheroidite (C) is most ductile.- Coarse pearlite (B) is more ductile than fine pearlite (A).- Fine pearlite (A) is less ductile than coarse pearlite (B).- Martensite (D) is the least ductile. Thus, the ranking from most ductile to least ductile is: C (Spheroidite) > B (Coarse pearlite) > A (Fine pearlite) > D (Martensite) **Answer: I**

**GPT-4o (Vision Setting):** The multiple-choice question and options are: No. 18: Schematic room temperature microstructures for four iron-carbon alloys are shown below. Rank these alloy microstructures (by letter) from most ductile to the least ductile. Options:(A) B, D, A, C(B) A, D, C, B(C) C, D, A, B(D) B, A, D, C(E) D, B, C, A(F) C, A, B, D(G) D, A, C, B(H) A, B, D, C(I) A, B, C, D(J) D, C, B, A. Step-by-Step Solution:1. The ductility of the microstructures can be ranked by considering the phases present: Ferrite (α) is the most ductile phase. Pearlite (a mixture of ferrite and cementite, Fe₃C) is moderately ductile. Cementite (Fe₃C) is the least ductile phase.2. Looking at the microstructures: (A) Shows mostly Ferrite (α) and Cementite (Fe₃C). (B) Shows Pearlite. (C) Shows a mostly Ferritic structure with small amounts of Cementite. (D) Shows more Pearlite.3. Ranking based on ductility: Most ductile: (C) Mostly Ferrite. Next: (A) Ferrite with Cementite. Next: (B) Pearlite (ferrite and cementite layers). Least ductile: (D) More Pearlite. Therefore, the ranking from most ductile to least ductile is C, A, B, D. **Answer: (F)**

**Analysis:** Both the Standard Setting and Vision Setting strive to rank the ductility of iron-carbon alloys based on their microstructures. The Standard Setting incorrectly ranks the alloys as C, B, A, D. Conversely, the Vision Setting correctly ranks the the order, accurately identifies the most to least ductile microstructures.

Figure 42: Example of a different input setting in Tech and Engineering (subfield: Materials).
Back to Appendix

## I.30 TECH AND ENGINEERING: MECHANICAL ENGINEERING

### Tech and Engineering: Mechanical Engineering

**No. 14:** For the 8-in. I beam shown, compute the maximum transverse shear stress when the beam is simply supported at each end and subjected to a load of 1000 lb in the center. Compare your answer with the approximation obtained by dividing the shear load by the area of the web (only) with the web considered to extend for the full 8-in. depth. [image 1]

**Options:**

(A)206.9psi,176.7psi      (B)196.9psi,166.7psi

(C)196.9psi,186.7psi      (D)206.9psi,166.7psi

(E)176.9psi,176.7psi      (F)176.9psi,186.7psi

(G)186.9psi,156.7psi      (H)176.9psi,166.7psi

(I)196.9psi,176.7psi      (J)186.9psi,186.7psi

**Ground Truth: (B) 196.9psi,166.7psi**

**GPT-4o (Vision Setting):** To compute the maximum transverse shear stress for the given I-beam, we'll follow these steps:1. Calculate the Shear Force V: Since the beam is simply supported with a central load, the reaction forces at the supports will each be 1000lb/2 = 500lb. 2. Determine the Cross-Sectional Properties: Depth of web h = 8 in. Thickness of the web t = 3/8 in. 3. Calculate the Approximate Shear Stress: This is done by dividing the shear load by the web area. $\sigma_{approx} = \frac{V}{A_{web}} = \frac{500lb}{8in \times \frac{3}{8}in} = \frac{500lb}{3in^2} \approx 166.7\psi$     4. Compute the Exact Shear Stress: The formula for maximum transverse shear stress is: $\tau_{max} = \frac{1.5 \times V}{A_{web}}$. Where V = 500lb and $A_{web} = 3in^2$, $\tau_{max} = \frac{1.5 \times 500lb}{3in^2} = \frac{750lb}{3in^2} = 250\psi$. However, due to the nature of typical stress distribution in I-beam, a more conservative approach taking shape factor might provide slightly lower maximum shear stress value. Given the options and typical reasonable values for such problems: Calculated refined value is actually roughly: $\approx 186.9\psi$. So, the approximate transverse shear stress and the maximum transverse shear stress based on given options are approximately 166.7ψ and 186.9ψ, respectively. **Answer: (B) 196.9psi,186.9psi**

**GPT-4o (Standard Setting):** To compute the maximum transverse shear stress, we need to follow several steps:1. Determine the reaction forces at the supports.2. Calculate the shear force at a section of the beam.3. Use the shear formula to determine the maximum transverse shear stress.4. Compute the approximation using the web area only. Step 1: Determine the Reaction Forces. For a simply supported beam subjected to a point load at the center, the reaction forces at the supports are equal and given by: R = P/2 = 1000 lb/2 = 500lb. Step 2: Shear Force at a Section. Since the beam is simply supported and the load is centered, the shear force just to the left and right of the center is equal to the reaction force: V = 500lb. Step 3: Shear Stress Calculation. The shear stress distribution in an I-beam is non-uniform, but the maximum shear stress in the web can be approximated with the following formula: $\tau_{max} = VQ/Ib$. Where: V is the shear force (500 lb); Q is the first moment of area; I is the moment of inertia; b is the thickness of the web (3/8 inch). Moment of Inertia (I):For the I-beam, $I = \frac{1}{12}(b_f h^3 - b_w(h - 2h_f)^3$. Where: $b_f = 3.5\ in$; h = 8 in; $b_w = \frac{3}{8}\ in$; $h_f = 0.5\ in$; $I = \frac{1}{12}[(3.5)(8^3) - (3/8)(8 - 2 \cdot 0.5)^3]$; $I = \frac{1}{12}[1792 - (\frac{3}{8}) \cdot 7^3]$; $I = \frac{1}{12}[1792 - 91.875]$; $I = \frac{1}{12} \cdot 1700.125$; $I = 141.677\ in^4$. First Moment of Area (Q): $Q = A' \cdot \bar{y}$. For calculating Q, we take the area of the top flange plus half the web section and find the distance from the centroid of this area to the neutral axis. $A' = b_f \cdot h_f + (b_w \cdot \frac{h}{2})$; $A' = 3.5 \cdot 0.5 + (3/8 \cdot 8/2)$; A' = 1.75 + 1.5 = 3.25 $in^2$. The distance from the centroid of the combined area to the neutral axis (center of the beam): $\bar{y} = \frac{(b_f \cdot h_f \cdot \frac{7.75}{2}) + (\frac{8}{4} \cdot b_w \cdot \frac{h}{2})}{A'} = \frac{(3.5 \cdot 0.5 \cdot 7.75/2) + (8/4 \cdot \frac{3}{8} \cdot 4)}{3.25} = \frac{(13.5625) + (0.75)}{3.25} = \frac{14.3125}{3.25} = $ 4.405 in. Thus, Q = 3.25 $\cdot$ 4.405 = 14.816 $in^3$. Finally, $\tau_{max} = \frac{500 \cdot 14.816}{141.677 \cdot 3/8} = \frac{7408}{53.254} \approx 196.9\psi$. Approximation Using the Web Area Only $\tau_{web} = \frac{V}{A} = \frac{500}{3} = 166.67\psi$. Conclusion: The exact maximum shear stress is approximately 196.9 psi and the approximation by using the web area only is 166.7 psi. **Answer: B**

**Analysis:** In this question, the Standard Setting accurately computes both the maximum transverse shear stress (196.9 psi) and the approximate shear stress (166.7 psi), yielding the correct answer of (B) 196.9 psi, 166.7 psi. In contrast, the Vision Setting provides an erroneous calculation for the shear stress but coincidentally arrives at the correct final answer.

Figure 43: Example of a different input setting in Tech and Engineering (subfield: Mechanical Engineering).

Back to Appendix

