# OpenReview forum: "MMMU-Pro: A More Robust  Multi-discipline Multimodal Understanding Benchmark"
_ICLR.cc/2025/Conference — Submitted to ICLR 2025_

### Official Review · Reviewer_9C1c · 2024-10-19

**Soundness:** 2
**Presentation:** 3
**Contribution:** 2
**Rating:** 6
**Confidence:** 5

**Summary:**

This paper MMMU-Pro presents an upgraded version of MMMU. MMMU-Pro improves MMMU via (1) filtering out questions answerable by text-only models, (2) augmenting candidate options, and (3) introducing a vision-only input setting where questions are embedded within images. All these are reasonable upgrades upon MMMU, and the paper also presents some nice insights based on existing models' performances. But the upgrades seem to be incremental and not sure whether the efforts are enough for one standalone work.

**Strengths:**

1/ The paper is easy to follow

2/ The three upgrades upon MMMU are reasonable and can better examine the MLLM's capability in making sense and reasoning about vision

3/ The paper evaluates a wide range of existing MLLMs and share some insights.

**Weaknesses:**

1/ Fig 1 compares almost 20 models, is it necessary to compare so many models? We can probably derive similar conclusions & insights based on comparing just some representative models. In this era, new MLLMs come out every a few days. I understand from the view of maintaining a benchmark/competition, it is good to be inclusive while seems not helpful to have giant table and figures like Table 1 and Fig 1.

2. Despite the 3 upgrades are reasonable, they seem to be incremental given the wonderful work of MMMU. I am not sure whether the efforts in this pro work are enough for one standalone paper. It could be just a nice technical report; while I guess if people want to pass it, I am also fine.

3. The last upgrade of vision-only input is interesting and the analysis of OCR/COT are good. While it feels to be only just scratching the surface, and if the authors would like to create a more solid work, I would expect some deeper contributions e.g. design new model/algorithm that can better make sense of such purely vision input question, create a training dataset that can power existing MLLM to do much better job on this task.

**Questions:**

Please see the above weakness

---

> ### Author Response · Authors · 2024-11-23
>
> We appreciate the reviewers’ positive feedback and recognition of our work's strengths, particularly the clarity of the paper, the reasonableness of the MMMU upgrades, and the comprehensive evaluation of existing MLLMs.
>
> > **Figure 1 and Table 1 include almost 20 models and take much space**
>
> We aim to provide a comprehensive evaluation by including most of the available models, enhancing the robustness and breadth of our results, which we believe is a **strength of our work**. However, we understand the importance of allocating space to more insightful analyses. **To address this, we remove Figure 1** in the revised version to include newly added results and deeper analyses. Thank you for this valuable suggestion.
>
> > **Incremental Contribution of MMMU-Pro**
>
> We encourage reviewers to refer to the [$\textrm{\color{blue}General Response}$](https://openreview.net/forum?id=2jTdHYuguF&noteId=MbElXRypVx) for a detailed explanation of our rationale and contributions. Thank you again for your thoughtful comments, which help refine and enhance our work.
>
> > **CoT and OCR analysis are scratching the surface**
>
> Thank you for highlighting this point. To deepen our analysis of CoT and OCR capabilities, **we have added** several key experiments and insights in the revised manuscript:
>
> 1. **Newly Added Analysis**:
>    - We conduct an **extensive error analysis** ([$\textrm{\color{blue}Figure 8}$](https://anonymous.4open.science/r/MMMU-Pro-ICLR-7174/supplementary%20material/error_type.pdf)) to quantify the impact of reasoning errors and OCR errors, revealing that reasoning across integrated modalities poses a greater challenge but OCR is not a bottleneck for capable models like GPT-4o.
>    - We investigate the influence of OCR capabilities on MMMU-Pro Vision performance ([$\textrm{\color{blue}Figure 6}$](https://anonymous.4open.science/r/MMMU-Pro-ICLR-7174/supplementary%20material/vis_ocr.pdf)) and demonstrate that good OCR performance alone does not ensure strong results in multimodal reasoning tasks.
>    - A comparative study of CoT reasoning across different subjects ([$\textrm{\color{blue}Figure 10}$](https://anonymous.4open.science/r/MMMU-Pro-ICLR-7174/supplementary%20material/gpt4o_llava72b.pdf), [$\textrm{\color{blue}Table 2}$](https://anonymous.4open.science/r/MMMU-Pro-ICLR-7174/supplementary%20material/table2.pdf)) highlights its benefits in reasoning-heavy domains like Science and Business, while showing **limited effects** in subjective areas like Art and Humanities.
>    - We discover that in the Vision input setting, GPT-4o generates fewer CoT tokens and focuses more on descriptive tokens rather than analytical reasoning ([$\textrm{\color{blue}Figure 9}$](https://anonymous.4open.science/r/MMMU-Pro-ICLR-7174/supplementary%20material/figure9.pdf)), highlighting the challenges of reasoning in integrated multimodal contexts.
>
> 2. **Key Conclusions**:
>    - OCR errors are not a bottleneck for advanced models like GPT-4o; reasoning across integrated modalities remains the primary challenge.
>    - CoT improves performance in reasoning-intensive subjects but needs **refinement** for integrated vision-text reasoning tasks.
>    - Vision-only settings reveal weaker CoT capabilities, with a tendency to focus on description rather than analytical reasoning.
>
> 3. **More Details in the General Response**:
>    These findings are elaborated in the [$\textrm{\color{blue}General Response}$](https://openreview.net/forum?id=2jTdHYuguF&noteId=tPmfhIQjHe), which includes **tables and figures for reference**. We invite reviewers to review it for a more comprehensive understanding of our contributions.
>
> > **New Tool Development to Facilitate Future Research**
>
> Thank you for highlighting this aspect. In our revised manuscript, we introduce **a new [$\textrm{\color{blue}Tool}$](https://anonymous.4open.science/r/MMMU-Pro-ICLR-7174/mmmu-pro/tool)** designed to automatically generate datasets in the MMMU-Pro style, enabling the creation of text-rich images and reasoning-intensive multimodal tasks. **This tool facilitates dataset generation and model analysis by automating question creation in real-world contexts**.
>
> This tool serves two primary purposes:
> 1. **Dataset Augmentation**: It enables efficient generation of datasets emulating real-world scenarios, supporting broader testing and fine-tuning of models.
> 2. **Analysis Facilitation**: By enabling controlled perturbations, it supports detailed analyses of model performance under varying conditions.
>
> We believe this tool **will benefit the community** by supporting future multimodal research and model development.
>
> > **More Insights and Fresh Perspectives**
>
> ### **We encourage reviewers to check out our [$\textrm{\color{blue}General Response}$](https://openreview.net/forum?id=2jTdHYuguF&noteId=tPmfhIQjHe) and the newly added results and analysis. Thank you for your valuable feedback, which has helped us significantly improve our work!**

---

> > ### Comment · Area_Chair_Xs1F · 2024-11-25
> >
> > Dear Reviewer,
> >
> > The authors have provided their responses. Could you please review them and share your feedback?
> >
> > Thank you!

---

> > > ### Author Response · Authors · 2024-11-26
> > >
> > > Dear Reviewer ```9C1c```,
> > >
> > > As today is the last day to update the paper PDF, we would like to learn if our response addresses your concerns and questions, and we invite any additional feedback or thoughts for improving our paper. If you feel that our responses resolve the issues raised, we would be grateful if you could consider reflecting this in the evaluation. We would be happy to address any further concerns or questions.
> > >
> > > Thank you again for your time and effort!

---

> > > > ### Comment · Area_Chair_Xs1F · 2024-11-30
> > > >
> > > > Dear Reviewer,
> > > >
> > > > The authors have provided their responses. Could you please review them and share your feedback?
> > > >
> > > > Thank you!

---

> > > > ### Comment · Reviewer_9C1c · 2024-11-30
> > > >
> > > > I thank authors for the response.
> > > >
> > > > For my comment 1, the authors remove the results in figure 1. I would suggest, as in my previous comment, still keep "just some representative models" to support the insights and analysis, instead of going from one extreme of giant results figure to another extreme of completely no results at all. But anyway, this is a minor writing suggestion.
> > > >
> > > > What concerns me most are still my comments 2 and 3. For 3, I thank the authors for their efforts to add a section "GUIDE FOR FUTURE MODEL TRAINING" and provide more detailed analysis etc. But still this paper does not "design new model/algorithm or create a new training dataset". Hence I still feel the technical contributions are incremental.
> > > >
> > > > Overall I think I stand with some fellow reviewers that the the novelty is somewhat limited but the authors did a lot of efforts. Hence I am fine to lift my score to acknowledge the authors' hard works while would like to encourage the authors to consider such extension work for journal in the future.

---

> > > > > ### Author Response · Authors · 2024-11-30
> > > > > **Thank you**
> > > > >
> > > > > We are pleased that the reviewer found our rebuttal to adequately address the concerns and are grateful for the **positive recognition, along with the raised score**.
> > > > >
> > > > > In the meantime, we would like to address the reviewer’s follow-up comments.
> > > > >
> > > > > > I would suggest, as in my previous comment, still keep "just some representative models" to support the insights and analysis.
> > > > >
> > > > > We will include a concise version of the original Figure 1 in the next version to address this suggestion.
> > > > >
> > > > > ---
> > > > >
> > > > > > This paper does not "design new model/algorithm or create a new training dataset."
> > > > >
> > > > > The primary focus of this paper is to create a **benchmark** that evaluates reasoning capabilities in an integrated multimodal context. Designing a new model or algorithm would likely extend beyond the scope of this work, and we intend to explore such directions in future research. However, in the revised version, we have implemented a tool for generating embedded questions within screenshots, based on the methodology used to create MMMU-Pro. This tool can be leveraged to generate new training datasets.
> > > > >
> > > > > Additionally, we would like to reiterate the motivation behind MMMU-Pro. As mentioned in the General Response, a truly capable foundation model of the future must seamlessly reason across multiple modalities. MMMU-Pro represents an early step toward achieving this paradigm, laying the groundwork for future advancements in multimodal reasoning. We believe that **MMMU-Pro serves as a fundamental capability assessment rather than an incremental update**, making it fundamentally distinct from the original MMMU dataset.
> > > > >
> > > > > ---
> > > > >
> > > > > ### We would like to thank the reviewer again for their great comments to improve the quality of the work!

---

### Official Review · Reviewer_bStp · 2024-11-03

**Soundness:** 3
**Presentation:** 3
**Contribution:** 3
**Rating:** 6
**Confidence:** 3

**Summary:**

This paper introduces a new benchmark MMMU-Pro for more accurately and rigorously assessment of a model’s true multimodal understanding and reasoning capabilities by filtering the question that can be answered by LLM dicrectly, adding more options and vision-only input. Experimental results using MMMU-Pro show a significant performance drop in existing model. This paper provides a  more rigorous evaluation tool,

**Strengths:**

1. The proposed benchmark assesses multimodal models’ understanding and reasoning capabilities in a more rigorous method.
2. The data collection pipeline are confidential due to the engagement of human.
3. Experiments are comprehensive, assessing current model's performance more accurate.

**Weaknesses:**

1. As the main contribution of this paper is a benchmark. The authors should provide more data analysis on the collected MMMU-Pro and give more insightful conclusion.
2. Apart from benchmark, the novelty is limited. This paper just tests many models on the proposed benchmark ( I don't think the exploration
 of OCR prompts and COT reasoning can be accounted as a novelty). On the other hand, the dataset collection pipeline is something more like engineering. That's the reason why I think this paper's novelty is limited. Of course, this is not the only criterion for determining whether this paper can be accepted. The proposed benchmark does make a contribution to MLLMs' understanding and reasoning capabilities. My main concern is that the workload and contribution may not be sufficient to be accepted by such a top-tier conference. I may change my rating based on the rebuttal.

**Questions:**

1. Can the author privode more insight anaysis on the proposed benchmark?
2. It's good to see the author's effort about how to improve the open-source model's performance on MMMU-Pro.

---

> ### Author Response · Authors · 2024-11-23
>
> Thank you for your insightful feedback! We acknowledge your concerns about the need for deeper data analysis and the perceived limitations in technical novelty. To address these points, we have significantly expanded on the insights derived from MMMU-Pro, including model rankings, OCR influence, reasoning error analysis, and the role of CoT across various subjects. These new findings, alongside our revised experiments and discussions, are elaborated in the [$\\textrm{\\color{blue}General Response}$](https://openreview.net/forum?id=2jTdHYuguF&noteId=tPmfhIQjHe).
>
> ### **We invite you to review the [$\\textrm{\\color{blue}General Response}$](https://openreview.net/forum?id=2jTdHYuguF&noteId=g3FG7vKTsY) for a comprehensive explanation of how MMMU-Pro provides valuable contributions to the evaluation of multimodal reasoning capabilities and guides the development of future models. Thank you again for your constructive comments and for helping us improve our work.**

---

> > ### Comment · Area_Chair_Xs1F · 2024-11-25
> >
> > Dear Reviewer,
> >
> > The authors have provided their responses. Could you please review them and share your feedback?
> >
> > Thank you!

---

> ### Comment · Reviewer_bStp · 2024-11-25
>
> Thank the author for their response, most of my concerns have been addressed. From my part, this paper can be accepted. However, It’s not enough for me to give a higher rating. Therefore, I will keep my rating unchanged which is marginally above the acceptance threshold.

---

> > ### Author Response · Authors · 2024-11-25
> > **Thank you**
> >
> > Thanks for the feedback! We are happy that most of the concerns were addressed and that **the paper is now considered acceptable**.
> >
> > Your constructive input has been invaluable in improving the quality of our work, and we’re grateful for your efforts throughout this review process. Thank you once again!

---

### Official Review · Reviewer_jvww · 2024-11-04

**Soundness:** 3
**Presentation:** 3
**Contribution:** 3
**Rating:** 5
**Confidence:** 3

**Summary:**

This paper is an extension of MMMU. The evaluation consists of three steps: 1) filtering out questions that can be answered by text-only models, 2) augmenting candidate options to reduce the chances of guessing correctly, and 3) introducing a "vision-only input" setting to challenge models to comprehend both visual and textual content simultaneously.
Experimental results demonstrate a significant drop in model performance on MMMU-Pro

**Strengths:**

Rigorous Evaluation:Using multiple LLMs to vote and filter out questions that can be solved by text-only models does enhance the benchmark’s ability to reflect more accurate visual capabilities. Overall, this benchmark is indeed more complex and better demonstrates the model’s ability to integrate text and visual information.

The findings and detailed analysis suggest avenues for future improvements, such as better vision-text integration strategies.

**Weaknesses:**

1. Expanding the number of options is an obvious way to reduce model performance. The emphasis should be on demonstrating the quality of these expanded options. However, the authors only mention this briefly with a single example.
2. The work appears straightforward, with most of the effort concentrated on non-technical human labor, making the overall contribution less innovative.

**Questions:**

Regarding the quality of options: Expanding the number of options does indeed reduce model performance. How do you ensure the quality and diversity of these additional options? If there is a method, could you elaborate further on the validation process?
﻿
Given the high construction cost of the benchmark, is it possible to reduce human effort through automated data generation or other technical means? For instance, could models be used to create new visual inputs or options?

---

> ### Author Response · Authors · 2024-11-23
>
> We sincerely thank the reviewers for their **positive** feedback and for highlighting the strengths of our work. We are particularly gratified by your recognition of our rigorous evaluation approach, including the use of LLM voting to filter text-only solvable questions, and your acknowledgment of MMMU-Pro as a more challenging and accurate benchmark for assessing multimodal integration. Additionally, we appreciate your positive remarks on our detailed findings and the future directions they suggest for improving vision-text integration strategies.
>
> > **The emphasis should be on demonstrating the quality of the expanded options.**
>
> Thanks for the suggestion! We added more details in **Appendix D** to describe the expanded options motivation and method. Note that our motivation was not to reduce the performance of the model but instead to create a more robust evaluation setup to lower the probability of finding short-cut answers and reflect the true reasoning capability of the model.
>
> Our option augmentation implemented a rigorous multi-stage validation process:
> 1. **Initial Model-Based Option Augmentation and Filtering**: As a first step, we utilized automated methods by leveraging an LLM GPT-4o to generate and another LLM Claude 3.5 to preliminarily filter the options that do not sound reasonable.
> 2. **Two Rounds of Human Review**:
> - In the first round, individual reviewers assessed the expanded options for each question. They ensured that the options were diverse, logically distinct, and free from any ambiguity that could compromise the validity of the question. If there was anything wrong with the options, they were asked to correct or augment new ones.
> - In the second round, a double-check process was conducted, where two additional human experts cross-validated each question and its options to eliminate any remaining inconsistencies or errors. This iterative process significantly enhanced the reliability and quality of the final benchmark.
> By combining automated and human validation steps, we ensured that each question's expanded options were robust and representative of the intended challenges.
>
> > **Automatic Data Generation Tool**
>
> We think our methodology for data augmentation and screenshot generation could be used for automatic training data creation as well. To this end, we have made the **scripts used for generating MMMU-Pro data publicly available** [here](https://anonymous.4open.science/r/MMMU-Pro-ICLR-7174/mmmu-pro/tool/README.md). The tool allows researchers to automate aspects of the data generation process, such as embedding questions into images and generating additional options, which can significantly reduce manual workload. By sharing these resources, we aim to empower the community to produce similar datasets more efficiently and at scale.
>
> > **Technical Contribution is Limited**
>
> ### **We appreciate the author’s feedback and re-state our technical contributions in the [$\\textrm{\\color{blue}General Response}$](https://openreview.net/forum?id=2jTdHYuguF&noteId=MbElXRypVx).**

---

> > ### Comment · Area_Chair_Xs1F · 2024-11-25
> >
> > Dear Reviewer,
> >
> > The authors have provided their responses. Could you please review them and share your feedback?
> >
> > Thank you!

---

> > > ### Author Response · Authors · 2024-11-26
> > >
> > > Dear Reviewer ```jvww```,
> > >
> > > We would like to learn if our response regarding **"The quality of the expanded options"** and **"Technical contribution"** addresses your concerns and questions, and we invite any additional feedback or thoughts for improving our paper. We would be happy to address any further concerns or questions.
> > >
> > > If you feel that our responses resolve the issues raised, we would be grateful if you could consider reflecting this in the evaluation.
> > >
> > > Thank you again for your time and effort!

---

> > > > ### Comment · Area_Chair_Xs1F · 2024-11-30
> > > >
> > > > Dear Reviewer,
> > > >
> > > > The authors have provided their responses. Could you please review them and share your feedback?
> > > >
> > > > Thank you!

---

### Official Review · Reviewer_3jHG · 2024-11-07

**Soundness:** 3
**Presentation:** 3
**Contribution:** 3
**Rating:** 6
**Confidence:** 4

**Summary:**

The paper introduces MMMU-Pro, an enhanced multimodal benchmark to rigorously test AI models’ understanding and reasoning by addressing limitations in the original MMMU benchmark. MMMU-Pro utilizes a three-step process to improve robustness: filtering questions answerable by text-only models, increasing candidate options to prevent guesswork, and implementing a vision-only input mode that embeds questions within images, thus requiring integrated visual and textual processing. Experimental results show a significant drop in model performance compared to MMMU, highlighting multimodal challenges. The study further investigates Chain of Thought prompting and OCR effectiveness, identifying areas where current models struggle, and setting directions for future multimodal research​.

**Strengths:**

- This paper addresses key limitations in existing benchmarks like MMMU. By introducing a vision-only input mode, MMMU-Pro uniquely challenges models to process visual and textual information in a more realistic, integrated manner. This work also enhances question difficulty and mitigates model reliance on shortcuts, providing an essential tool for testing and advancing multimodal AI.

- The clarity of the paper is strong, with well-organized sections detailing the benchmark's construction and evaluation.

- Additionally, the paper examines the impact of Chain of Thought prompting and OCR on performance within the proposed benchmark, further investigating the limitations present in current MLLMs.

**Weaknesses:**

- One limitation is that in the vision-only setting, images are manually captured photos and screenshots over a simulated display environment, but only differences in backgrounds, font styles, and font sizes are considered. However, the diversity of real images should also account for factors such as varied lighting conditions and different camera angles (e.g., rotated text in photos).

- While the paper discusses the Chain of Thought (CoT) prompting and OCR’s impact, these evaluations could be expanded to clarify where CoT specifically improves performance. For example, breaking down CoT's impact across different question types or modalities could reveal deeper insights, guiding future model improvements.

- Moreover, the analysis would benefit from more nuanced evaluation metrics that go beyond accuracy, such as tracking misinterpretation rates or identifying where models are most prone to visual-textual integration issues. This additional layer of analysis could provide more actionable insights for researchers looking to address specific multimodal weaknesses.

**Questions:**

Please refer to weakness section.

---

> ### Author Response · Authors · 2024-11-23
> **Rebuttal 1**
>
> We sincerely thank the reviewers for their thoughtful feedback and **positive** evaluation of our work. We are particularly encouraged by your recognition of our contributions to advancing multimodal AI benchmarking through MMMU-Pro, especially our efforts to address limitations in existing benchmarks like MMMU and to provide a more rigorous evaluation framework.
>
> > **Diversity of Images in the Vision-only Setting**
>
> We have carefully accounted for diversity and realism in our dataset design! Specifically:
> 1. **Varied Lighting and Environmental Conditions**: The photo portion of our dataset was collected by over 10 annotators using a range of display devices (e.g., phones, tablets, monitors) with varying resolutions, lighting, and environmental conditions. This ensures the dataset captures realistic variability in brightness, contrast, and ambient light, reflecting real-world scenarios.
> 2. **Annotator Variability and Subtle Rotations**: Differences in annotators’ habits during data collection naturally introduced variation in camera angles and shooting styles. As shown in Figure 3, many photos in the dataset include slight rotations, which are common in casual usage scenarios. However, we intentionally excluded extreme rotations or distortions, as these are less representative of realistic interactions and could introduce challenges unrelated to the benchmark’s primary objectives.
> Additionally, we have updated [$\\textrm{\\color{blue}Figure 2}$](https://anonymous.4open.science/r/MMMU-Pro-ICLR-7174/supplementary%20material/example.pdf) to showcase a wider range of display environments, including variations in rotation, angles, and aspect ratios, further highlighting the dataset's diversity.
>
> > **Breakdown of CoT’s impact**
>
> This is a great suggestion! We added the breakdown of CoT results for all the subjects in a representative proprietary model, GPT-4o, and an open-source model, LlavaOnevision 72B. Detailed scores for 30 subjects are added in Appendix [$\\textrm{\\color{blue}Figure 10}$](https://anonymous.4open.science/r/MMMU-Pro-ICLR-7174/supplementary%20material/gpt4o_llava72b.pdf) and [$\\textrm{\\color{blue}Table 2}$](https://anonymous.4open.science/r/MMMU-Pro-ICLR-7174/supplementary%20material/table2.pdf).
>
>
> | Domain                    | LLaVA OV 72B COT Acc | LLaVA OV 72B Direct Acc | LLaVA OV 72B Difference | GPT4o COT Acc | GPT4o Direct Acc | GPT4o Difference |
> |---------------------------|---------------|------------------|------------------|-----------------------|-------------------------|-------------------------|
> | Art and Design            | 20.42%        | 37.53%           | -17.12%         | 63.14%                | 61.55%                 | 1.58%                  |
> | Science                   | 23.89%        | 22.61%           | 1.28%           | 46.67%                | 38.46%                 | 8.22%                  |
> | Business                  | 29.26%        | 24.50%           | 4.76%           | 57.45%                | 42.79%                 | 14.66%                 |
> | Humanities and Social Science | 32.14%    | 36.60%           | -4.46%          | 60.08%                | 57.87%                 | 2.21%                  |
> | Health and Medicine       | 19.22%        | 20.78%           | -1.56%          | 49.68%                | 44.34%                 | 5.34%                  |
> | Tech and Engineering      | 22.98%        | 20.65%           | 2.33%           | 37.72%                | 23.23%                 | 14.49%                 |
>
> There are some interesting observations:
> 1. **Domains with Significant CoT Improvements**:
> The effectiveness of CoT prompting across disciplines is summarized in Table 1 and Figure 2, highlighting its varying impacts on different domains for GPT-4o and LLaVA-OneVision 72B. CoT demonstrates significant improvements in reasoning-heavy fields like Tech and Engineering, Science, and Business.
>
> 2. **Domains with Limited CoT Impact**:
> However, CoT's benefits are less pronounced or even negative in domains where subjective interpretation or direct knowledge retrieval dominates like Art, Humanities and Medicine.

---

> ### Author Response · Authors · 2024-11-23
> **Rebuttal 2**
>
> > **More Nuanced Evaluation and Analysis Beyond Accuracy**
>
> We agree that a detailed analysis beyond accuracy offers valuable insights. To this end, we conducted an in-depth error analysis by sampling 60 error cases from GPT-4 in the Vision input setting. We identified three major categories of errors, as shown in [$\\textrm{\\color{blue}Figure 8}$](https://anonymous.4open.science/r/MMMU-Pro-ICLR-7174/supplementary%20material/error_type.pdf):
>
> 1. **Reasoning Errors (46%)**: These represent the largest category of errors. In these cases, the model retrieves the correct knowledge but fails to apply logical reasoning steps to reach the correct conclusion.
>
> 2. **Perceptual Errors (27%)**: These occur when the model misinterprets visual inputs, such as misidentifying objects, texts, or relationships in the image, or overlooking key visual cues necessary for solving the problem. Notably, we manually checked for errors caused by inaccurate text recognition and found none (0%).
>
> 3. **Lack of Knowledge (25%)**: Some errors result from insufficient subject-specific or commonsense knowledge. For instance, in tasks involving the Fourier Transform, the model struggled to apply appropriate principles, highlighting knowledge gaps in specific domains.
>
> **Key Findings**
>
> From our analysis, we derive several interesting insights:
>
> - **Text Recognition as a Non-Bottleneck**: Our manual inspection confirmed that text recognition and OCR do not pose significant challenges for advanced models like GPT-4, as no errors were attributed to this factor.
>
> - **Increased Reasoning Complexity**: MMMU-Pro Vision introduces significantly more reasoning challenges compared to MMMU (46% vs. 26%, as noted in the MMMU paper). This underscores that reasoning in a modality-integrated setting is notably more difficult.
>
>
> > **New Insights and Fresh Perspective**
>
> ### **Please check out our [$\\textrm{\\color{blue}General Response}$](https://openreview.net/forum?id=2jTdHYuguF&noteId=tPmfhIQjHe) and newly added results and analysis.**

---

> > ### Comment · Area_Chair_Xs1F · 2024-11-25
> >
> > Dear Reviewer,
> >
> > The authors have provided their responses. Could you please review them and share your feedback?
> >
> > Thank you!

---

> > > ### Author Response · Authors · 2024-11-26
> > >
> > > Dear Reviewer ```3jHG```,
> > >
> > > As today is the last day to update the paper PDF, we would like to learn if our response addresses your concerns and questions, and we invite any additional feedback or thoughts for improving our paper. If you feel that our responses resolve the issues raised, we would be grateful if you could consider reflecting this in the evaluation. We would be happy to address any further concerns or questions.
> > >
> > > Thank you again for your time and effort!

---

> ### Comment · Reviewer_3jHG · 2024-11-29
>
> I believe my concerns have been largely addressed, and I am willing to maintain a positive rating for this paper. I think the paper is overall solid, but the degree of its overall novelty prevents me from giving it a higher rating.

---

> > ### Author Response · Authors · 2024-11-29
> > **Thank you**
> >
> > Thanks for the feedback! We are happy that your concerns were addressed and that **the paper is now overall solid.**
> >
> > Your constructive input has been invaluable in improving the quality of our work, and we’re grateful for your efforts throughout this review process. Thank you once again!

---

### Official Review · Reviewer_AV7u · 2024-11-08

**Soundness:** 2
**Presentation:** 3
**Contribution:** 2
**Rating:** 6
**Confidence:** 4

**Summary:**

This paper introduces MMMU-Pro, a more robust version of the MMMU benchmark. MMMU-Pro aims to more accurately assess multimodal models' true understanding and reasoning capabilities across diverse academic domains by addressing limitations found in the original MMMU. The authors achieve this through three main enhancements: (1) filtering out questions that can be answered using only text, ensuring models rely on multimodal input; (2) expanding the number of answer options to reduce reliance on guessing; and (3) introducing a vision-only input setting where questions are embedded in images, challenging models to integrate visual and textual information. These modifications result in a more rigorous benchmark that better approximates real-world scenarios. Experimental results demonstrate that MMMU-Pro challenges existing models more, revealing performance drops across multiple models and encouraging further exploration in multimodal understanding and reasoning.

**Strengths:**

- Significance: MMMU-Pro addresses critical gaps in existing benchmarks, promoting deeper multimodal understanding over shallow pattern recognition. It sets a higher evaluation standard, likely to drive future research and model development.
- Quality: The paper rigorously evaluates MMMU-Pro across multiple state-of-the-art models, showing significant performance drops that underscore the benchmark’s challenge.
- Insights: Experiments with methods like Chain of Thought reasoning and OCR prompts enrich the analysis, verifying the benchmark’s effectiveness in highlighting model limitations.

**Weaknesses:**

Unclear Justification for OCR Requirement: One of MMMU-Pro's main contributions is embedding text within images to increase difficulty by requiring OCR. However, this addition may detract from the benchmark’s core goal of evaluating multimodal understanding, as it primarily tests the model’s OCR capabilities rather than its deeper multimodal comprehension. Although it is true that embedding text within images is more realistic, whether the extra difficulty from OCR is significant for LMMs needs more justification, as the extra focus on OCR could potentially obscure the true reasoning ability of models that struggle with OCR but perform well in multimodal integration tasks.

Limited Impact on Model Performance Ranking: While it’s acceptable for a benchmark to yield similar performance scores, MMMU-Pro does not alter the ranking of current models, nor does it reveal new insights into their strengths and weaknesses. This lack of differentiation reduces the benchmark’s ability to provide fresh perspectives on model capabilities, potentially weakening its contribution as an evaluation tool.

**Questions:**

A more thorough justification for the OCR requirement and a clearer explanation of the new benchmark's significance could enhance the paper’s impact.

---

> ### Author Response · Authors · 2024-11-23
>
> We thank the reviewers for their **positive** feedback and their recognition of the strengths of our work, particularly the significance of MMMU-Pro in addressing critical gaps in multimodal benchmarks, the rigor of our evaluations, and the insightful experiments that validate the benchmark's effectiveness.
>
> > **Justification for the Motivation of MMMU-Pro**
>
> Our motivation for creating MMMU-Pro is to benchmark **reasoning capabilities** within an **integrated multimodal context**. We claim that future foundational models will increasingly adopt an **integrated modalities paradigm**, where input types—whether images, text, or videos—are no longer treated as separate modalities.
> We acknowledge the reviewer’s point that OCR challenges could obscure the true reasoning abilities of models that excel in multimodal integration but struggle with OCR. However, **existing high-performing models—both open-source and closed-source—demonstrate strong OCR capabilities but still struggle with reasoning in integrated contexts (check more details below)**. MMMU-Pro is designed to evaluate reasoning in a more realistic and challenging way, serving as an advanced test for reasoning ability in multimodal scenarios.
>
> **Human perception seamlessly integrates and transitions between textual and visual signals** through a unified interface: our eyes. If we aim for intelligent systems capable of reasoning in real-world scenarios, all forms of information—text, images, and videos—would be processed cohesively through a single interface. Such systems need to operate effectively in these more complex and dynamic environments. MMMU-Pro represents an early step toward developing intelligent systems capable of reasoning in real-world conditions.
>
> For a more detailed justification of our motivation, please refer to our [$\\textrm{\\color{blue}General Response}$](https://openreview.net/forum?id=2jTdHYuguF&noteId=Een70UMqfN).
>
> >  **OCR Requirement**
>
> Regarding the role of OCR in understanding MMMU-Pro style questions, we argue that **MMMU-Pro operates within a vision setting that requires deeper integration of visual perception, OCR, and reasoning capabilities**. We assert that **OCR is a necessary but insufficient condition** for success. As shown in the newly added [$\\textrm{\\color{blue}Figure 6}$](https://anonymous.4open.science/r/MMMU-Pro-ICLR-7174/supplementary%20material/vis_ocr.pdf), high OCR accuracy does not necessarily translate to high MMMU-Pro Vision accuracy, whereas models excelling in multimodal reasoning consistently demonstrate strong OCR performance. For example, LLaVA-OneVision-72B and Pixtral-12B achieve comparable OCR accuracy (\~85%) to InternVL2-Llama3-76B but significantly lower MMMU-Pro Vision accuracy (\~25% vs. \~38%).
> Additionally, our newly added error analysis in [$\\textrm{\\color{blue}Figure 10}$](https://anonymous.4open.science/r/MMMU-Pro-ICLR-7174/supplementary%20material/error_type.pdf) reveals that **text recognition and OCR are not the primary bottlenecks for capable models** like GPT-4o. **Among perception errors, none are caused by OCR failures**. Instead, we observe **a significantly higher proportion of reasoning errors compared to MMMU error analysis (46% vs 26%)**. This finding underscores that reasoning in a more modality-integrated setting is considerably more challenging.
>
>
> > **Limited Impact on Model Performance Ranking**
>
> We updated [$\\textrm{\\color{blue}Table 1}$](https://anonymous.4open.science/r/MMMU-Pro-ICLR-7174/supplementary%20material/table1.pdf) to show the detailed ranking changes of models when transitioning from MMMU (val) to MMMU-Pro Standard and MMMU-Pro Vision.
> On one hand, it is encouraging that rankings remain relatively stable when moving from MMMU (val) to MMMU-Pro Standard, indicating that **overfitting on MMMU (val) is not a significant issue**.
> On the other hand, we observe **a notable shift in rankings when comparing MMMU (val) to MMMU-Pro Vision**. This suggests that **MMMU-Pro Vision demands a higher level of multimodal integration capability**, which MMMU lacks. As mentioned in the [$\\textrm{\\color{blue}General Response}$](https://openreview.net/forum?id=2jTdHYuguF&noteId=tPmfhIQjHe), while the community has yet to reach a consensus, we foresee **foundational models evolving toward an integrated modalities paradigm**. In light of this, we treat MMMU-Pro Vision as the primary evaluation setting, where these shifts in performance rankings underscore the challenges of reasoning in more complex, modality-integrated scenarios.
>
> > **New Insights and Fresh Perspective**
>
> ### **Please check out our [$\\textrm{\\color{blue}General Response}$](https://openreview.net/forum?id=2jTdHYuguF&noteId=tPmfhIQjHe).**

---

> > ### Comment · Area_Chair_Xs1F · 2024-11-25
> >
> > Dear Reviewer,
> >
> > The authors have provided their responses. Could you please review them and share your feedback?
> >
> > Thank you!

---

> > ### Author Response · Authors · 2024-11-26
> >
> > Dear Reviewer **```AV7u```**,
> >
> > We would like to learn if our response addresses your concerns and questions, and we invite any additional feedback or thoughts for improving our paper. If you feel that our responses resolve the issues raised, we would be grateful if you could consider reflecting this in the evaluation. We would be happy to address any further concerns or questions.
> >
> > Thank you again for your time and effort!

---

> > > ### Comment · Reviewer_AV7u · 2024-11-29
> > >
> > > Thank you for the response and now I believe the significance of the new benchmark. The paper can be accepted.

---

### Author Response · Authors · 2024-11-23
**General Response 4**

## **Re “Technical Contribution of MMMU-Pro”**

### **1. Introducing “Reasoning on Integrated Modalities”**
We propose the concept of "reasoning on integrated modalities," which is likely to become a key focus for future foundation models. MMMU-Pro serves as a robust testbed for evaluating these reasoning capabilities. **(See more in *Re Motivation*)**.

### **2. Insights from MMMU-Pro Analysis**
Through analyzing over 20 models, we provide new insights from our overall experimental results, ablation studies, error analysis and model training. **(See more in *Re New Insights*)**.

### **3. Guidance for Future Modeling**
Based on the new insights, we offer recommendations for improving future models. **(See more in *Re Guide for Future Modeling*)**.

### **4. Automatic Data Generator**
We developed an **[$\\textrm{\\color{blue}automatic data generation tool}$](https://anonymous.4open.science/r/MMMU-Pro-ICLR-7174/mmmu-pro/tool/README.md)** based on our data curation pipeline, capable of augmenting options and generating screenshot-based questions automatically. This tool facilitates automatic training data generation and supports further analysis, such as studying the impact of different perturbation methods on accuracy.


### **We summarize the new content we revised and added in the rebuttal:**

| **Type**        | **Content**                                                                  | **Position**                                                                | **Reviewers**          |
|:---------------|:----------------------------------------------------------------------------|:----------------------------------------------------------------------------|:----------------------|
| New Result      | Influence of Vision Encoders                                                 | Future Guide (Line 497-509); [$\\textrm{\\color{blue}Table 4}$](https://anonymous.4open.science/r/MMMU-Pro-ICLR-7174/supplementary%20material/table4.pdf)                                                          | bStp                   |
| New Result      | Error Analysis                                                               | Experiments (Line 469-476); [$\\textrm{\\color{blue}Figure 8}$](https://anonymous.4open.science/r/MMMU-Pro-ICLR-7174/supplementary%20material/error_type.pdf)                                        | 3jHG, bStp             |
| New Result      | CoT’s impact on 30 subjects                                                  | Experiments (Line 337-344); [$\\textrm{\\color{blue}Table 2}$](https://anonymous.4open.science/r/MMMU-Pro-ICLR-7174/supplementary%20material/table2.pdf), [$\\textrm{\\color{blue}Figure 10}$](https://anonymous.4open.science/r/MMMU-Pro-ICLR-7174/supplementary%20material/gpt4o_llava72b.pdf)                                | 3jHG, bStp, 9C1c, jvww |
| New Result      | The influence of OCR and Reasoning capability on MMMU-Pro Vision             | Experiments (Line 411-421); [$\\textrm{\\color{blue}Figure 6}$](https://anonymous.4open.science/r/MMMU-Pro-ICLR-7174/supplementary%20material/vis_ocr.pdf)                                         | AV7u, bStp, 9C1c, jvww |
| New Tool        | Automatic screenshot generation of MMMU-Pro style questions| [$\\textrm{\\color{blue}Tool}$](https://anonymous.4open.science/r/MMMU-Pro-ICLR-7174/mmmu-pro/tool/README.md) | 9C1c, jvww |
| New Discussions | Guide for future modeling | Section 4  | bStp |
| Revised Result  | Vision Input as the primary setting of MMMU-Pro. Rank significantly changes. | [$\\textrm{\\color{blue}Table 1}$](https://anonymous.4open.science/r/MMMU-Pro-ICLR-7174/supplementary%20material/table1.pdf)                                                                      | AV7u, 9C1c             |
| Revised Figure  | Replace examples to show diverse display environment                         | [$\\textrm{\\color{blue}Figure 2}$](https://anonymous.4open.science/r/MMMU-Pro-ICLR-7174/supplementary%20material/example.pdf)                                                                     | 3jHG                   |



We sincerely appreciate the insightful comments, which have prompted us to reevaluate the positioning of this work and significantly improve its clarity, soundness, and technical contributions. We believe that MMMU-Pro holds great potential to guide the development and training of next-generation multimodal models. We firmly advocate that a **truly capable foundation model of the future must seamlessly transition between different modalities**. MMMU-Pro is explicitly designed to rigorously evaluate this capability. Unlike standalone OCR or reasoning benchmarks that fail to capture the essence of multimodal integration, MMMU-Pro challenges models to perform seamless reasoning across integrated modalities.

### **If our rebuttal addresses your concerns and proves useful, we kindly ask you to consider adjusting your review and scores. We remain open and eager to address any further concerns or questions you may have.**

---

### Author Response · Authors · 2024-11-23
**General Response 3**

## **Re: "Guide for Future Model Training"**
We added a new **section 4 “Guide for Future Model Training”** in the revised manuscript.

### **1. Scaling of LLM Backbones**
- Scaling LLMs improves both perception and reasoning capabilities, as shown in performance trends:
  - Examples: GPT-4o > GPT-4o mini, Llava-OneVision-72B > Llava-OneVision-7B, InternVL2-78B > InternVL2-8B.

### **2. Better Vision Encoders that Highlight Visual Representation Learning**
- Vision encoder choice impacts the performance of MMMU-Pro. Self-supervised encoders like DINOv2 show better performance than language-supervised encoders like SigLip.
- Future encoders may combine strengths of language-supervised and self-supervised approaches.

### **3. Better Vision-Text Integration**
- **Cross-modal integration remains challenging** for tasks requiring deep understanding.
- Improved cross-modal feature fusion techniques are promising for bridging this gap.

### **4. CoT Data Generation**
- **CoT prompting significantly enhances reasoning-heavy subjects** like **Tech and Engineering** but has limited or even negative effects in subjective areas like **Art and Design**.
- Generating diverse reasoning-intensive CoT datasets can improve model performance, especially in reasoning-heavy domains.

### **5. Text-Rich Image Generation**
- **Strong OCR and reasoning performance do not guarantee success on MMMU-Pro Vision** due to a lack of training data with text-rich images in reasoning-intensive scenarios.
- A new [$\\textrm{\\color{blue}tool}$](https://anonymous.4open.science/r/MMMU-Pro-ICLR-7174/mmmu-pro/tool/README.md) based on the MMMU-Pro Vision annotation process was developed to **generate screenshots embedding questions and images**.
- This tool can scale the generation of reasoning-specific datasets, improving models' ability to handle integrated visual-text tasks.

By addressing these areas, future models can overcome limitations identified by MMMU-Pro and advance multimodal understanding and reasoning capabilities.

---

### Author Response · Authors · 2024-11-23
**General Response 2**

## **Re “New insights from MMMU-Pro”**

### **1. Model Rankings**
- Model **rankings change significantly from MMMU to MMMU-Pro Vision** as noted in revised [$\\textrm{\\color{blue}Table 1}$](https://anonymous.4open.science/r/MMMU-Pro-ICLR-7174/supplementary%20material/table1.pdf), particularly for open-source models. This illustrates reasoning in an integrated multimodal context is more challenging compared with separate multimodal inputs.

### **2. OCR Influence**
- Good OCR ability is **necessary but insufficient** condition for high accuracy on MMMU-Pro Vision.
- High OCR accuracy does not correlate strongly with high performance on MMMU-Pro Vision (e.g., Llava OV-72B, Pixtral-12B), as shown in the newly-added [$\\textrm{\\color{blue}Figure 6}$](https://anonymous.4open.science/r/MMMU-Pro-ICLR-7174/supplementary%20material/vis_ocr.pdf).

### **3. Error Analysis**
- As shown in the new [$\\textrm{\\color{blue}Figure 8}$](https://anonymous.4open.science/r/MMMU-Pro-ICLR-7174/supplementary%20material/error_type.pdf), **Reasoning errors increase significantly in MMMU-Pro Vision compared to MMMU** (46% vs. 26%), highlighting that while MMMU-Pro primarily aims to benchmark the model's reasoning capabilities, **reasoning across integrated modalities remains a greater challenge**.
- OCR Errors: **OCR does not pose a bottleneck for highly capable models like GPT-4o** (no errors due to incorrect text recognition).
- These findings emphasize that multimodal reasoning requires capabilities beyond OCR in integrated modality environments.

### **4. CoT Impact Across Subjects**
- As shown in the new [$\\textrm{\\color{blue}Figure 10}$](https://anonymous.4open.science/r/MMMU-Pro-ICLR-7174/supplementary%20material/gpt4o_llava72b.pdf) and [$\\textrm{\\color{blue}Table 2}$](https://anonymous.4open.science/r/MMMU-Pro-ICLR-7174/supplementary%20material/table2.pdf), CoT prompting significantly improves performance in reasoning-heavy subjects like Science, Business, and Engineering but shows limited or negative benefits in subjects like Art, Humanities and Health.

### **5. Weaker CoT Ability in Integrated Multimodal Reasoning**
An intriguing phenomenon was observed in [$\\textrm{\\color{blue}Figure 9}$](https://anonymous.4open.science/r/MMMU-Pro-ICLR-7174/supplementary%20material/figure9.pdf): **GPT-4o typically generates fewer CoT tokens in the Vision input setting** compared to the Standard setting. Additionally, the model tends to **allocate more tokens to description rather than analysis**. This highlights that reasoning within integrated multimodal contexts is more challenging, and the CoT ability in such scenarios is comparatively weaker.

### **6. Influence of Vision Encoders**
- In [$\\textrm{\\color{blue}Table 4}$](https://anonymous.4open.science/r/MMMU-Pro-ICLR-7174/supplementary%20material/table4.pdf), we train two MLLMs on 1M samples, utilizing different types of vision encoders: a language-supervised encoder (Siglip) and a self-supervised encoder (DINOv2).
- The self-supervised vision encoder, DINOv2, which emphasizes visual feature learning, achieves better performance on MMMU-Pro Vision. In contrast, the language-supervised vision encoder, Siglip, performs better on MMMU (val). Future work may focus on further enhancing visual feature learning while exploring the integration of language-based training objectives with self-supervised training objectives.


| **Method**               | **MMMU (Val)** | **MMMU-Pro Vision** |
|----------------------|------------|-----------------|
| DINOv2 ViT-G-14     | 37.1       | 17.4            |
| Siglip ViT-SO400M-14 | 37.9       | 16.7            |

---

### Author Response · Authors · 2024-11-23
**General Response 1**

We appreciate all the constructive comments by the reviewers. Most of the major concerns lie in the position of MMMU-Pro compared with MMM, technical contributions, and what kind of new insights we bring by building MMMU-Pro. We respond to these major concerns in the general response and other comments in each individual response.


## **Re “Motivation of building MMMU-Pro”**

Beyond creating a more robust dataset—by filtering for text-only answerable questions and incorporating more diverse options—the primary motivation for introducing MMMU-Pro is to **benchmark reasoning capabilities in an integrated multimodal context**. While the community has yet to reach a consensus, we anticipate that future foundational models will increasingly adopt an integrated modalities paradigm: models will no longer distinguish between input modalities, whether they are images, text, or videos.

This prediction is inspired by how humans perceive and reason about the world. When processing visual signals, **humans do not explicitly segregate them by modality; instead, we seamlessly integrate and transition between textual and visual inputs**. These inputs are perceived holistically through a unified sensory interface—our eyes. Similarly, we believe a foundational model—or any intelligent system—should exhibit this capability. Such models should not be constrained to processing separate multimodal inputs but should handle integrated inputs (e.g., a single screenshot combining text and images) with equal proficiency, seamlessly reasoning across them.
MMMU-Pro represents one of the early steps toward realizing this paradigm, paving the way for future advancements in multimodal reasoning.

---

### Meta-Review · Area_Chair_Xs1F · 2024-12-14

**Metareview:**

This paper introduces MMMU-Pro, an enhanced multimodal benchmark that addresses limitations of the original MMMU by ensuring reliance on multimodal inputs, expanding answer options, and introducing vision-only settings, providing a more rigorous evaluation of multimodal models' reasoning capabilities.

The paper is generally well-written. The original MMMU has already brought significant impacts to the community. This work is an extension of that, which will be useful for the community as well.

The paper has generally received scores of 6, with the exception of Reviewer jvww, who did not follow up after the rebuttal.

Upon reviewing all responses, AC noticed that several reviewers were not very excited or enthusiastic about the work, even while supporting its acceptance. They expressed concerns regarding the paper's limited novelty. Specifically, the concerns include:

Similarities between the benchmark dataset and the existing MMMU benchmark. Limited insights derived from extensive model comparisons. The annotations and dataset curation involve significant human effort, with minimal technical innovation introduced.

After discussion among AC and reviewers, due to the limited novelty, the AC decided to reject the paper. However, the authors are highly encouraged to submit an extended version of MMMU to a journal, which often accommodates expansions of original work.

**Additional Comments On Reviewer Discussion:**

The paper has generally received scores of 6, with the exception of Reviewer jvww, who did not follow up after the rebuttal.

Upon reviewing all responses, AC noticed that several reviewers were not very excited or enthusiastic about the work, even while supporting its acceptance. They expressed concerns regarding the paper's limited novelty. Specifically, the concerns include:

Similarities between the benchmark dataset and the existing MMMU benchmark. Limited insights derived from extensive model comparisons. The annotations and dataset curation involve significant human effort, with minimal technical innovation introduced.

Given the current status of the paper after the rebuttal, a rejection decision is made as the novelty is insufficient for top-tier conferences, such as ICLR.

---

### Decision · Program_Chairs · 2025-01-22

Reject